# Universal Latent Homeomorphic Manifolds: A Framework for Cross-Domain Representation Unification

**Tong Wu**[*]                                                                *tong.wu@ucf.edu*
*University of Central Florida, Orlando, USA*

**Tayab Uddin Wara**                                                *tayabuddin.wara@ucf.edu*
*University of Central Florida, Orlando, USA*

**Daniel Hernandez**                                                *daniel.hernandez4@ucf.edu*
*University of Central Florida, Orlando, USA*

**Sidong Lei**                                                                *sidong.lei@ucf.edu*
*University of Central Florida, Orlando, USA*

**Reviewed on OpenReview:** *https://openreview.net/forum?id=YoZSpRWhZH*

## Abstract

We present the Universal Latent Homeomorphic Manifold (ULHM), a framework that unifies semantic representations (e.g., human descriptions, diagnostic labels) and observation-driven machine representations (e.g., pixel intensities, sensor readings) into a single latent structure. Despite originating from fundamentally different pathways, both modalities capture the same underlying reality. We establish *homeomorphism*, a continuous bijection preserving topological structure, as the mathematical criterion for determining when latent manifolds induced by different semantic-observation pairs can be rigorously unified. When this homeomorphic criterion is satisfied, it enables three critical applications: (1) semantic-guided sparse recovery from incomplete observations, (2) cross-domain transfer learning with empirically assessed structural compatibility, and (3) transductive zero-shot compositional learning via valid transfer from semantic to observation space. Our framework learns continuous manifold-to-manifold transformations through conditional variational inference, with training objectives explicitly designed to enforce bi-Lipschitz homeomorphic properties. We develop practical verification algorithms, including trust, continuity, and Wasserstein distance metrics, that empirically indicate whether the learned representations exhibit properties consistent with homeomorphic structure from finite samples. Experiments demonstrate substantial improvements over state-of-the-art (SOTA) baselines: (1) sparse recovery from 8% of pixels with much lower MSE than SOTA on CelebA under noise, (2) cross-domain transfer achieving 86.73% MNIST→Fashion-MNIST accuracy without retraining, and (3) transductive zero-shot classification achieving 78.76% on CIFAR-10, exceeding prior work by 16.66%. Critically, the homeomorphism criterion determines when different semantic-observation pairs share compatible latent structure, enabling principled unification into shared representations within the tested domains and suggesting a structured basis for decomposing broad models into domain-specific components.

---

[*]Corresponding author.

# 1 Introduction

## 1.1 Background and Motivation

Deep learning models learn representations through pathways fundamentally different from semantic descriptions (Bengio et al., 2013; Locatello et al., 2019). While traditional paradigms prioritize mapping data to specific task targets, they often lack the structural grounding inherent in semantic frameworks. Semantic information captures high-level conceptual structure through annotations. For example, describing a person as "wearing glasses" or "having black hair," recognizing that handwritten digits 0, 6, 9 have one circle while 8 has two, or labeling medical images as "irregular morphology." In contrast, deep learning systems extract hierarchical features directly from raw measurements (pixel intensities, time-series values, sensor outputs, or spectral data) through statistical pattern recognition. Despite these different origins, both representations aim to capture the same underlying reality (Huh et al., 2024). This divergence raises two fundamental questions: *Can semantic and data-driven representations be unified into a single latent manifold where both converge to a shared geometric structure? Furthermore, under what mathematical conditions can we establish a universal latent manifold that enables a learning model to remain structurally consistent, suggesting a structured path toward decomposing broad models into domain-specific components?*

Answering these questions has direct implications for three practical representation learning challenges. While we primarily validate the framework on image benchmarks in this paper, the formulation is designed to extend to any domain where semantic information (such as human descriptions, state variables in control systems, or symbolic labels) coexists with raw measurements, including signal processing, network control, and sensor fusion. To address these diverse applications, our theoretical framework provides a rigorous foundation for overcoming the following three technical bottlenecks.

- **Sparse recovery with semantic priors.** During training, the model observes complete measurements paired with rich semantic descriptions. In contrast, deployment relies on highly undersampled measurements without accompanying semantic annotations. This creates an asymmetric scenario where high-level knowledge is available during learning but strictly absent during inference. This leads to a fundamental question: Can we establish a latent mapping that achieves effectively lossless compression, and what are the theoretical limits of such recovery? Specifically, can a learning model encode enough structural logic during training that a drastic reduction in data does not result in a loss of signal integrity, allowing the network to implicitly leverage high-level constraints to reconstruct full-dimensional signals from incomplete observations?

- **Cross Domain Geometric Alignment and Domain Transfer.** Multiple related semantic observation pairs, such as clinical descriptions paired with different imaging modalities, intuitively share an underlying latent structure. However, no principled mathematical criterion exists to determine whether the manifolds induced by these heterogeneous pairs can be rigorously unified. Current ad hoc alignment methods remain fragile, often failing when distributions shift or new modalities emerge. This raises a fundamental question: Under what conditions can diverse semantic observation pairs be unified into a single geometric framework? Specifically, can we establish a universal latent structure that remains consistent across disparate domains, providing a formal basis for the cross modality transfer of knowledge?

- **Zero-shot attribute composition.** Models learn semantic attributes from training examples where each attribute typically appears in isolation. However, training data rarely cover all $2^K$ possible combinations of $K$ attributes. At test time, images often exhibit novel combinations never jointly observed, creating a zero shot scenario where standard learning models fail because they treat attributes as independent entities without capturing their underlying structural relationships. This leads to a fundamental question: Can the latent manifold components be structured to enable valid zero shot composition within the observation space? Specifically, can the manifold capture the geometric logic of attribute interactions such that novel combinations remain consistent with the learned manifold structure, even when they were never explicitly seen during training? We address this in a transductive zero-shot setting where the model observes the visual content of unseen classes during training but withholds their labels from the classifier.

We establish homeomorphism as the fundamental criterion for semantic observation unification. In our framework, each semantic observation pair induces a latent manifold through representation learning. When the latent manifolds originating from disparate pairs are homeomorphic, sharing an identical topological structure through a continuous and invertible bijection, they can be rigorously unified into a universal representation space. This condition provides a unified mathematical foundation for addressing the three previously identified challenges: semantic priors can guide sparse recovery under empirically assessed structural conditions, heterogeneous data can be integrated with empirically assessed structural conditions for transfer learning, and compositional reasoning in the semantic space transfers validly to the observation space for transductive zero shot learning.

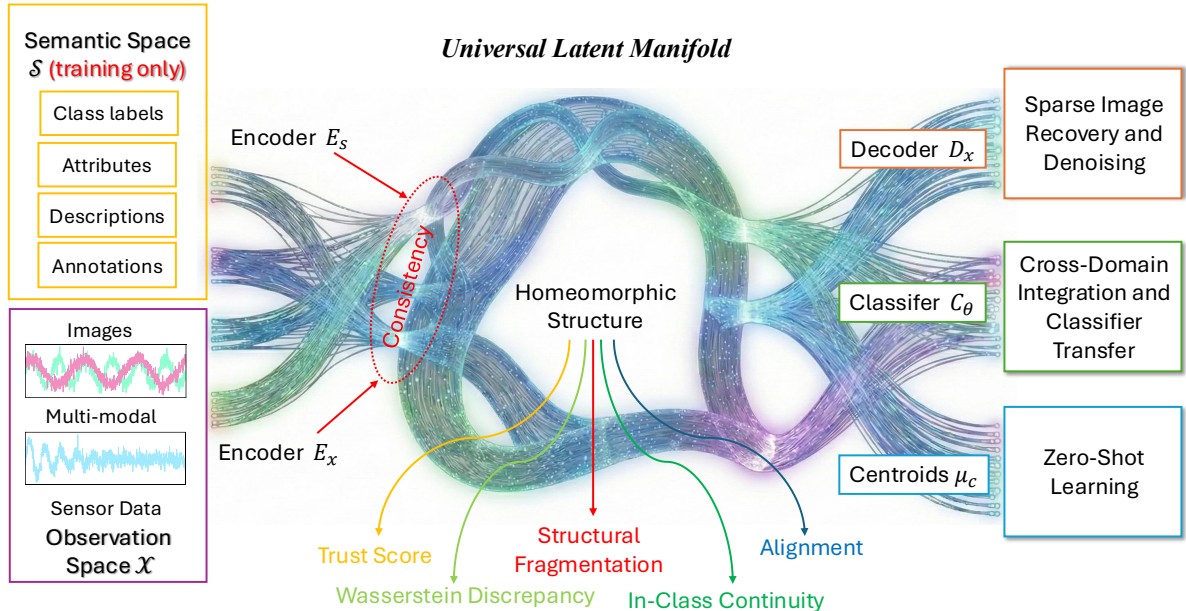

Figure 1: Overview of the proposed Universal Latent Manifold (ULHM) framework. The architecture bridges the **Semantic Space** $\mathcal{S}$ (utilized during training with class labels, attributes, and descriptions) and the **Observation Space** $\mathcal{X}$ (comprising images, multi-modal inputs, and sensor data). This integration enables diverse downstream capabilities, including Sparse Image Recovery and Denoising, Cross-Domain Integration and Transfer, and Zero-Shot Learning.

## 1.2 Related Work

We review prior work on latent geometry and manifold-based embeddings, organizing the discussion around five key perspectives that align with our methodological contributions: universal geometry of embeddings, sparse recovery with semantic priors, cross-domain transfer learning, and zero-shot compositional generalization.

### 1.2.1 Universal Latent Manifold

Existing approaches to universal representations remain incomplete. Recent empirical work on the Platonic Representation Hypothesis (Huh et al., 2024) observes that neural networks trained on different modalities converge toward aligned geometric structures. While subsequent research has provided information-geometric perspectives (Lobashev), evidence of universal weight subspaces (Kaushik et al., 2025), and methods to harness universal embedding geometries (Jha et al., 2025), these observations lack a rigorous formalization of *when* such alignment occurs and *why* it succeeds or fails. Alignment methods based on optimal transport (Alvarez-Melis & Jaakkola, 2018; Grave et al., 2019; Chen et al., 2020a) and representation similarity metrics (Kornblith et al., 2019) can match embedding spaces across modalities, yet they provide no criterion for determining structural compatibility prior to alignment. Similarly, multi-view learning methods (Wang et al., 2015) construct shared representations but assume fusion is always valid, which leads to failure when underlying structures are fundamentally incompatible. Furthermore, while recent advances in manifold learning with topological and geometric regularization (Wang & Zhou, 2025; Rhodes et al., 2025) and analyses of

neural representation topology (Lin & Kriegeskorte, 2024) have improved our understanding of latent structure, these approaches typically focus on reconstruction and local geometry without the explicit semantic integration required for compositional reasoning.

**Our Difference:** The critical gap in these existing methods is that they analyze input and output distributions separately. For instance, two semantic-observation pairs may possess identical input geometries but map to opposite output labels (e.g., high intensity representing "positive" in one modality and "negative" in another). Even if both form manifolds of the same shape, existing approaches cannot distinguish whether they can be meaningfully unified. Our framework addresses this by analyzing the *joint* structure of semantic-observation pairs. We establish *homeomorphism* not just as a property to be observed, but as the rigorous mathematical criterion to determine *validity*, providing an empirical criterion for assessing whether the latent manifolds induced by different pairs exhibit structural compatibility consistent with topological equivalence.

### 1.2.2 Homeomorphic Learning

The concept of homeomorphism has been increasingly utilized to model structural equivalence in machine learning. In unsupervised domain adaptation, Zhou et al. (Zhou et al., 2023) employ invertible neural networks to construct a homeomorphic alignment between source and target feature distributions, ensuring that discriminative structures are preserved during transfer. In the context of optimization, homeomorphic projection methods (Liang et al., 2024) map unconstrained neural network outputs onto manifolds defined by physical requirements to ensure solution validity. Other approaches leverage homeomorphism to preserve intrinsic data topology during dimensionality reduction (Moor et al., 2020) or to resolve the manifold mismatch between complex data distributions and simple latent priors in variational autoencoders (Falorsi et al., 2018). Earlier work on Homeomorphic Manifold Analysis (Elgammal & Lee, 2013) utilized these mappings to separate style and content within a single domain. However, these methods typically treat homeomorphism as a mapping tool to align marginal distributions or enforce **assumed latent topologies**. By relying on these predefined constraints, such approaches effectively force the data to fit a geometric shape guessed beforehand (e.g., a unit circle or torus), rather than discovering the true unified structure inherent in the joint semantic-observation relationship. For instance, two datasets may have identical feature distributions but opposite label semantics (e.g., high temperature indicating 'stable' in one system but 'critical' in another). Our framework addresses this by establishing homeomorphism not merely as a mapping mechanism, but as a rigorous criterion for the *joint semantic-observation structure*, enabling us to reject unification when the underlying mappings are topologically incompatible despite geometric similarity. We demonstrate this rejection capability empirically in Section 4, where we show that the verification protocol correctly identifies incompatible domain pairs (e.g., MNIST vs. random noise, label-shuffled controls) and produces near-chance transfer accuracy when bypassed.

### 1.2.3 Sparse Recovery and Compressed Sensing

Traditional compressed sensing methods recover sparse signals using hand-crafted priors such as wavelet bases or total variation. While foundational deep generative approaches (Bora et al., 2017) and unsupervised methods like Deep Image Prior (Ulyanov et al., 2018) demonstrated that learnable priors could effectively model reconstruction mappings, recent advancements have shifted toward score-based generative models (Song et al., 2022) and diffusion restoration methods (Kawar et al., 2022; Chung et al., 2022), which solve inverse problems by iteratively refining noisy distributions, and transformer-based architectures (Liang et al., 2021) that exploit long-range dependencies for restoration. Although some recent works incorporate semantic information (such as semantic-aware sensing matrices (Zhang et al., 2023) or latent diffusion models conditioned on explicit semantic inputs (Rombach et al., 2022)), these approaches typically lack rigorous topological guarantees and often require semantic availability during inference. In contrast, our approach learns homeomorphic mappings between observation and latent spaces through topology-preserving training objectives, approximating lossless topological compression. Post-training verification then provides finite-sample evidence consistent with this structural compatibility, supporting the claim that semantic priors learned during training can implicitly guide sparse reconstruction at test time without explicit semantic input.

### 1.2.4  Cross-Domain Transfer and Universal Classifiers

Recent advances in transfer learning have shifted from full fine-tuning toward parameter-efficient modularity. Techniques such as Low-Rank Adaptation (LoRA)(Hu et al., 2022) and adapter-based methods(Houlsby et al., 2019) enable domain transfer by optimizing sparse updates within fixed weight spaces. Concurrently, model merging approaches like Task Arithmetic (Ilharco et al., 2023) and TIES-Merging (Yadav et al., 2023) demonstrate that learned capabilities can be transferred or combined through linear parameter manipulation. This transferability has been attributed to the emergence of universal weight subspaces (Kaushik et al., 2025), with extensions to graph learning frameworks that accommodate topological variations (Wu et al., 2025).

While these methods achieve remarkable parameter efficiency, often reducing trainable parameters by orders of magnitude, they implicitly assume that source and target domains share compatible underlying geometry (Jha et al., 2025). Critically, even methods designed to handle structural variations lack explicit mechanisms to both *enforce* and *verify* topological compatibility. Our framework addresses this dual gap: training objectives actively enforce bi-Lipschitz homeomorphic properties during learning, while post-training verification assesses structural compatibility before deployment. When latent manifolds exhibit empirically indicated structural compatibility consistent with homeomorphism, classifier decision boundaries can be continuously deformed across domains, enabling principled zero-shot transfer with empirically supported structural conditions rather than unvalidated heuristics.

### 1.2.5  Zero-Shot Learning and Compositional Generalization

Zero-shot learning methods recognize unseen classes by composing semantic attributes, a paradigm established by foundational attribute-based classification models (Pourpanah et al., 2022; Lampert et al., 2013). To overcome the bias toward seen classes inherent in these mapping-based approaches, generative frameworks (Xian et al., 2018) were developed to synthesize visual features for unseen categories, effectively converting zero-shot tasks into supervised learning problems. However, standard generative models often struggle with *compositional generalization*, failing to capture the causal structure required to distinguish valid attribute-object combinations from spurious correlations (Atzmon et al., 2020). Recent state-of-the-art methods have addressed this limitation through compositional soft prompting (Nayak et al., 2023), which optimizes learnable context vectors to disentangle attributes in vision-language models, and through diffusion-based classifiers (Li et al., 2023) that exploit density estimation for robust zero-shot recognition. Despite these advancements, these approaches rely on statistical alignment without topological guarantees. Our framework addresses this by establishing homeomorphic structure during training through topology-preserving objectives and assessing it post-training through empirical diagnostics. This ensures that unseen classes occupy geometrically separated regions, enabling zero-shot classification in a transductive setting where unseen-class images are available during encoder training via nearest-centroid assignment without requiring explicit supervision on unseen classes.

### 1.3  Contributions

To address these challenges, we propose the manifold learning framework illustrated in Fig. 1, which unifies semantic descriptions and raw observations through learned mappings and empirical structural-compatibility diagnostics. The key insight is that when semantic information and observations originate from the same underlying reality, their latent representations should be homeomorphic, preserving topological structure through continuous bijection. We establish homeomorphism as the rigorous mathematical criterion for determining when latent manifolds from different semantic-observation pairs can be validly unified. Unlike existing methods that analyze distributions separately without mechanisms to achieve or verify structural compatibility, our framework learns homeomorphic mappings through topology-preserving training objectives while providing practical verification algorithms to assess whether the learned representations exhibit properties consistent with homeomorphic structure from finite samples, enabling us to reject incompatible unifications before deployment. The main contributions of this work are as follows.

- **Asymmetric semantic-observation manifold learning.** We propose a framework that learns from semantic-observation pairs during training but performs inference from observations alone

at test time. By embedding semantic structure as geometric constraints in the latent manifold through joint reconstruction and consistency objectives, the framework enables semantic priors to implicitly guide reconstruction and reasoning without requiring semantic annotations at deployment. This asymmetric paradigm, which leverages semantic supervision during training while maintaining observation-only inference, provides a principled approach to incorporating high-level knowledge into observation-driven systems.

- **Homeomorphism as both optimization target and verification criterion.** We establish homeomorphism as the mathematical condition under which latent manifolds induced by different semantic-observation pairs can be rigorously unified. The ULHM training objectives are explicitly designed to enforce bi-Lipschitz homeomorphic properties during learning, with post-training diagnostics assessing finite-sample evidence consistent with homeomorphic structure preservation. This criterion ensures that the joint relational structure between semantics and observations is preserved through continuous bijection, providing formal conditions under idealized assumptions (Theorems 1–2) and empirical diagnostics from finite samples for assessing when cross-domain transfer, sparse recovery with semantic priors, and compositional reasoning are expected to remain valid.

- **Sparse recovery via homeomorphic compression limits.** We establish that when the encoder-decoder mapping achieves homeomorphism through topology-preserving training objectives, the latent representation is designed to approximate a lossless topological compression preserving the structural integrity of signals. Post-training diagnostics then assess whether the learned representation exhibits properties consistent with this homeomorphic regime, which enables us to characterize the practical limits of compression: the framework can reconstruct full-dimensional signals from highly undersampled observations by traversing the manifold's intrinsic geometry. Homeomorphism thus provides a theoretical criterion for compression limits under idealized conditions and, when the empirical diagnostics pass, supports practical superior performance in sparse recovery without requiring semantic annotations at test time.

- **Homeomorphic structure enables zero-shot cross-domain transfer.** We establish that when latent manifolds achieve homeomorphic structure through topology-preserving training objectives, with post-training diagnostics indicating structural compatibility consistent with homeomorphism, classifier decision boundaries can be continuously deformed across domains, enabling zero-shot transfer. A universal classifier trained on one domain transfers directly to heterogeneous domains without retraining, substantially outperforming existing domain adaptation methods under our experimental protocol. Empirical results indicate that empirically indicated structural compatibility correctly predicts when zero-shot transfer will succeed.

- **Zero-shot classification via topological preservation.** We demonstrate that homeomorphic structure, learned through topology-preserving objectives (particularly local consistency constraints) and assessed post-training through empirical diagnostics, enables transductive zero-shot recognition of unseen classes through nearest-centroid classification in the learned latent manifold. By preserving topological structure, the framework achieves superior performance compared to SOTA zero-shot learning methods across multiple datasets, validating that preserved topology enables geometric separation without explicit supervision on unseen classes. All baselines are evaluated under matched transductive access to ensure fair comparison (see Section 4).

The remainder of this paper is organized as follows. Section II presents the mathematical framework for universal latent homeomorphic manifolds. Section III develops the manifold learning architecture with topology-preserving training objectives and homeomorphism verification methods. Section IV presents experimental results demonstrating the framework's effectiveness on sparse recovery, heterogeneous data integration, and zero-shot learning. Section V concludes with discussions and future directions.

## 2 Universal Latent Homeomorphic Manifold Framework

We present the mathematical framework for learning universal latent homeomorphic manifolds that unify semantic and observation representations. The key insight is treating each semantic-observation pair as induc-

ing a latent manifold through representation learning with topology-preserving objectives, then empirically assessing whether the learned manifolds from different pairs exhibit properties consistent with homeomorphic structure. When homeomorphism is both learned and empirically indicated by the diagnostic metrics, the manifolds share topological structure and can be unified into a universal representation space.

## 2.1 Problem Formulation and Notation

Consider a scenario where we have access to heterogeneous information about the same underlying phenomenon: *semantic descriptions* and *raw observations*. These information sources can take multiple forms depending on the application domain. For handwritten digit recognition, semantic descriptions might include structural attributes like "has one closed loop" alongside class labels, while raw observations consist of pixel intensities from MNIST images. For imaging reconstruction, observations might include sparse measurements and sampling masks, while semantics encompass ground truth complete images and diagnostic annotations. For multi-sensor systems, observations may involve heterogeneous sensor readings at different spatial or temporal resolutions, while semantics provide state labels or physical parameters.

We formally define the following notation to accommodate this heterogeneity. Let $\mathcal{X} = \{\mathcal{X}^{(1)}, \mathcal{X}^{(2)}, \ldots, \mathcal{X}^{(M)}\}$ denote a collection of observation spaces, where each $\mathcal{X}^{(m)}$ represents a distinct type of measurement. For example:

- $\mathcal{X}^{(1)} \subset \mathbb{R}^{D_1}$: sparse or incomplete measurements (e.g., sampled pixel values, sensor readings)

- $\mathcal{X}^{(2)} \subset \{0,1\}^{D_2}$: observation mask matrix indicating which measurements are available

- $\mathcal{X}^{(3)} \subset \mathbb{R}^{D_3}$: auxiliary measurements from complementary modalities

We denote $\mathbf{x} = \{x^{(1)}, x^{(2)}, \ldots, x^{(M)}\}$ as a multi-modal observation tuple, where $x^{(m)} \in \mathcal{X}^{(m)}$. Not all modalities need be present for every sample. Similarly, let $\mathcal{S} = \{\mathcal{S}^{(1)}, \mathcal{S}^{(2)}, \ldots, \mathcal{S}^{(L)}\}$ denote a collection of semantic spaces capturing different forms of high-level information. For example:

- $\mathcal{S}^{(1)} \subset \mathbb{R}^{H \times W \times C}$: ground truth complete signals or images

- $\mathcal{S}^{(2)} \subset \{0,1\}^K$: binary attribute vectors or multi-label annotations

- $\mathcal{S}^{(3)} \subset \mathbb{R}^{D_{\text{text}}}$: text embeddings from natural language descriptions

- $\mathcal{S}^{(4)} \subset \mathbb{Z}_+$: discrete class labels or categorical variables

We denote $\mathbf{s} = \{s^{(1)}, s^{(2)}, \ldots, s^{(L)}\}$ as a multi-modal semantic tuple, where $s^{(l)} \in \mathcal{S}^{(l)}$.

Let $\mathcal{Z} \subset \mathbb{R}^d$ denote the universal latent space containing the *latent manifold* $\mathcal{M}_z \subset \mathcal{Z}$, where $z \in \mathcal{M}_z$ represents learned representations. We denote $(\mathbf{x}_i, \mathbf{s}_i)$ as a paired sample, where multi-modal observations $\mathbf{x}_i$ are associated with multi-modal semantic information $\mathbf{s}_i$, and $\mathcal{D} = \{(\mathbf{x}_i, \mathbf{s}_i)\}_{i=1}^N$ as the training dataset of $N$ paired samples. When considering multiple datasets from different sources, we use bracket notation $\mathcal{D}^{[k]} = \{(\mathbf{x}_i^{[k]}, \mathbf{s}_i^{[k]})\}_{i=1}^{N_k}$ to distinguish dataset $k$, where the bracket superscript $[\cdot]$ denotes dataset index and the parenthesis superscript $(\cdot)$ denotes modality index within a dataset.

**The Role of the Latent Manifold** While high-dimensional data may concentrate near low-dimensional geometric structures under the manifold hypothesis, the learned latent manifold $\mathcal{M}_z \subset \mathcal{Z}$ serves a fundamentally different purpose than mere dimensionality reduction. The latent space is designed to be a *standardized and informationally enriched* representation that integrates structure from both observation space $\mathcal{X}$ and semantic space $\mathcal{S}$. During training, the framework learns to embed paired semantic-observation information $(\mathbf{s}, \mathbf{x})$ into $\mathcal{M}_z$, creating a representation that is simultaneously:

- *Standardized*: Different modalities and datasets map to a common geometric structure

- *Semantically enriched*: Captures high-level conceptual relationships encoded in $\mathbf{s}$

- *Observation-grounded*: Preserves discriminative information from raw measurements $\mathbf{x}$

Critically, at test time, the latent representation can be inferred from observations alone via encoder mappings $E_x^{(m)} : \mathcal{X}^{(m)} \to \mathcal{M}_z$, even when semantic information is unavailable. The semantic structure learned during training becomes implicitly embedded in the latent geometry, enabling the observation encoders to map to semantically meaningful regions of $\mathcal{M}_z$ without requiring explicit semantic input. This asymmetry, which uses both modalities during training but only observations during testing, is key to the framework's practical utility.

**The Universal Manifold Problem** Given multiple datasets with heterogeneous domains, for instance, $\mathcal{D}^{[1]}$ containing semantic attributes paired with MNIST handwritten digits and $\mathcal{D}^{[2]}$ containing semantics paired with Fashion-MNIST clothing items, the fundamental question is: *how can we learn and empirically assess that these heterogeneous semantic-observation pairs can be unified into a single universal latent manifold $\mathcal{M}_z$ that preserves the joint relational structure of each dataset?*

The challenge is threefold. **First**, we must learn encoder mappings $\{E_x^{(m)}, E_s^{(l)}\}$ that embed both semantics and observations from different domains into a shared latent manifold $\mathcal{M}_z$ while preserving their joint topological structure. **Second**, we must learn decoder mappings $\{D_\theta^{(m)}, D_\theta^{(l)}\}$ parameterized by $\theta$ that faithfully reconstruct each modality from latent codes, ensuring the latent representation captures the semantic-observation correspondence. **Third**, and most critically, we must design training objectives that achieve topological compatibility and establish verification methods to determine *when such unification appears to have succeeded based on finite-sample diagnostics*, that is, when the latent manifolds induced by different semantic-observation pairs are homeomorphic.

Our key insight is that successful unification requires these latent manifolds to be **homeomorphic**, preserving the relational structure between semantics and observations through continuous bijection. We achieve this through topology-preserving training objectives that learn homeomorphic mappings, with post-training diagnostics assessing whether the learned representations exhibit properties consistent with that goal. This homeomorphic structure enables three critical capabilities: semantic-guided sparse recovery from incomplete observations, zero-shot cross-domain classifier transfer, and compositional zero-shot learning. We formalize the conditions for homeomorphic unification in Subsection 2.3 and develop practical verification methods in Algorithm 1.

**Notation Summary** For clarity, we summarize the key notation used throughout:

- **Superscripts**: $(m)$ = observation modality index, $(l)$ = semantic modality index, $[k]$ = dataset index

- **Subscripts**: $i, j$ = sample indices, $\theta, \phi$ = neural network parameters

- **Spaces**: $\mathcal{X}^{(m)}$ = observation space for modality $m$, $\mathcal{S}^{(l)}$ = semantic space for modality $l$, $\mathcal{Z}$ = universal latent space (ambient), $\mathcal{M}_z \subset \mathcal{Z}$ = latent manifold

- **Mappings**: $E_x^{(m)} : \mathcal{X}^{(m)} \to \mathcal{M}_z$ = observation encoder, $E_s^{(l)} : \mathcal{S}^{(l)} \to \mathcal{M}_z$ = semantic encoder, $D_\theta^{(m)} : \mathcal{M}_z \to \mathcal{X}^{(m)}$ = observation decoder

## 2.2 Latent Manifold Learning Architecture

We learn latent manifold representations that explicitly capture how semantic-observation correspondences induce geometric structure in latent space. The framework maps high-dimensional observations and semantic information onto a continuous latent manifold $\mathcal{M}_z$ where geometric proximity simultaneously reflects data similarity and semantic relatedness.

### 2.2.1 Latent Manifold Representation

Our goal is to learn a latent manifold $\mathcal{M}_z \subset \mathbb{R}^d$ that serves as a common geometric space where both observation modalities from $\mathcal{X}$ and semantic modalities from $\mathcal{S}$ can be embedded while preserving their

intrinsic structure. We learn encoder mappings that project data onto this latent manifold:

$$E_x^{(m)} : \mathcal{X}^{(m)} \to \mathcal{M}_z, \quad E_s^{(l)} : \mathcal{S}^{(l)} \to \mathcal{M}_z \tag{1}$$

implemented as neural networks parameterized by $\phi$. For a given input, the latent representation is obtained deterministically as $z = E_x^{(m)}(x^{(m)}; \phi)$ or $z = E_s^{(l)}(s^{(l)}; \phi)$.

Given a latent code $z \in \mathcal{M}_z$, we define conditional likelihood models for reconstructing each modality. The reconstruction model for observations is

$$p_\theta(x^{(m)}|z) = \mathcal{N}(D_\theta^{(m)}(z), \sigma_x^2 I) \tag{2}$$

where $D_\theta^{(m)} : \mathcal{M}_z \to \mathcal{X}^{(m)}$ is a neural network decoder parameterized by $\theta$. For semantic modalities, we use appropriate likelihood functions:

$$p_\theta(s^{(l)}|z) = \begin{cases} \prod_j \text{Bernoulli}(D_\theta^{(l)}(z)_j) & \text{binary} \\ \mathcal{N}(D_\theta^{(l)}(z), \sigma_s^2 I) & \text{continuous} \\ \text{Categorical}(D_\theta^{(l)}(z)) & \text{categorical} \\ \prod_{t=1}^T p(s_t^{(l)}|z, s_{<t}^{(l)}) & \text{sequential} \end{cases} \tag{3}$$

where the sequential case handles variable-length tokenized representations (e.g., annotation sequences, text descriptions) through autoregressive modeling. All encoders and decoders are implemented as Lipschitz continuous neural networks to ensure stable manifold mappings.

### 2.2.2 Encoder and Decoder Architecture

We learn encoder mappings that project observations and semantics onto the shared latent manifold $\mathcal{M}_z$, and decoder mappings that reconstruct each modality from latent codes.

**Encoder Networks** For each observation modality $m$, we define the encoder as a neural network:

$$z = E_x^{(m)}(x^{(m)}; \phi) : \mathcal{X}^{(m)} \to \mathcal{M}_z \subset \mathbb{R}^d \tag{4}$$

parameterized by $\phi$, which deterministically maps observations to latent codes. Similarly, for semantic modalities $l$, we define encoders:

$$z = E_s^{(l)}(s^{(l)}; \phi) : \mathcal{S}^{(l)} \to \mathcal{M}_z \subset \mathbb{R}^d \tag{5}$$

**Decoder Networks** For each modality, we define decoders that reconstruct from latent codes. For observation modality $m$:

$$\hat{x}^{(m)} = D_\theta^{(m)}(z) : \mathcal{M}_z \to \mathcal{X}^{(m)} \tag{6}$$

parameterized by $\theta$. Similarly, for semantic modalities $l$:

$$\hat{s}^{(l)} = D_\theta^{(l)}(z) : \mathcal{M}_z \to \mathcal{S}^{(l)} \tag{7}$$

When a sample contains multiple modalities, we denote the available observation modalities as $\mathcal{M}_i^x \subseteq \{1, \ldots, M\}$ and available semantic modalities as $\mathcal{M}_i^s \subseteq \{1, \ldots, L\}$ for sample $i$. The framework naturally handles missing modalities: if modality $m \notin \mathcal{M}_i^x$, we simply do not compute $E_x^{(m)}(x_i^{(m)})$ for that sample.

This manifold-to-manifold perspective differs fundamentally from point-to-point mappings: the encoders $E_x^{(m)}$ and $E_s^{(l)}$ are continuous and preserve neighborhood relationships, meaning nearby points on the data manifolds map to nearby points on the latent manifold. This topological preservation property is enforced through the local consistency loss.

### 2.2.3 Multi-Objective Training

We train the framework through a composite objective that simultaneously ensures reconstruction fidelity, cross-modal alignment, and topological preservation. The loss components are designed to enforce homeomorphic properties: $\mathcal{L}_{\text{recon}}$ maintains injectivity by preventing collapse, $\mathcal{L}_{\text{local}}$ ensures continuity by preserving neighborhoods, and $\mathcal{L}_{\text{consist}}$ enables unification through geometric alignment. For a dataset $\mathcal{D} = \{(\mathbf{x}_i, \mathbf{s}_i)\}_{i=1}^N$, the complete loss is

$$\mathcal{L}_{\text{ULHM}} = \mathcal{L}_{\text{recon}}^x + \mathcal{L}_{\text{recon}}^s + \lambda_c \mathcal{L}_{\text{consist}} + \lambda_\ell \mathcal{L}_{\text{local}} + \lambda_p \mathcal{L}_{\text{percep}} \tag{8}$$

where $\lambda_c, \lambda_\ell, \lambda_p \geq 0$ are hyper-parameters. We now detail each component.

**Reconstruction Objectives** For each modality with available data, we minimize the negative log-likelihood of the reconstruction. The observation reconstruction loss is

$$\mathcal{L}_{\text{recon}}^x = \sum_{i=1}^N \sum_{m \in \mathcal{M}_i^x} -\log p_\theta(x_i^{(m)} | z_i^{(m)}) \tag{9}$$

where $z_i^{(m)} = E_x^{(m)}(x_i^{(m)}; \phi)$ is the latent encoding. Similarly, the semantic reconstruction loss is

$$\mathcal{L}_{\text{recon}}^s = \sum_{i=1}^N \sum_{l \in \mathcal{M}_i^s} -\log p_\theta(s_i^{(l)} | z_i^{(l)}) \tag{10}$$

where $z_i^{(l)} = E_s^{(l)}(s_i^{(l)}; \phi)$. For Gaussian likelihoods, $-\log p_\theta(x|z) \propto \|D_\theta^{(m)}(z) - x\|_2^2$, which corresponds to mean squared error. The reconstruction terms ensure that points on the latent manifold decode back to realistic samples on the data manifolds.

**Cross-Modal Consistency** To learn a shared latent manifold rather than separate manifolds for each modality, we enforce that different modalities from the same sample map to nearby points through a composite consistency objective combining both angular and magnitude alignment:

$$\begin{aligned}
\mathcal{L}_{\text{consist}} = & \lambda_{\cos} \sum_{i=1}^N \left[ \sum_{\substack{m,m' \in \mathcal{M}_i^x \\ m < m'}} \left( 1 - \frac{z_i^{(m)} \cdot z_i^{(m')}}{\|z_i^{(m)}\|_2 \|z_i^{(m')}\|_2} \right) + \sum_{\substack{m \in \mathcal{M}_i^x \\ l \in \mathcal{M}_i^s}} \left( 1 - \frac{z_i^{(m)} \cdot z_i^{(l)}}{\|z_i^{(m)}\|_2 \|z_i^{(l)}\|_2} \right) \right] \\
& + \lambda_{\text{eucl}} \sum_{i=1}^N \left[ \sum_{\substack{m,m' \in \mathcal{M}_i^x \\ m < m'}} \|z_i^{(m)} - z_i^{(m')}\|_2^2 + \sum_{\substack{m \in \mathcal{M}_i^x \\ l \in \mathcal{M}_i^s}} \|z_i^{(m)} - z_i^{(l)}\|_2^2 \right]
\end{aligned} \tag{11}$$

where $z_i^{(m)} = E_x^{(m)}(x_i^{(m)}; \phi)$ and $z_i^{(l)} = E_s^{(l)}(s_i^{(l)}; \phi)$, and $\lambda_{\cos}, \lambda_{\text{eucl}} \geq 0$ control the strength of angular and magnitude-based alignment, respectively. The cosine similarity term (weighted by $\lambda_{\cos}$) enables directional alignment while permitting magnitude differences, which is essential for reconstruction tasks with sparse observations where information density varies significantly between modalities. The Euclidean distance term (weighted by $\lambda_{\text{eucl}}$) provides additional geometric constraints on embedding magnitudes, ensuring tighter clustering in scenarios where magnitude differences carry semantic meaning. For homogeneous multi-modal scenarios with comparable information density (e.g., multi-view imaging, synchronized sensor fusion, or fully-paired text-image data), we set both $\lambda_{\cos}, \lambda_{\text{eucl}} > 0$ to enforce both angular and magnitude alignment. For heterogeneous scenarios with asymmetric information content (e.g., sparse vs. complete observations), we set $\lambda_{\text{eucl}} = 0$ and $\lambda_{\cos} > 0$ to prioritize directional consistency. Geometrically, this composite loss constrains embeddings from all available modalities to overlap significantly in the latent space, creating a unified latent manifold $\mathcal{M}_z$.

**Topological Preservation** Preserving the topological structure of the data manifolds is critical for learning meaningful representations. We enforce that local neighborhood structure is preserved through

$$\mathcal{L}_{\text{local}} = \sum_{m=1}^{M} \sum_{i:m\in\mathcal{M}_i^x} \sum_{j\in\mathcal{N}_\kappa(i,m)} \|z_i^{(m)} - z_j^{(m)}\|_2^2 + \sum_{l=1}^{L} \sum_{i:l\in\mathcal{M}_i^s} \sum_{j\in\mathcal{N}_\kappa(i,l)} \|z_i^{(l)} - z_j^{(l)}\|_2^2 \tag{12}$$

where $\mathcal{N}_\kappa(i,m)$ denotes the indices of the $\kappa$-nearest neighbors of sample $i$ in modality $m$'s data space, and $z_i^{(m)} = E_x^{(m)}(x_i^{(m)};\phi)$, $z_j^{(m)} = E_x^{(m)}(x_j^{(m)};\phi)$. This loss acts as a manifold regularizer: if two samples are connected by a short geodesic path on the data manifold, they should also be nearby on the latent manifold.

**Perceptual Quality (Optional)** For observation modalities in the image domain with complex textures, we optionally employ a perceptual loss to improve visual reconstruction quality:

$$\mathcal{L}_{\text{percep}} = \sum_{m\in\mathcal{I}_{\text{img}}} \sum_{i:m\in\mathcal{M}_i^x} \sum_{r=1}^{L_{\text{feat}}} w_r \|\Phi_r(\hat{x}_i^{(m)}) - \Phi_r(x_i^{(m)})\|_2^2 \tag{13}$$

where $\hat{x}_i^{(m)} = D_\theta^{(m)}(z_i^{(m)})$ with $z_i^{(m)} = E_x^{(m)}(x_i^{(m)};\phi)$, $\mathcal{I}_{\text{img}} \subseteq \{1,\dots,M\}$ denotes the subset of image modality indices, $\Phi_r(\cdot)$ extracts features from layer $r$ of a pretrained feature extraction network, $L_{\text{feat}}$ is the number of feature layers, and $w_r > 0$ are layer-specific weights. While not essential for homeomorphism preservation, this loss improves perceptual quality on high-resolution image tasks.

## 2.3 Homeomorphism Verification Methods

We establish the conditions under which latent manifolds from different datasets can be assessed for structural compatibility toward unification. For datasets $\mathcal{D}^{[1]}$ and $\mathcal{D}^{[2]}$ (e.g., MNIST and Fashion-MNIST with shared attributes), both map to latent representations, but when can they share the same latent space while preserving structure? We establish homeomorphism as a sufficient criterion for valid unification and develop practical methods to assess whether finite-sample evidence is consistent with this criterion.

### 2.3.1 Theoretical Analysis

We establish strict conditions under which disjoint datasets form a unified, topologically consistent latent manifold. Central to our framework is the requirement that the representation learning process maintains a homeomorphism between the data space and the latent space, ensuring that no topological information is lost during compression.

**Definition 1** (Bi-Lipschitz Map). *A map $f : (\mathcal{M}_x, d_X) \to (\mathcal{M}_z, d_Z)$ is $(c_1, c_2)$-bi-Lipschitz if there exist constants $0 < c_1 \le c_2 < \infty$ such that $\forall x, y \in \mathcal{M}_x$:*

$$c_1 d_X(x,y) \le d_Z(f(x), f(y)) \le c_2 d_X(x,y) \tag{14}$$

*where $d_X : \mathcal{M}_x \times \mathcal{M}_x \to \mathbb{R}_{\ge 0}$ and $d_Z : \mathcal{M}_z \times \mathcal{M}_z \to \mathbb{R}_{\ge 0}$ denote the geodesic distance metrics on the respective manifolds.*

**Theorem 1. *(Topological Unification via Latent Identifications).*** *Let $\{\mathcal{M}_x^{[k]}\}_{k=1}^K$ be pairwise disjoint topological spaces with continuous encoders $E^{[k]} : \mathcal{M}_x^{[k]} \to \mathbb{R}^d$. Define the latent images $\mathcal{M}_z^{[k]} := E^{[k]}(\mathcal{M}_x^{[k]})$ and unified latent support*

$$\mathcal{U} := \bigcup_{k=1}^K \mathcal{M}_z^{[k]} \subset \mathbb{R}^d$$

*equipped with the subspace topology induced from $\mathbb{R}^d$. Let $X := \bigsqcup_{k=1}^K \mathcal{M}_x^{[k]}$ (disjoint union with coproduct topology) and define $E : X \to \mathcal{U}$ by $E|_{\mathcal{M}_x^{[k]}} = E^{[k]}$. Define an equivalence relation $\sim$ on $X$ by $x \sim x'$ iff $E(x) = E(x')$. Then the induced map $\widetilde{E} : X/\sim \to \mathcal{U}$ is a homeomorphism provided:*

(i) (Local Homeomorphic Embeddings) *Each $E^{[k]}$ is a homeomorphism onto its image $\mathcal{M}_z^{[k]}$ (with the subspace topology inherited from $\mathbb{R}^d$), and either all $E^{[k]}$ are open maps or all $E^{[k]}$ are closed maps.*

(ii) (Open/Closed Cover) *If all $E^{[k]}$ are open, then each $\mathcal{M}_z^{[k]}$ is open in $\mathcal{U}$ and $\{\mathcal{M}_z^{[k]}\}_{k=1}^K$ covers $\mathcal{U}$. If all $E^{[k]}$ are closed, then each $\mathcal{M}_z^{[k]}$ is closed in $\mathcal{U}$ and $\{\mathcal{M}_z^{[k]}\}_{k=1}^K$ covers $\mathcal{U}$.*

**Remark 1** (Empirical Verification). *In practical applications, Conditions (i) and (ii) cannot be proved directly from finite samples, so the hierarchical protocol in Algorithm 1 is used as a diagnostic layer rather than a formal verifier. The Local Match metrics (Trust $\tau_t$ and Continuity $\tau_c$) assess whether the learned encoders exhibit empirical behavior consistent with Condition (i), such as neighborhood preservation and the absence of obvious collapse or tearing. The Global Match metrics (Betti number $\beta_0$ and Sliced Wasserstein distance $W_2$) assess empirical signatures relevant to Condition (ii): $\beta_0 \approx 1$ suggests connected latent support, while small $W_2$ suggests geometric co-location and overlap among domain manifolds. Together, these metrics provide finite-sample evidence consistent with, but do not establish, the open/closed cover and homeomorphic-embedding conditions.*

**Remark 2** (Important Caveat on Empirical Verification). *The verification protocol (Algorithm 1) provides* necessary but not sufficient *empirical evidence for homeomorphism. Theorems 1 and 2 are proved under idealized conditions (continuous maps, infinite samples, exact bi-Lipschitz bounds). The Trust score, Continuity, Sliced Wasserstein distance $W_2$, Betti number $\beta_0$, and Alignment Error are finite-sample heuristic indicators that detect violations of these conditions. Passing all thresholds indicates that the learned representations* exhibit properties consistent with *homeomorphic structure, but does not constitute a proof of homeomorphism. Throughout the remainder of this paper, "verified homeomorphism" should be understood as "empirically indicated structural compatibility consistent with homeomorphism."*

**Theorem 2** (Bi-Lipschitz Sufficiency). *Let $(\mathcal{M}_x, d_X)$ be a metric space and let $\mathcal{M}_z \subseteq \mathbb{R}^d$ be equipped with the Euclidean metric $d_Z = \|\cdot\|_2$ (i.e., $\mathcal{M}_z$ has the subspace topology inherited from $\mathbb{R}^d$). If the encoder $E : \mathcal{M}_x \to \mathcal{M}_z$ is $(c_1, c_2)$-bi-Lipschitz with $c_1 > 0$, then $E$ is an open homeomorphism onto its image. Thus, the bi-Lipschitz constraint satisfies Condition (i) of Theorem 1.*

**Remark 3** (Connection to Training Objectives). *The bi-Lipschitz property enforced by Condition (i) is optimized through the ULHM training objective (Eq. 8). The reconstruction losses $L_{recon}^x$ and $L_{recon}^s$ prevent manifold collapse by maintaining the lower bound $c_1 > 0$, while the local consistency loss $L_{local}$ prevents excessive stretching by bounding the upper constant $c_2$. The Trust and Continuity metrics in Algorithm 1 provide empirical diagnostics relevant to whether these bi-Lipschitz conditions appear to hold for the learned encoders across domains.*

### 2.3.2 Empirical Verification Protocol

Direct computation of the bi-Lipschitz constants $c_1, c_2$ and verification of the open cover condition (Theorem 1, Condition ii) is intractable for real datasets. We bridge theory and practice through a hierarchical verification framework that employs computable metrics to detect violations of the theoretical conditions across structural, geometric, and homeomorphic dimensions.

**Verification Strategy.** While Theorems 1 and 2 provide guarantees under ideal conditions (continuous maps, infinite samples), we assess their approximate empirical satisfaction through complementary metrics organized in a hierarchical dependency structure. Each metric targets a specific failure mode:

1. **Structural Fragmentation:** $\beta_0(\mathcal{Z}_{\text{total}}) > 1$ with high pairwise $W_2$ indicates the latent space contains disjoint components, violating the unified manifold assumption. A slightly elevated $\beta_0$ under low $W_2$ typically reflects finite-sample artifacts rather than fundamental unification failure.

2. **Geometric Misalignment:** $\beta_0 \approx 1$ but large $W_2(\hat{\mathbb{P}}_z^{[i]}, \hat{\mathbb{P}}_z^{[j]})$ indicates domains occupy distant regions with insufficient density overlap, violating the open cover requirement (Theorem 1, Condition ii).

3. **Manifold Collapse:** Low Trust Score $\tau_t$ indicates distinct semantic neighborhoods merge in latent space, signaling violation of the bi-Lipschitz lower bound ($c_1 \to 0$) and thus failure of injectivity (Theorem 2).

4. **Structural Incoherence:** Low Continuity $\tau_c$ indicates the encoder tears or folds the manifold—neighbors in input space become distant in latent space—suggesting violation of smooth homeomorphism.

5. **Inconsistent Mapping:** High Alignment Error $\tau_a$ indicates different modalities representing the same entity map to distant latent points, violating cross-modal semantic consistency.

These failure modes form a hierarchical dependency: global unification (items 1–2) is necessary for local preservation (items 3–4) to be meaningful, and both are necessary for semantic coherence (item 5).

**Three-Level Verification Framework** We use a hierarchical diagnostic protocol organized into three levels—*global match*, *local match*, and *semantic match*—to assess whether the learned latent space exhibits finite-sample properties consistent with a bi-Lipschitz homeomorphic regime. These levels probe whether the latent space $\mathcal{Z}$ appears to integrate both the observation space $\mathcal{X}$ and the semantic space $\mathcal{S}$ without obvious fragmentation, tearing, or semantic mismatch.

**1. Global Match.** These metrics assess whether disparate domains appear to have unified into a single, coherent geometric framework:

- **Betti Number** ($\beta_0$): Measures the number of connected components in the latent space via persistent homology. Structural integration is empirically indicated when $\beta_0 = 1$, indicating that disparate domains have fused into a single connected manifold rather than remaining as fragmented, disjoint parts. Values of $\beta_0 > 1$ indicate structural fragmentation, while $\beta_0 = 1$ is consistent with topological unification.

- **Sliced Wasserstein Distance** ($W_2$): Measures the geometric distance between probability distributions in latent space using optimal transport theory (Courty et al., 2017). For empirical distributions $\hat{\mathbb{P}}_z^{[i]}$ and $\hat{\mathbb{P}}_z^{[j]}$, small values indicate that latent representations from different domains occupy the same region with substantial density overlap, while large values suggest geometric misalignment. The metric quantifies support coincidence: whether the manifolds induced by different semantic-observation pairs are geometrically coincident.

**2. Local Match.** These metrics assess whether the local topology of the observation space appears to be faithfully and smoothly preserved in the latent space:

- **Trust Score** (Jiang et al., 2018): Measures the agreement between the classifier and a modified nearest-neighbor classifier, quantifying whether neighborhoods in the input space are preserved in the embedding space. High trust scores are consistent with the encoder maintaining local injectivity ($c_1 > 0$), suggesting that distinct semantic neighborhoods do not collapse into the same latent region. The metric detects manifold collapse where the bi-Lipschitz lower bound fails.

- **Continuity** (Venna & Kaski, 2001): Measures whether points that are neighbors in the latent space were also neighbors in the original space, quantifying the smoothness of the learned manifold. High continuity scores indicate that the embedding preserves local structure without introducing spurious proximities or shattering the neighborhood relationships. This metric detects structural incoherence where local variance is high relative to mean distance.

**3. Semantic Match.** This metric assesses the synchronization between semantic and observation modalities:

- **Alignment Error**: Measures the mean distance between embeddings of paired semantic-observation samples from the same underlying entity. For paired data $(x_i^{(m)}, s_i^{(l)})$ representing the same reality, this metric quantifies whether they map to nearby coordinates in $\mathcal{Z}$:

$$\tau_a = \frac{1}{N_{\text{paired}}} \sum_{i=1}^{N_{\text{paired}}} \|E_x^{(m)}(x_i^{(m)}) - E_s^{(l)}(s_i^{(l)})\|_2 \tag{15}$$

---

**Algorithm 1** Hierarchical Homeomorphism Verification

---

**Require:** Integrated latent pool $\mathcal{Z}_{\text{total}} = \bigcup_{k=1}^{K} \mathcal{Z}^{[k]}$, thresholds $\tau_w, \tau_t, \tau_a, \tau_c$
**Require:** Flags: HASPAIREDMODALITIES, REQUIRESCLUSTERING
  1: **Step 1: Global Match Check**
  2: **if** $\beta_0(\mathcal{Z}_{\text{total}}) \gg 1$ and $W_2(\mathcal{Z}^{[i]}, \mathcal{Z}^{[j]}) > \tau_w$ **then**
  3:     **return** FAIL: Structural Fragmentation
  4: **end if**
  5: **for** $i = 1$ to $K, j = i + 1$ to $K$ **do**
  6:     **if** $W_2(\mathcal{Z}^{[i]}, \mathcal{Z}^{[j]}) > \tau_w$ **then**
  7:         **return** FAIL: Geometric Misalignment
  8:     **end if**
  9: **end for**
 10: **Step 2: Local Match Check**
 11: **for** $k = 1$ to $K$ **do**
 12:     **if** $\text{Trust}_\kappa(\mathcal{Z}^{[k]}) < \tau_t$ **then**
 13:         **return** FAIL: Local Manifold Collapse
 14:     **end if**
 15:     **if** REQUIRESCLUSTERING and $\text{Cont}_\kappa(\mathcal{Z}^{[k]}) < \tau_c$ **then**
 16:         **return** FAIL: Structural Incoherence
 17:     **end if**
 18: **end for**
 19: **Step 3: Semantic Match Check**
 20: **if** HASPAIREDMODALITIES **then**
 21:     **if** AlignmentError($\mathcal{Z}_{\text{total}}$) $> \tau_a$ **then**
 22:         **return** FAIL: Inconsistent Cross-Modal Mapping
 23:     **end if**
 24: **end if**
 25: **return** PASS: Empirical diagnostics consistent with $\bigcup_{k=1}^{K} \mathcal{Z}^{[k]}$ forming a unified manifold $\mathcal{M}_{\text{univ}}$.

---

Low alignment error indicates consistent cross-modal mapping where the encoder respects the semantic-observation correspondence dictated by the joint data structure. The acceptable threshold depends on whether modalities are homogeneous (same domain) or heterogeneous (cross-domain scenarios).

**Remark 4** (Threshold Selection and Numerical Unity). *Thresholds $\tau_w, \tau_t, \tau_a, \tau_c$ are determined empirically through validation on datasets with known compatible and incompatible domain pairs. In practical verification, $\beta_0, W_2$, and $\tau_c$ must be evaluated in tandem. A manifold may numerically present $\beta_0 > 1$ due to sampling sparsity, yet exhibit $W_2 \approx 0$ and high $\tau_c$, indicating that the domains are geometrically coincident and locally smooth. We define successful unification as the state where domains share a common support that is either path-connected ($\beta_0 = 1$) or geometrically indistinguishable ($W_2$ approaching zero) and locally consistent (high continuity).*

### 2.3.3 Theoretical Connection Between ULHM Objectives and Verification Metrics

The ULHM training objectives (Eq. 8) are deliberately designed to enforce the bi-Lipschitz homeomorphism conditions that our empirical metrics are intended to probe. We establish the logical chain connecting each training component to its corresponding verification criterion.

**Preventing Manifold Collapse (Trust Score $\tau_t$).** The reconstruction losses $\mathcal{L}_{\text{recon}}^x$ and $\mathcal{L}_{\text{recon}}^s$ enforce that the encoder must preserve sufficient information to faithfully reconstruct the original inputs. If the encoder were to collapse distinct semantic classes to the same latent region (violating the bi-Lipschitz lower bound $c_1 > 0$), reconstruction quality would degrade as the decoder cannot disambiguate merged representations. This information-preserving pressure maintains local injectivity, which manifests empirically as

high Trust scores, i.e., semantic neighbors in data space remain semantic neighbors in latent space without spurious mixing.

**Preserving Local Topology (Continuity $\tau_c$).** The local consistency loss $\mathcal{L}_{\text{local}}$ explicitly minimizes distances between $\kappa$-nearest neighbors in data space. This directly implements the continuity requirement of homeomorphism: points close in $\mathcal{M}_x$ must map to points close in $\mathcal{M}_z$. By construction, this objective optimizes precisely for the Continuity metric, ensuring the encoder does not tear the manifold by mapping nearby data points to distant latent regions.

**Achieving Geometric Unification (Wasserstein Distance $W_2$ and Alignment Error $\tau_a$).** The cross-modal consistency loss $\mathcal{L}_{\text{consist}}$ operates at two complementary levels. The cosine similarity component (weighted by $\lambda_{\cos}$) enforces directional alignment, ensuring different modalities representing the same semantic content point toward the same region in latent space, which directly reduces the distributional distance between domains measured by $W_2$. The Euclidean distance component (weighted by $\lambda_{\text{eucl}}$) further constrains the magnitude alignment, minimizing $\tau_a$ by ensuring paired semantic-observation samples $(s_i, x_i)$ map to nearby coordinates. Together, these components enforce the global connectivity condition (Theorem 1) by ensuring non-trivial overlap $\nu(\mathcal{M}_z^{[i]} \cap \mathcal{M}_z^{[j]}) > 0$ between domain manifolds.

**Ensuring Topological Unity (Betti Number $\beta_0$).** The combination of reconstruction, consistency, and local preservation losses creates a unified manifold structure. Reconstruction losses prevent the latent space from fragmenting into disconnected task-specific subspaces by requiring all modalities to decode successfully from shared representations. Cross-modal consistency explicitly bridges domains by constraining them to occupy overlapping regions. These combined pressures ensure the latent space forms a single connected component ($\beta_0 = 1$) rather than disjoint manifolds.

In summary, the ULHM objective function is not an ad hoc combination of losses, but a principled implementation of the bi-Lipschitz homeomorphism criterion: $\mathcal{L}_{\text{recon}}$ enforces the lower bound $c_1 > 0$ (preventing collapse), $\mathcal{L}_{\text{local}}$ enforces the upper bound $c_2 < \infty$ (preventing tearing), and $\mathcal{L}_{\text{consist}}$ enforces global connectivity (enabling unification). Each verification metric subsequently assesses whether these theoretical conditions appear to hold empirically on finite samples.

# 3 Applications: Sparse Recovery, Transfer, and Zero-Shot Learning

Having established the ULHM framework for learning homeomorphic latent manifolds and assessing their structure empirically, we now demonstrate how this dual approach enables three key applications. Each application exploits a distinct property: sparse recovery leverages learned semantic structure for reconstruction from incomplete observations, cross-domain transfer exploits empirically assessed manifold overlap for zero-shot classifier transfer, and zero-shot learning performs compositional reasoning via topology-preserved manifold interpolation. Critically, all three applications depend on homeomorphic structure being both enforced during training and assessed diagnostically before deployment.

## 3.1 Sparse Recovery with Semantic Priors

We demonstrate how the ULHM framework enables sparse recovery: recovering complete observations from highly incomplete measurements using semantic priors learned during training, even when semantic annotations are unavailable at test time.

**Problem Setup:** During training, we observe complete semantic-observation pairs $(\mathbf{s}_i, \mathbf{x}_i) \in \mathcal{D}$ with multiple modalities. For facial image reconstruction on CelebA, the observation modalities consist of: (1) complete ground truth images $x^{(1)} \in \mathbb{R}^{H \times W \times C}$, and (2) sparse masked observations $x^{(2)} = x^{(1)} \odot M$, where $\odot$ denotes element-wise (Hadamard) multiplication and binary mask $M \in \{0, 1\}^{H \times W}$ retains only fraction $\rho$ of pixels. The semantic modalities comprise binary facial attributes $s^{(1)} \in \{0, 1\}^K$ encoding properties such as hair color, presence of glasses, and gender. At deployment, we observe only sparse measurements $\tilde{x}^{(2)}$ without semantic annotations, and must reconstruct the complete image $x^{(1)}$.

**Training with the ULHM Framework:** We train the sparse recovery task using the complete ULHM objective (Eq. 8). The key mechanism enabling sparse recovery is the cross-modal consistency loss $\mathcal{L}_{\text{consist}}$, which enforces that sparse observations $x^{(2)}$, complete images $x^{(1)}$, and semantic attributes $s^{(1)}$ from the same sample map to nearby regions in $\mathcal{M}_z$. This geometric alignment enables the sparse encoder to implicitly access semantic structure learned from complete observations, even when semantic annotations are unavailable at test time. To enable semantic guidance during training while maintaining robustness at test time, we employ attribute dropout: with probability $p_{\text{drop}}$, we zero the attribute embeddings used for conditioning. This forces the encoder to alternate between semantic-guided training (aligning sparse observations with explicit attributes) and observation-only training (extracting semantic structure from visual content alone), ensuring the encoder can leverage implicit semantic geometry at deployment when attributes are unavailable.

**Test-Time Recovery via Amortized Inference:** At deployment, given only sparse observations $\tilde{x}^{(2)}$ and mask $M$ without semantic annotations, we reconstruct the complete image through direct amortized inference:

$$x^* = D_\theta^{(1)}(E_x^{(2)}([\tilde{x}^{(2)}, M])) \tag{16}$$

where the sparse encoder $E_x^{(2)}$ maps incomplete observations to the latent manifold $\mathcal{M}_z$, and the decoder $D_\theta^{(1)}$ reconstructs the complete image modality. If Algorithm 1 indicates structural compatibility consistent with homeomorphism, semantic structure learned during training is embedded as geometric constraints in $\mathcal{M}_z$, enabling recovery from incomplete observations without explicit semantic input.

**Dependence on Learned and Verified Homeomorphism:** The effectiveness of this approach depends critically on achieving and empirically assessing structural compatibility consistent with homeomorphic structure. During training, the topology-preserving objectives (Eq. 8) learn mappings designed to satisfy the bi-Lipschitz conditions. Post-training diagnostics then assess whether obvious violations remain: if $\text{Trust}_\kappa$ or $\text{Cont}_\kappa$ falls below threshold, the learned manifold has failed to preserve data topology, and sparse observations may map to regions where semantic structure is unreliable, causing semantically incoherent reconstructions. Conversely, when verification passes, the bi-Lipschitz property (Theorem 2) implies, under its idealized assumptions, that distinct semantic states occupy geometrically separated regions in $\mathcal{M}_z$, enabling reliable reconstruction from sparse measurements.

## 3.2 Cross-Domain Integration and Transfer

A key advantage of the universal latent manifold is enabling classifier transfer across heterogeneous datasets without retraining. Consider two datasets: $\mathcal{D}^{[1]} = \{(\mathbf{s}_i^{[1]}, \mathbf{x}_i^{[1]})\}_{i=1}^{N_1}$ and $\mathcal{D}^{[2]} = \{(\mathbf{s}_j^{[2]}, \mathbf{x}_j^{[2]})\}_{j=1}^{N_2}$ from different domains but sharing similar semantic structure (e.g., MNIST handwritten digits and Fashion-MNIST clothing items, both with shared class labels). The central question is: can a classifier trained solely on the first dataset directly classify samples from the second dataset without any adaptation or retraining?

Our framework addresses this through learning homeomorphic structure and assessing it diagnostically after training. During training, the ULHM objectives (Eq. 8) enforce topology preservation and manifold alignment for both datasets. Post-training, we apply Algorithm 1 to assess three empirical conditions: (1) individual homeomorphism through $\text{Trust}_\kappa$ and $\text{Cont}_\kappa$ for both datasets to assess whether each preserves its topology in latent space, (2) manifold overlap through $W_2(\hat{P}_z^{[1]}, \hat{P}_z^{[2]})$ to assess whether the latent manifolds induced by different datasets occupy the same region of $\mathcal{M}_z$, and (3) alignment consistency to assess semantic-observation correspondence. When all three conditions pass, cross-domain transfer becomes theoretically grounded and empirically reliable.

**Three-Stage Training Protocol** Unlike methods that jointly train all components, our framework employs a three-stage protocol that ensures fair transfer evaluation by strictly separating domain-specific learning from universal alignment. For cross-domain transfer, we treat each dataset as a unified entity with domain-specific encoder and decoder networks.

**Stage 1: Domain-Specific Encoder Pre-training.** We independently pre-train domain-specific encoders $E^{[k]} : \mathcal{X}^{[k]} \to \mathcal{Z}^{[k]}$ and decoders $D^{[k]} : \mathcal{Z}^{[k]} \to \mathcal{X}^{[k]}$ for each dataset using the complete ULHM objective (Eq. 8) applied within each domain independently:

$$\mathcal{L}_{\text{pretrain}}^{[k]} = \mathcal{L}_{\text{ULHM}}\big|_{\mathcal{D}^{[k]}} \tag{17}$$

where $\tilde{x}_i^{[k]} = x_i^{[k]} \odot M_i$ represents sparse observations with mask $M_i$ retaining fraction $\rho$ of measurements, and $x_i^{[k]}$ is the complete ground truth. This stage learns semantically-enriched domain-specific representations without cross-domain interaction, establishing independent baseline representations for each dataset.

**Stage 2: Domain-Invariant Projection with Alignment.** After freezing the pre-trained encoders, we train projection networks $\Pi^{[k]} : \mathcal{Z}^{[k]} \to \mathcal{M}_z$ that map domain-specific latent codes to the universal manifold:

$$z_{\text{univ},i}^{[k]} = \Pi^{[k]}(E^{[k]}([\tilde{x}_i^{[k]}, M_i])) \tag{18}$$

The projection networks are trained jointly using a composite objective that combines reconstruction, contrastive alignment (Chen et al., 2020b), and centroid alignment (Wen et al., 2016):

$$\mathcal{L}_{\text{align}} = \sum_{k=1}^{K} \sum_{i=1}^{N_k} \|D^{[k]}(z_{\text{univ},i}^{[k]}) - x_i^{[k]}\|_2^2 + \lambda_{\text{cont}}\mathcal{L}_{\text{contrastive}} + \lambda_{\text{cent}}\mathcal{L}_{\text{centroid}} \tag{19}$$

where the contrastive loss encourages samples with matching labels from different domains to cluster together:

$$\mathcal{L}_{\text{contrastive}} = -\sum_{i,j} \mathbb{1}[y_i^{[k]} = y_j^{[k']}] \log \frac{\exp(\text{sim}(z_{\text{univ},i}^{[k]}, z_{\text{univ},j}^{[k']})/\tau)}{\sum_n \exp(\text{sim}(z_{\text{univ},i}^{[k]}, z_{\text{univ},n})/\tau)} \tag{20}$$

where $\mathbb{1}[\cdot]$ is the indicator function, $\text{sim}(\cdot,\cdot)$ denotes cosine similarity, and $\tau$ is the temperature parameter. The centroid loss enforces that class centroids $\mu_c^{[k]} = \frac{1}{|\mathcal{I}_c^{[k]}|} \sum_{i \in \mathcal{I}_c^{[k]}} z_{\text{univ},i}^{[k]}$, where $\mathcal{I}_c^{[k]} = \{i : y_i^{[k]} = c\}$ is the set of sample indices with class label $c$ in dataset $k$, coincide across domains:

$$\mathcal{L}_{\text{centroid}} = \sum_{c=0}^{C-1} \sum_{k,k'=1,k<k'}^{K} \|\mu_c^{[k]} - \mu_c^{[k']}\|_2^2 \tag{21}$$

with $\mathcal{I}_c^{[k]} = \{i : y_i^{[k]} = c\}$ denoting sample indices with class label $c$ in domain $k$, $\text{sim}(\cdot,\cdot)$ as cosine similarity, and $\tau$ as temperature. Critically, no classifier is trained during this stage, i.e., the alignment losses operate solely on the learned universal manifold structure, ensuring that semantic correspondence emerges from geometric constraints rather than supervised classification pressure.

**Stage 3: Single-Domain Classifier Training.** After the universal manifold is established, we train a classifier $C_\theta : \mathcal{M}_z \to \mathcal{Y}$ exclusively on one domain while keeping all encoders and projections frozen:

$$\mathcal{L}_{\text{class}}^{[1]} = \sum_{i=1}^{N_1} \mathcal{L}_{\text{CE}}(C_\theta(z_{\text{univ},i}^{[1]}), y_i^{[1]}) \tag{22}$$

where $z_{\text{univ},i}^{[1]} = \Pi^{[1]}(E^{[1]}([\tilde{x}_i^{[1]}, M_i]))$ and $\mathcal{L}_{\text{CE}}$ is the cross-entropy loss. This single-domain training protocol is essential for fair transfer evaluation: the classifier never observes samples from the target domain, ensuring that any successful transfer provides empirical evidence consistent with the learned homeomorphic structure rather than merely reflecting memorized cross-domain patterns.

**Zero-Shot Cross-Domain Transfer:** At test time, the trained classifier directly operates on the target domain (e.g., Fashion-MNIST) without any adaptation. For a test sample $\tilde{x}^{[2]}$ from the target domain:

$$\hat{y} = \arg\max_c C_\theta(z_{\text{univ}}^{[2]})_c, \quad z_{\text{univ}}^{[2]} = \Pi^{[2]}(E^{[2]}([\tilde{x}^{[2]}, M])) \tag{23}$$

When the diagnostic metrics indicate structural compatibility, transfer succeeds because: (1) the projection $\Pi^{[2]}$ maps target domain samples to the same universal manifold $\mathcal{M}_z$ where the classifier was trained, and (2) samples with the same semantic label from different domains occupy nearby regions due to empirically indicated alignment, enabling the decision boundaries learned on the source domain to generalize to the target domain.

This zero-shot capability eliminates the need for: (1) labeled data from the target domain, (2) fine-tuning or domain adaptation procedures, and (3) separate classifiers per domain. A single classifier trained on one domain transfers across datasets that pass the structural-compatibility diagnostics.

**Dependence on Homeomorphic Structure:** The zero-shot transfer capability depends critically on achieving and empirically assessing structural compatibility consistent with homeomorphic structure. During training, the alignment objectives (Eq. 8) learn mappings designed to achieve topological compatibility across domains. When verification (Algorithm 1) indicates structural compatibility, the classifier successfully transfers across domains with minimal accuracy degradation. If verification fails, indicating that obvious structural incompatibilities remain, cross-domain transfer produces unreliable predictions. Conversely, when verification fails—for instance, when MNIST is paired with random Gaussian noise—cross-domain transfer produces near-chance accuracy, demonstrating that the verification protocol correctly predicts failure before deployment (see failure case analysis in Section 4). Our experiments in Section 4 demonstrate that when topology-preserving structure is learned and the diagnostics indicate compatibility, a classifier trained exclusively on MNIST achieves substantial accuracy on Fashion-MNIST without ever observing Fashion-MNIST samples during training, and vice versa, supporting the claim that the diagnostic protocol predicts when zero-shot transfer is likely to succeed.

### 3.3 Transductive Zero-Shot Learning via Manifold Learning

Having sparse recovery (Section 3.1) and cross-domain transfer (Section 3.2), we now show how the ULHM framework enables zero-shot classification through topological preservation. Unlike the previous applications that rely on cross-modal alignment or domain-invariant projections, zero-shot learning exploits the local consistency loss $\mathcal{L}_{\text{local}}$ to ensure unseen classes naturally cluster in geometrically separated regions of $\mathcal{M}_z$.

**Problem Setup:** During training, we observe observation-label pairs $(\mathbf{x}_i, \mathbf{s}_i) \in \mathcal{D}$ where the semantic modality consists of class labels $s^{(l)} \in \{0, 1, \ldots, C-1\}$. We deliberately withhold certain classes from classifier training by partitioning the label space into seen classes $\mathcal{Y}^s = \{0, \ldots, C_s - 1\}$ and unseen classes $\mathcal{Y}^u = \{C_s, \ldots, C-1\}$. Critically, we train encoders $E_x^{(m)}$ and decoders $D_\theta^{(m)}$ on *all* classes using the complete ULHM objective (Eq. 8), while the auxiliary classifier $C_\theta : \mathcal{M}_z \to \mathbb{R}^{C_s}$ sees only the seen classes subset. This constitutes a **transductive zero-shot setting**: the encoder observes the visual content of all classes (including unseen ones) during training through $\mathcal{L}_{\text{recon}}^x$ and $\mathcal{L}_{\text{local}}$, but class labels for unseen classes are strictly withheld from the classifier. This is weaker than fully inductive zero-shot learning, where unseen-class images are entirely absent from training. At test time, we face this transductive zero-shot scenario: can the framework correctly predict classes the classifier has never been trained on?

**Training with the ULHM Framework:** We apply the ULHM objective (Eq. 8) with class labels as the only semantic modality. For unseen classes, we use only the observation reconstruction term $\mathcal{L}_{\text{recon}}^x$ and topological preservation loss $\mathcal{L}_{\text{local}}$, omitting semantic reconstruction $\mathcal{L}_{\text{recon}}^s$ and cross-modal consistency $\mathcal{L}_{\text{consist}}$ since we deliberately withhold their labels:

$$\mathcal{L}_{\text{zero-shot}} = \underbrace{\sum_{i: s_i^{(l)} \in \mathcal{Y}^s} \mathcal{L}_{\text{ULHM}}\big|_i}_{\text{seen classes: full ULHM}} + \underbrace{\sum_{i: s_i^{(l)} \in \mathcal{Y}^u} (\mathcal{L}_{\text{recon}}^x + \lambda_\ell \mathcal{L}_{\text{local}} + \lambda_p \mathcal{L}_{\text{percep}})\big|_i}_{\text{unseen classes: observation-only}} \tag{24}$$

Additionally, we train an auxiliary classifier only on seen classes:

$$\mathcal{L}_{\text{classifier}} = \sum_{i: s_i^{(l)} \in \mathcal{Y}^s} \mathcal{L}_{\text{CE}}(C_\theta(E_x^{(m)}(x_i^{(m)})), s_i^{(l)}) \tag{25}$$

The key mechanism enabling zero-shot transfer is the topological preservation loss $\mathcal{L}_{\text{local}}$, which enforces that visually similar samples cluster together regardless of whether their labels are used for classifier training. For unseen classes, this clustering emerges purely from visual similarity preserved through reconstruction and neighborhood structure, ensuring they occupy well-defined, geometrically separated regions in $\mathcal{M}_z$ even without classifier supervision.

**Nearest-Centroid Classification in Latent Space:** For a test sample $x_{\text{test}}^{(m)}$ from an unseen class, we perform nearest-centroid classification directly in the learned latent manifold. For each class $c \in \{0, \ldots, C-1\}$ (including unseen classes), we compute the centroid using all available training samples:

$$\mu_c = \frac{1}{|\mathcal{I}_c|} \sum_{i \in \mathcal{I}_c} E_x^{(m)}(x_i^{(m)}; \phi) \tag{26}$$

where $\mathcal{I}_c = \{i : s_i^{(l)} = c\}$ is the set of training sample indices with class label $c$. Crucially, we can compute centroids for unseen classes because the encoders processed these samples during training through the ULHM reconstruction objectives, even though the classifier never saw their labels.

Classification proceeds by finding the nearest centroid in latent space:

$$\hat{c} = \arg \min_{c \in \{0, \ldots, C-1\}} \|E_x^{(m)}(x_{\text{test}}^{(m)}; \phi) - \mu_c\|_2 \tag{27}$$

For generalized zero-shot learning (GZSL), the search space includes both seen and unseen classes. For conventional zero-shot learning (ZSL), the search space is restricted to $\mathcal{Y}^u$ only.

**How Homeomorphism Enables Zero-Shot Transfer:** The zero-shot capability fundamentally relies on homeomorphic structure being learned through topology-preserving objectives and assessed post-training through empirical diagnostics. By Theorem 2, when the encoder $E_x^{(m)}$ satisfies the bi-Lipschitz condition with $c_1 > 0$, it forms a homeomorphism onto its image, and the theoretical framework (under idealized conditions) implies three critical properties, which our empirical diagnostics are designed to detect: (1) **Injectivity** ($c_1 > 0$): distinct visual samples map to distinct latent codes, preventing manifold collapse and ensuring unseen classes occupy well-defined, separable regions in $\mathcal{M}_z$ even without classifier supervision. (2) **Local continuity** (bounded $c_2$): the local consistency loss $\mathcal{L}_{\text{local}}$ enforces that samples from the same class cluster together, with unseen classes clustering purely from geometric similarity through preserved neighborhood structure. (3) **Global structure preservation**: reconstruction objectives $\mathcal{L}_{\text{recon}}^x$ and $\mathcal{L}_{\text{recon}}^s$ ensure the latent manifold captures discriminative features, supporting the expectation that unseen class centroids $\{\mu_c\}_{c \in \mathcal{Y}^u}$ occupy regions geometrically separated from seen class centroids.

Nearest-centroid classification succeeds because the homeomorphic encoder maps the entire data manifold $\mathcal{M}_\mathcal{X}$ (including unseen classes) to $\mathcal{M}_z$ while preserving topological invariants. Although the classifier $C_\theta$ learns decision boundaries only on seen classes, homeomorphic structure, when the empirical diagnostics indicate structural compatibility, supports the expectation that geometric relationships extend globally, enabling unseen class samples to naturally cluster at their correct centroids through preserved topology. If topology preservation fails during training (detected by Trust scores dropping below threshold during verification) zero-shot classification produces arbitrary predictions, as unseen class samples no longer cluster coherently in latent space.

## 4 Experimental Validation

We validate the ULHM framework across three experimental domains corresponding to the applications in Section III: (1) sparse recovery with semantic priors using MNIST digits and CelebA facial images with binary attributes, (2) cross-domain integration and transfer using MNIST-Fashion-MNIST pairs, and (3) zero-shot classification on MNIST, Fashion-MNIST, and CIFAR-10. For each application, homeomorphism preservation is enforced during training through the complete ULHM objective (Eq. 8), which is explicitly designed to satisfy the bi-Lipschitz conditions of Theorems 1 and 2. Post-training verification via Algorithm 1

provides finite-sample diagnostics indicating whether the learned representations exhibit properties consistent with homeomorphic structure, providing empirical evidence that the structural conditions for valid unification and transfer are approximately satisfied before deployment. In addition to these three domains, we include failure case analyses demonstrating the verification protocol on incompatible domain pairs (Section 4). All baselines are evaluated under matched supervision assumptions.

## 4.1 Implementation Details

**Latent Dimensionality.** We set latent dimension $d = 512$ for MNIST/Fashion-MNIST to support universal classifier learning and cross-domain transfer, $d = 1024$ for CelebA to accommodate high-dimensional attribute embeddings and reconstruction quality, and $d = 512$ for CIFAR-10 to balance capacity with computational efficiency.

**Architecture.** We implement encoders $E_x^{(m)}$ and decoders $D_\theta^{(m)}$ using convolutional neural networks for image modalities. For MNIST and Fashion-MNIST (28×28 grayscale), encoders consist of 4 convolutional layers (32, 64, 128, 256 channels) with 3×3 kernels, stride 2, and LeakyReLU activations, followed by fully connected layers projecting to $z \in \mathbb{R}^d$. Decoders mirror this architecture with transposed convolutions. For CelebA (64×64 RGB), we use ResNet-based encoders with Adaptive Instance Normalization (AdaIN) blocks for semantic conditioning, incorporating attribute embedding networks that project binary attributes into continuous embedding space before modulating feature statistics. For CIFAR-10 (32×32 RGB), we use Wide Residual Networks (Zagoruyko & Komodakis, 2016) as the backbone for zero-shot learning, adapted with 4 convolutional layers to the spatial resolution. The auxiliary classifier $C_\theta$ for class labels uses either a 3-layer MLP with hidden dimensions [512, 256, num_classes] for MNIST/Fashion-MNIST, or a spatial classifier with convolutional feature extraction followed by adaptive pooling for cross-domain transfer tasks.

**Datasets.** We evaluate our framework on six benchmark datasets. MNIST contains 60,000 training and 10,000 test grayscale images ($28 \times 28$) of handwritten digits across 10 classes. Fashion-MNIST follows the same structure with clothing items. USPS contains 7,291 training and 2,007 test grayscale images of handwritten digits (postal service scans) across 10 classes; images are resized to $28 \times 28$ for compatibility with MNIST. SVHN contains 73,257 training and 26,032 test RGB images ($32 \times 32$) of street-view house numbers across 10 digit classes; images are resized to $28 \times 28$ and converted to grayscale for cross-domain transfer experiments. CelebA contains 162,770 training and 19,962 test facial images ($64 \times 64$ after center cropping and resizing) with 40 binary attributes including hair color, accessories, and facial features. CIFAR-10 contains 50,000 training and 10,000 test RGB images ($32 \times 32$) across 10 natural object categories. All images are normalized to $[0, 1]$ range.

**Evaluation Metrics.** We report Mean Squared Error (MSE) for reconstruction quality and accuracy for classification tasks. For homeomorphism verification, we compute Trust Score (Jiang et al., 2018), Continuity (Venna & Kaski, 2001), Sliced Wasserstein-2 distance, Betti number $\beta_0$, and Alignment Error as defined in Section 2.3, with neighborhood size $\kappa = 5$.

We set verification thresholds as follows: Trust $\tau_t \geq 0.80$ following the original trust score formulation (Jiang et al., 2018), which demonstrates that scores above 0.80 reliably indicate preserved neighborhood structure in manifold learning. Continuity $\tau_c \geq 0.70$ aligns with empirical observations in manifold learning literature (Venna & Kaski, 2001) where values above 0.70 indicate acceptable local structure preservation. For Betti number, structural integration requires $\beta_0 = 1$, indicating a single connected component. Sliced Wasserstein distance $\tau_w \leq 0.30$ is set based on domain adaptation studies (Courty et al., 2017), where inter-domain distances below 0.3 indicate successful alignment relative to typical intra-domain variation. Alignment Error $\tau_a \leq 0.30$ accommodates the natural variation in cross-domain heterogeneous scenarios while ensuring meaningful semantic-observation correspondence, as opposed to stricter thresholds ($\tau_a \leq 0.15$) suitable for single-domain paired modalities.

**Baseline Comparability.** Several baselines originate from different literatures and were originally designed for different problem settings. To ensure fair evaluation, we reimplement all baselines under our exact experimental setup, matching data splits, supervision assumptions, and (for zero-shot experiments) transductive access to unseen-class images.

**Baselines.** We evaluate our framework against a comprehensive suite of classical benchmarks and state-of-the-art methods.

- For *sparse recovery*, we compare against the following leading methods: Attention U-Net (Oktay et al., 2018), Dense U-Net (Li et al., 2018), DANet (Fu et al., 2019), Residual Attention (Wang et al., 2017), CBAM U-Net (Woo et al., 2018), ResUNet-PSP (Diakogiannis et al., 2020), and ResUNet (Zhang et al., 2018).

- For *cross-domain transfer learning*, we compare against several representative and state-of-the-art domain adaptation and transfer learning methods, including Partial Convolution (PConv) (Liu et al., 2018), Maximum Classifier Discrepancy (MCD) (Saito et al., 2018), Conditional Domain Self-supervised Learning (CDSL) (Kim et al., 2020), Minimum Class Confusion (MCC) (Jin et al., 2020), Conditional Domain Adversarial Networks (CDAN) (Long et al., 2018), Deep Correlation Alignment (CORAL) (Sun & Saenko, 2016), and Prototypical Networks (ProtoNet) (Snell et al., 2017).

- For *zero-shot learning*, we compare against widely used state-of-the-art baselines including Contrastive Learning (SimCLR) (Chen et al., 2020b), Prototypical Networks (Snell et al., 2017), Triplet Networks (Hoffer & Ailon, 2015), Variational Autoencoders (VAE) (Kingma & Welling, 2014), Maximum Classifier Discrepancy (MCD) (Saito et al., 2018), Deep Correlation Alignment (CORAL) (Sun & Saenko, 2016), Mixup Augmentation (Zhang et al., 2017), and Self-supervised Rotation Prediction (Gidaris et al., 2018). All baselines are given the same transductive access to unseen-class images that ULHM has, ensuring comparable evaluation conditions.

## 4.2 Image Sparse Recovery

We evaluate the ULHM framework's ability to recover complete observations from sparse measurements using semantic priors learned during training. During training, models observe complete semantic-observation pairs $(s_i, x_i)$ at a single sparsity level ($\rho = 0.10$). At test time, we evaluate recovery performance across multiple sparsity levels $\rho \in \{0.10, 0.15, 0.20, 0.25\}$ by providing only incomplete observations $\tilde{x}^{(m)} = x^{(m)} \odot M$, where mask $M$ retains only fraction $\rho$ of measurements, with no semantic guidance.

### 4.2.1 MNIST

**Setup.** MNIST digits ($28 \times 28 = 784$ pixels) are paired with semantic information consisting of: (1) complete ground truth images $x^{(1)} \in \mathbb{R}^{28 \times 28}$ and (2) class labels $s^{(1)} \in \{0, 1, ..., 9\}$. The framework is trained exclusively on $\rho = 0.10$ sparsity, where sparse observations $x^{(2)}$ retain only 10% of pixels through random masking with uniform spatial distribution as shown in Figure 2. At test time, we evaluate generalization by testing on sparsity levels ranging from 0.10 to 0.25 without any semantic information provided.

Table 1: Homeomorphism verification metrics of MNIST ($\kappa = 5$, Cosine distance)

| Level | Metric | $\rho = 0.10$ | $\rho = 0.15$ | $\rho = 0.20$ | $\rho = 0.25$ |
|---|---|---|---|---|---|
| Global Match | Betti number $\beta_0$ | 1 | 1 | 1 | 1 |
| | Sliced $W_2$ $\tau_w$ | 0.014 | 0.019 | 0.024 | 0.027 |
| Local Match | Trust $\tau_t$ | 0.800 | 0.902 | 0.937 | 0.953 |
| | Continuity $\tau_c$ | 0.906 | 0.899 | 0.890 | 0.882 |
| Semantic Match | Alignment $\tau_a$ | 0.027 | 0.030 | 0.034 | 0.038 |

**Homeomorphism Verification.** We assess the learned latent manifold across different sparsity regimes using five diagnostic metrics computed over the complete dataset (60,000 training and 10,000 test samples). Table 1 presents results with neighborhood size $\kappa = 5$. The 0-th Betti number $\beta_0 = 1$ across all sparsity levels is consistent with the latent space forming one connected manifold, suggesting that sparse and full encoders learn a unified geometric structure rather than separate components. Trust scores increase monotonically from 0.80 at $\rho = 0.10$ to 0.953 at $\rho = 0.25$, demonstrating that higher measurement density strengthens

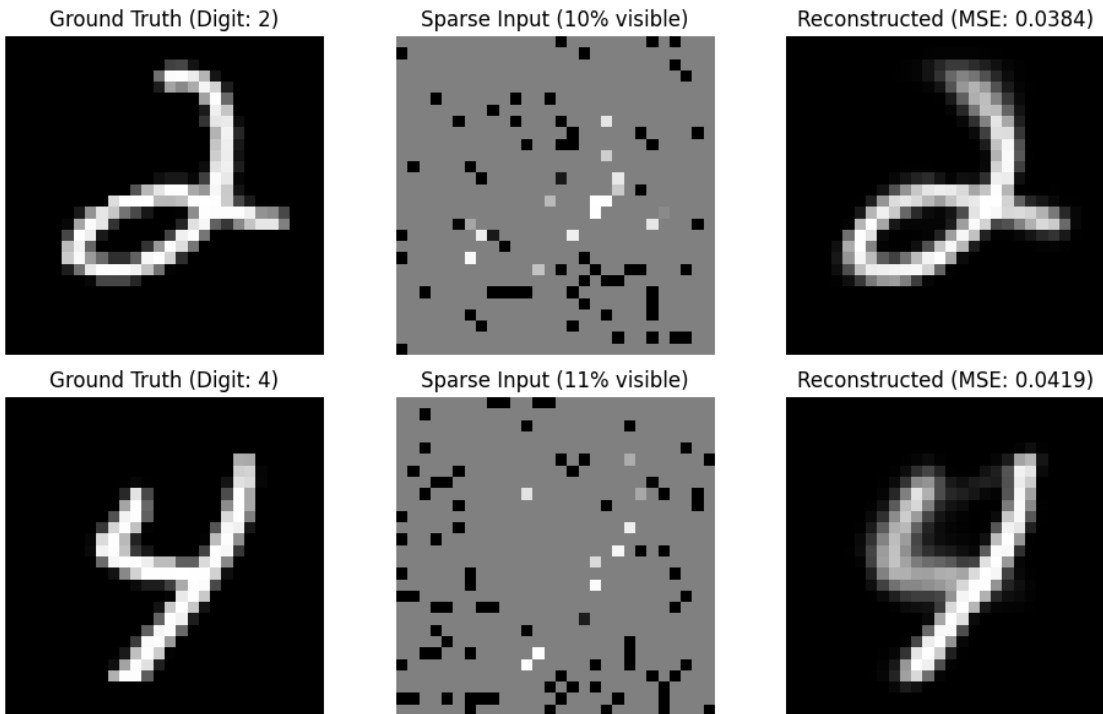

Figure 2: Sparse recovery on the MNIST digits. It shows ground truth images, heavily masked 10%–11% inputs, and the successful corresponding reconstructed image with low MSE.

semantic clustering. The negligible train-test difference (average $\Delta = 0.004$) and small Wasserstein distance ($W_2 \approx 0.03$, one order of magnitude below threshold 0.3) indicate robust generalization with no domain shift. Alignment error increases from 0.027 to 0.038, reflecting the framework's adaptive encoding: at low sparsity, encoders converge to canonical class representations (leveraging semantic priors), while at higher sparsity, encoders capture instance-specific details. This behavior, combined with $W_2 \approx 0.03$, is consistent with encoding at different granularities within the same manifold rather than mapping to separate manifolds. All sparsity levels satisfy the connectivity ($\beta_0 = 1$), trust ($\geq 0.80$), and co-location ($W_2 \ll 0.3$) criteria, providing finite-sample evidence consistent with one latent manifold with preserved structure.

Table 2: MNIST Sparse Recovery Performance Across Sparsity Levels

| Method | $\rho = 0.10$ | $\rho = 0.15$ | $\rho = 0.20$ | $\rho = 0.25$ |
|---|---|---|---|---|
| UNet (Ronneberger et al., 2015) | 0.084 | 0.051 | 0.035 | 0.027 |
| ResNet (He et al., 2016) | 0.074 | 0.044 | 0.031 | 0.023 |
| DenseNet (Huang et al., 2017) | 0.092 | 0.055 | 0.037 | 0.027 |
| AttentionUNet (Oktay et al., 2018) | 0.084 | 0.051 | 0.035 | 0.026 |
| VAE (Kingma & Welling, 2014) | 0.097 | 0.077 | 0.070 | 0.068 |
| **Ours** | **0.054** | **0.035** | **0.027** | **0.021** |

**Recovery Performance.** Table 2 presents reconstruction quality measured by mean squared error (MSE). Despite training exclusively on $\rho = 0.10$ data, ULHM achieves superior performance across all sparsity levels. At the training sparsity ($\rho = 0.10$), ULHM achieves MSE of 0.054, outperforming the best baseline ResNet (0.074) by 27.0%. This advantage becomes more pronounced at higher sparsities: at $\rho = 0.15$, ULHM (0.035) outperforms ResNet (0.044) by 20.5%; at $\rho = 0.20$, ULHM (0.027) outperforms ResNet (0.031) by 12.9%; and at $\rho = 0.25$, ULHM (0.021) outperforms ResNet (0.023) by 8.7%. Notably, ULHM's performance at $\rho = 0.25$ (0.021) is 61.1% better than its own performance at the training sparsity $\rho = 0.10$ (0.054), demonstrating effective generalization beyond the training distribution. The monotonic decrease in

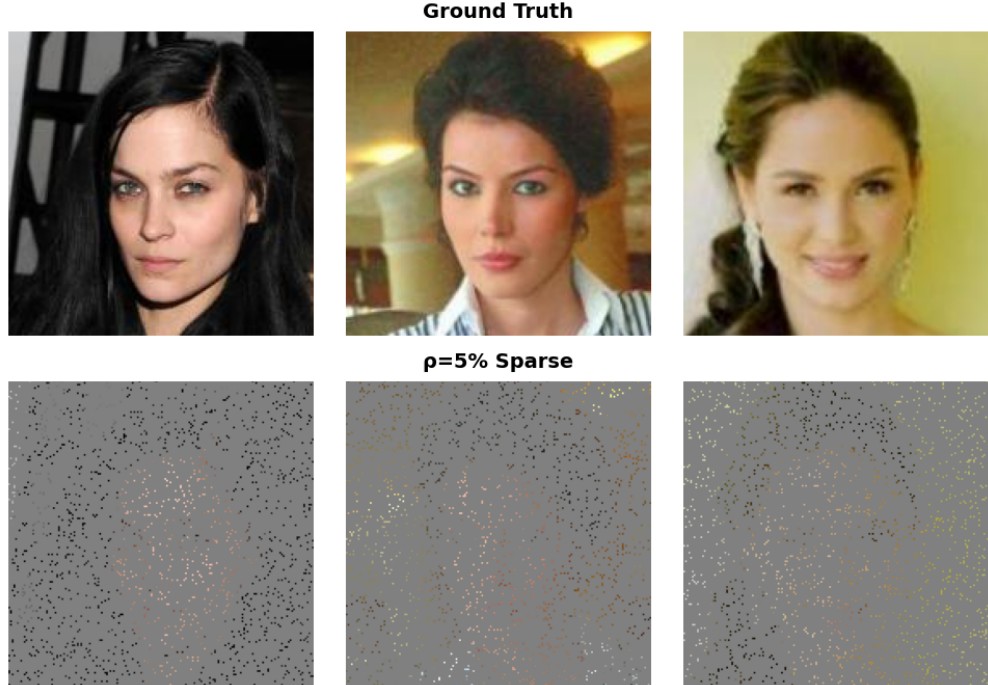

Figure 3: Illustrating the sparse CelebA-style image recovery settings with ground truth and random 5% sparsity.

MSE as more observations become available validates that semantic priors learned during training provide increasingly stronger geometric constraints for reconstruction.

### 4.2.2 CelebA

**Setup.** CelebA facial images ($64{\times}64{\times}3 = 12{,}288$ values) are paired with 40 binary facial attributes including *Black_Hair*, *Blond_Hair*, *Eyeglasses*, *Male*, *Smiling*, *Young*, etc. We test spatially structured masking patterns: (1) random pixel masking (Fig. 3), (2) block masking (removing $16{\times}16$ patches), and (3) inpainting (masking central $32{\times}32$ region).

Table 3: Topological Unification Verification - CelebA ($\kappa = 5$, Cosine distance)

| Level | Metric | $\rho = 0.05$ | $\rho = 0.08$ | $\rho = 0.10$ | $\rho = 0.12$ | $\rho = 0.15$ |
|---|---|---|---|---|---|---|
| Global Match | Betti number $\beta_0$ | 1 | 1 | 1 | 1 | 1 |
| | Sliced $W_2$ $\tau_w$ | 0.006 | 0.008 | 0.011 | 0.012 | 0.013 |
| Local Match | Trust $\tau_t$ | 0.877 | 0.873 | 0.871 | 0.868 | 0.866 |
| | Continuity $\tau_c$ | 0.888 | 0.889 | 0.885 | 0.886 | 0.879 |
| Semantic Match | Alignment $\tau_a$ | 0.043 | 0.041 | 0.043 | 0.046 | 0.055 |

**Homeomorphism Verification.** Table 3 provides five-metric diagnostic evidence that the latent structure remains compatible across all sparsity levels, with $\text{Trust}_5 \geq 0.866 > 0.80$ and $\text{Cont}_5 \geq 0.879 > 0.70$, meeting verification thresholds. The Betti number $\beta_0 = 1$ across all sparsity levels is consistent with the latent space forming a single connected manifold, indicating structural integration. Sliced $W_2$ distances remain extremely small ($W_2 \leq 0.013 \ll 0.30$), demonstrating that sparse and complete encoders map to geometrically coincident regions with substantial density overlap. Alignment error increases modestly from 0.043 to 0.055 as sparsity increases, reflecting adaptive encoding at different granularities while maintaining manifold coherence. The consistent satisfaction of all verification criteria (connectivity with $\beta_0 = 1$, local preservation with Trust > 0.80 and Continuity > 0.80, and geometric co-location with $W_2 \ll 0.30$) provides

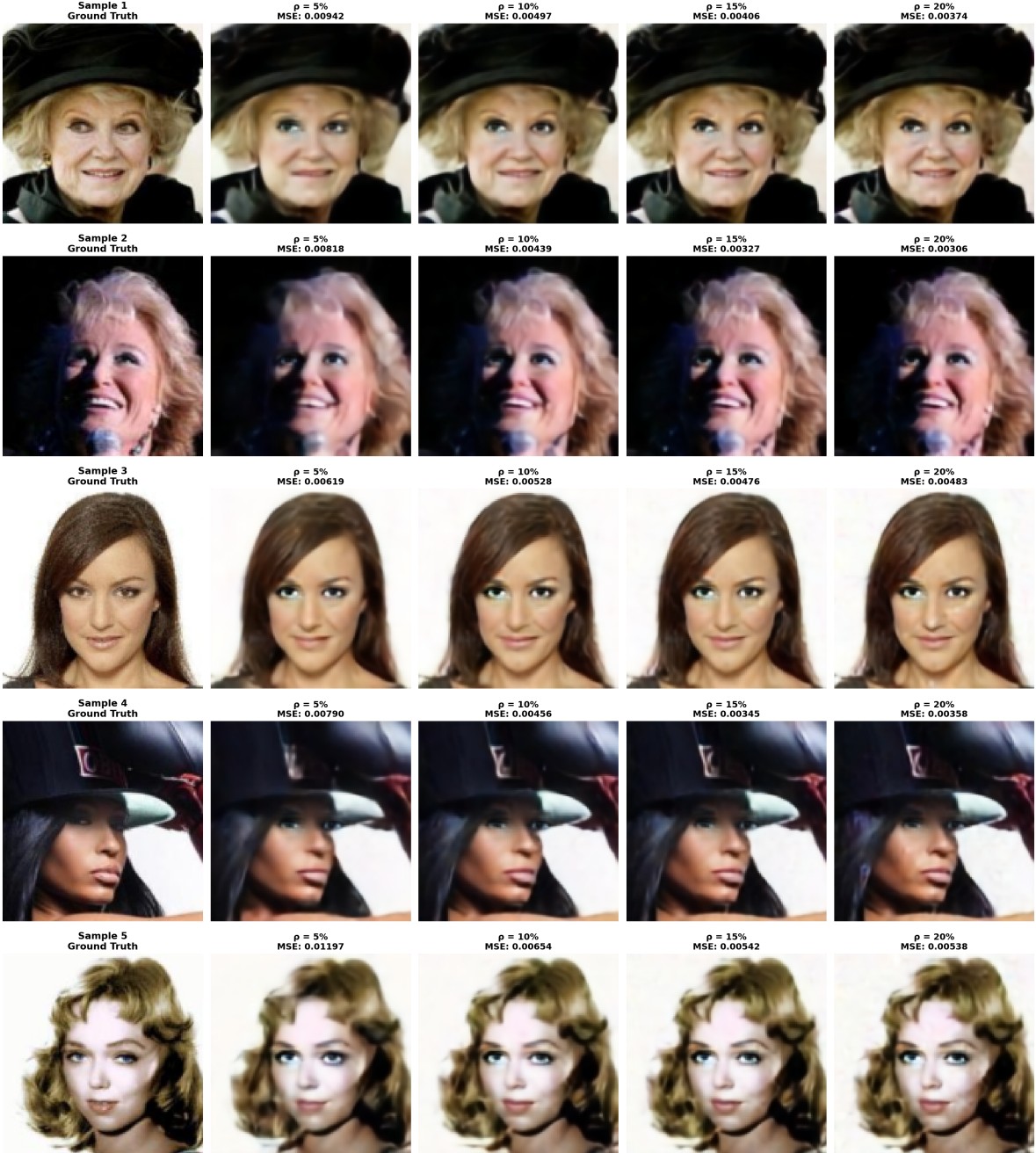

Figure 4: Sparse face reconstructions under increasing observation rates $\rho(5\% - 20\%)$, showing consistent visual improvement, and decreasing MSE.

finite-sample evidence that CelebA sparse recovery operates on a structurally compatible latent manifold consistent with homeomorphic properties.

**Sparse Recovery Results.** Table 4 shows reconstruction performance under Gaussian noise ($\sigma = 0.6$) across different sparsity levels. ULHM demonstrates consistent improvement as sparsity increases, achieving MSE of 0.0280 at $\rho = 0.10$, outperforming all baselines including Res-Attention (0.0307) and CBAM U-Net (0.0307). At higher sparsity levels, the gap widens significantly: at $\rho = 0.15$, ULHM achieves MSE of 0.0247 compared to the best baseline CBAM U-Net at 0.0273, demonstrating a 9.5% improvement. Notably, while Res-Attention performs best among baselines at $\rho = 0.05$ (0.0325 vs. ULHM's 0.0353), ULHM's

Table 4: CelebA sparse recovery with Gaussian noise ($\sigma = 0.6$)

| Method | $\rho = 0.05$ | $\rho = 0.08$ | $\rho = 0.10$ | $\rho = 0.12$ | $\rho = 0.15$ |
|---|---|---|---|---|---|
| Attention U-Net (Oktay et al., 2018) | 0.0406 | 0.0404 | 0.0401 | 0.0396 | 0.0388 |
| Dense U-Net (Li et al., 2018) | 0.0342 | 0.0334 | 0.0328 | 0.0321 | 0.0312 |
| DANet (Fu et al., 2019) | 0.0387 | 0.0364 | 0.0348 | 0.0332 | 0.0309 |
| Res-Attention (Wang et al., 2017) | **0.0325** | 0.0313 | 0.0307 | 0.0301 | 0.0292 |
| CBAM U-Net (Woo et al., 2018) | 0.0348 | 0.0322 | 0.0307 | 0.0293 | 0.0273 |
| ResUNet-PSP (Diakogiannis et al., 2020) | 0.0336 | 0.0319 | 0.0313 | 0.0306 | 0.0297 |
| ResUNet (Zhang et al., 2018) | 0.0360 | 0.0342 | 0.0330 | 0.0317 | 0.0297 |
| **ULHM (Ours)** | 0.0353 | **0.0305** | **0.0280** | **0.0262** | **0.0247** |

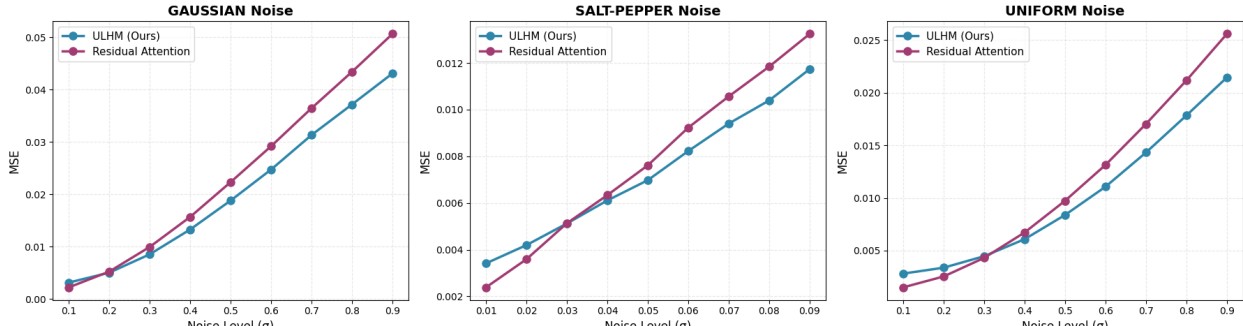

Figure 5: Mean Squared Error (MSE) under increasing Gaussian, Salt-Pepper, and Uniform noise. ULHM achieved lower than the Residual Attention baseline, demonstrating superior robustness.

performance scales more effectively with increased measurements, validating that semantic structure learned during training provides stronger geometric constraints as more observations become available.

**Robustness against noise.** Figure 5 depicts robustness of our method against different noise models. This was to evaluate how the reconstruction or prediction error increases with the input noise severity. ULHM consistently outperforms Residual Attention across Gaussian, salt-and-pepper and uniform noise, reducing MSE by 15-20% at high noise levels (e.g., 0.043 vs. 0.051 for Gaussian $\sigma = 0.9$, 0.012 vs. 0.014 for salt-and-pepper $\sigma = 0.09$, and 0.021 vs. 0.026 for uniform $\sigma = 0.9$). It resembles the ULHM's capability of handling any noise and impulsive pixel corruption.

## 4.3 Cross-Domain Integration and Transfer

We validate the framework's ability to integrate heterogeneous datasets through empirically assessed structural compatibility. To show that Algorithm 1 provides actionable predictions rather than post-hoc diagnostics, we evaluate three domain pairings that span a full spectrum of structural compatibility: complete failure, moderate compatibility, and successful unification.

Table 5: Zero-shot cross-domain transfer accuracy (%) — Incompatible: MNIST + Sine + ScrambledMNIST. Rows: target domain; columns: classifier source.

| **Target ↓ / Classifier →** | MNIST | Sine | Scrambled |
|---|---|---|---|
| MNIST | 96.70 | 9.77 | 10.13 |
| Sine | 10.33 | 8.77 | 10.33 |
| Scrambled | 10.83 | 10.83 | 9.97 |

Table 6: Zero-shot cross-domain transfer accuracy (%) — Incompatible: MNIST + Random Noise + R-MNIST. Rows: target domain; columns: classifier source.

| Target ↓ / Classifier → | MNIST | Noise | R-MNIST |
|---|---|---|---|
| MNIST | 96.67 | 10.27 | 11.50 |
| Noise | 9.90 | 9.83 | 9.90 |
| R-MNIST | 11.13 | 10.57 | 10.93 |

### 4.3.1 Scenario 1 — Complete Failure: Incompatible Domain Pairs

We construct two incompatible three-domain configurations as negative controls. **MNIST + Sine + ScrambledMNIST**: SineWave images are globally periodic with random labels, carrying no semantic structure; ScrambledMNIST applies per-sample pixel scrambling plus a checkerboard pattern, destroying all spatial structure while preserving label cardinality. **MNIST + Random Noise + R-MNIST**: Random Gaussian noise provides a structureless baseline; R-MNIST shares identical pixel statistics with MNIST but has randomly permuted class labels—the critical control that rules out label supervision as the driver of cross-domain transfer.

**Verification.** Both incompatible configurations fail four of five metrics in Algorithm 1: $\beta_0 > 1$, Trust $< 0.40$, Continuity $< 0.40$, and Alignment $> 0.70$. Crucially, $W_2$ passes in both cases (Table 7), confirming that distributional co-location alone is insufficient and must be interpreted jointly with $\beta_0$, Trust, and Continuity.

Table 7: Verification metrics for compatible versus incompatible domain triples. All configurations trained with identical objectives and hyperparameters.

| Configuration | $\beta_0$ | Trust | Cont. | $W_2$ | Align. | **Pass** |
|---|---|---|---|---|---|---|
| MNIST+USPS+SVHN (compat.) | 1 | 0.922 | 0.933 | 0.004 | 0.083 | ✓ |
| MNIST+Sine+Scrambled | 2 | 0.315 | 0.315 | 0.0013 | 0.733 | ✗ |
| MNIST+Noise+R-MNIST | 3 | 0.352 | 0.352 | 0.0012 | 0.817 | ✗ |

**Transfer.** Transfer to and from non-MNIST domains collapses to near-chance ($\leq 12\%$) in all directions, while within-MNIST accuracy remains high ($\geq 96\%$) (Tables 5 and 6). Near-chance transfer involving R-MNIST directly falsifies the hypothesis that label supervision alone drives cross-domain generalization: homeomorphic latent structure is the necessary condition, and the verification protocol correctly identifies its absence before deployment.

### 4.3.2 Scenario 2 — Moderate Compatibility: MNIST and Fashion-MNIST

We train ULHM jointly on MNIST handwritten digits and Fashion-MNIST clothing items using shared class labels $\{0, 1, ..., 9\}$. Table 8 shows comprehensive homeomorphism verification across the three-level hierarchy.

Table 8: Topological Unification Verification - Joint MNIST/Fashion-MNIST ($\kappa = 5$, Cosine distance)

| Level | Metric | $\rho = 0.10$ | $\rho = 0.15$ | $\rho = 0.20$ | $\rho = 0.25$ |
|---|---|---|---|---|---|
| Global Match | Betti number $\beta_0$ | 1 | 1 | 1 | 1 |
| | Sliced $W_2$ $\tau_w$ | 0.016 | 0.016 | 0.018 | 0.021 |
| Local Match | Trust $\tau_t$ | 0.860 | 0.920 | 0.940 | 0.949 |
| | Continuity $\tau_c$ | 0.782 | 0.779 | 0.770 | 0.769 |
| Semantic Match | Alignment $\tau_a$ | 0.288 | 0.236 | 0.218 | 0.203 |

The Betti number $\beta_0 = 1$ across all sparsity levels is consistent with MNIST and Fashion-MNIST occupying a single connected latent support rather than remaining disconnected. The sliced Wasserstein distance remains extremely small ($W_2 \leq 0.021 \ll 0.30$), demonstrating substantial density overlap where both domains occupy geometrically coincident regions. Trust scores exceed the threshold across all sparsity levels

($\text{Trust}_5 \geq 0.860 > 0.80$), which is consistent with the encoder maintaining local injectivity with bi-Lipschitz lower bound $c_1 > 0$, preventing manifold collapse. Continuity scores range from 0.769 to 0.782, which is higher than the 0.70 threshold, indicating reasonable local structure preservation. Alignment error decreases monotonically from 0.288 at $\rho = 0.10$ to 0.203 at $\rho = 0.25$, falling within the relaxed threshold of 0.30 for heterogeneous cross-domain integration. This reveals a universal manifold where MNIST and Fashion-MNIST occupy distinct but geometrically adjacent regions within a unified topological structure. All three conditions pass decisively, providing empirical evidence that MNIST and Fashion-MNIST exhibit structural compatibility consistent with a unified latent manifold, meeting the empirical thresholds associated with the homeomorphism criterion, enabling valid cross-domain transfer without structural incompatibility.

**Universal Classifier Transfer.** Table 9 shows cross-domain transfer results in both directions.

Table 9: Cross-domain classifier transfer between MNIST and Fashion-MNIST

| Method | MNIST→F-MNIST | F-MNIST→MNIST |
|---|---|---|
| Partial Convolution (Liu et al., 2018) | 11.71 | 13.80 |
| Max. Classifier Discrepancy (Saito et al., 2018) | 49.35 | 42.40 |
| Cross-Domain Self-supervised (Kim et al., 2020) | 50.74 | 20.28 |
| Min. Class Confusion (Jin et al., 2020) | 61.65 | 58.87 |
| Cond. Domain Adversarial (Long et al., 2018) | 78.66 | 89.38 |
| Correlation Alignment (Sun & Saenko, 2016) | 79.05 | 83.60 |
| Prototypical Networks (Snell et al., 2017) | 84.15 | 93.32 |
| **ULHM (Ours)** | **86.73** | **96.97** |

All values in %.

For MNIST→Fashion-MNIST transfer, our universal classifier trained exclusively on MNIST achieves 86.73% accuracy on Fashion-MNIST with zero additional training, outperforming the best baseline Prototypical Networks (84.15%) by 2.58 percentage points and substantially exceeding other methods such as Correlation Alignment (79.05%) and Conditional Domain Adversarial (78.66%). For the reverse direction Fashion-MNIST→MNIST, ULHM achieves 96.97% accuracy, outperforming Prototypical Networks (93.32%) by 3.65 percentage points. The asymmetric transfer performance (96.97% vs. 86.73%) reflects the inherent complexity difference between the target domains: MNIST digits have simpler, more canonical structures compared to Fashion-MNIST clothing items with higher intra-class variability. Consistent with the moderate verification scores (Trust marginally above threshold, Alignment = 0.288 at $\rho = 0.10$), transfer performance is strong but not maximal, confirming that the degree of structural compatibility predicts the level of transfer accuracy.

### 4.3.3 Scenario 3 — Successful Unification: MNIST, USPS, and SVHN

We train ULHM on three digit-recognition domains—MNIST (handwritten), USPS (postal scans), and SVHN (street-view house numbers)—which share semantic structure across substantially different visual statistics. Table 10 reports verification metrics at $\rho = 0.10$.

Table 10: Topological Unification Verification — MNIST / USPS / SVHN

| Level | Metric | Value |
|---|---|---|
| Global Match | Betti number $\beta_0$ | 1 |
| | Sliced $W_2$ $\tau_w$ | 0.004 |
| Local Match | Trust $\tau_t$ | 0.922 |
| | Continuity $\tau_c$ | 0.933 |
| Semantic Match | Alignment $\tau_a$ | 0.083 |

All five metrics pass their thresholds decisively: $\beta_0 = 1$ is consistent with a single connected manifold, $W_2 = 0.004$ indicates near-perfect geometric co-location, Trust = 0.922 and Continuity = 0.933 indicate strong local structure preservation, and Alignment = 0.083 demonstrates tight cross-domain semantic correspondence.

These scores are markedly stronger than the MNIST/Fashion-MNIST case, reflecting the closer semantic and structural alignment among digit-recognition domains.

**Zero-shot transfer across all domain pairs.** Table 11 shows that a classifier trained on any one domain transfers successfully to both others, with accuracy $\geq 83\%$ in all nine source-target combinations.

Table 11: Zero-shot cross-domain transfer accuracy (%) for MNIST / USPS / SVHN. Each row is the target domain; each column is the classifier source. All nine combinations exceed 83%.

| Target ↓ / Classifier → | MNIST | USPS | SVHN |
|---|---|---|---|
| MNIST | 97.37 | 96.10 | 97.40 |
| USPS | 98.36 | 98.36 | 97.81 |
| SVHN | 86.46 | 83.79 | 86.87 |

The uniformly high transfer accuracy in all directions supports the conclusion that when structural compatibility is strong—as indicated by verification scores well above threshold—zero-shot transfer generalizes robustly regardless of which domain supplies the classifier.

**Summary across three scenarios.** Together, the three scenarios form a consistent gradient: complete failure ($\leq 12\%$ transfer, verification fail) $\rightarrow$ moderate compatibility (86.73% transfer, verification marginal) $\rightarrow$ successful unification ($\geq 83\%$ in all directions, verification strong). This progression supports the interpretation that the verification protocol provides a graded, interpretable predictor of downstream transfer performance, and that homeomorphic latent structure—not label supervision alone—is the operative condition for successful cross-domain generalization.

## 4.4 Transductive Zero-Shot Classification

We evaluate transductive zero-shot classification where the auxiliary classifier $C_\theta$ is trained on a subset of classes, then tested on unseen classes via nearest-centroid classification in the learned latent space. We use three datasets of varying complexity: MNIST (10 classes, grayscale), Fashion-MNIST (10 classes, grayscale clothing), and CIFAR-10 (10 classes, RGB natural images).

**Setup.** For all three datasets, we train the encoder $E_x^{(m)}$ and decoder $D_\theta^{(m)}$ on all classes $\{0, 1, ..., 9\}$ through reconstruction objectives. However, the auxiliary classifier $C_\theta$ is trained on only classes $\{0, 1, 2, 3, 4\}$, with classes $\{5, 6, 7, 8, 9\}$ held out. In this transductive setting, the framework has access to the complete set of visual observations (pixels) for every image in the dataset (classes 0-9) during training. However, explicit class labels are only provided for the seen subset (classes 0-4). Critically, the model never accesses the specific label associated with any individual image in the unseen subset (classes 5-9). The encoder observes the visual structure of unseen data through reconstruction loss $\mathcal{L}_{\text{recon}}^x$ and learns its geometric organization through topology preservation $\mathcal{L}_{\text{local}}$, but is never told "this specific image belongs to class $c$" for any $c \in \{5, 6, 7, 8, 9\}$. At test time, we classify samples from all classes using nearest-centroid classification in latent space.

**Centroid Computation.** For each class $c$, we compute the latent centroid:

$$\mu_c = \frac{1}{|\mathcal{I}_c|} \sum_{i \in \mathcal{I}_c} E_x^{(m)}(x_i^{(m)}; \phi)$$

where $\mathcal{I}_c$ is the set of training sample indices with class label $c$. Crucially, centroids for unseen classes can be computed because the encoder processed these samples during reconstruction training, even though the classifier never saw their labels.

**Zero-Shot Results.** Table 12 shows classification performance across all three datasets. ULHM achieves 89.47% accuracy on MNIST unseen classes, outperforming the best baseline Contrastive Learning (87.90%) by 1.57 percentage points. On Fashion-MNIST, ULHM achieves 84.70% accuracy, outperforming Mixup Augmentation (82.69%) by 2.01 percentage points. Most notably, on CIFAR-10 with complex natural RGB images, ULHM achieves 78.76% accuracy, substantially outperforming Contrastive Learning (62.10%) by

Table 12: Transductive zero-shot classification accuracy on unseen classes (5–9)

| Method | MNIST | Fashion-MNIST | CIFAR-10 |
|---|---|---|---|
| Contrastive Learning (Chen et al., 2020b) | 87.90 | 78.53 | 62.10 |
| Mixup Augmentation (Zhang et al., 2017) | 87.79 | 82.69 | 45.60 |
| Prototypical Networks (Snell et al., 2017) | 67.76 | 49.01 | 53.64 |
| Correlation Alignment (Sun & Saenko, 2016) | 75.58 | 54.02 | 53.97 |
| Self-Supervised Rotation (Gidaris et al., 2018) | 38.26 | 60.45 | 47.70 |
| Max. Classifier Discrepancy (Saito et al., 2018) | 62.75 | 56.58 | 57.86 |
| Triplet Networks (Hoffer & Ailon, 2015) | 77.25 | 60.40 | 51.03 |
| Variational Autoencoders (Kingma & Welling, 2014) | 85.76 | 80.80 | 54.52 |
| **ULHM (Ours)** | **89.47** | **84.70** | **78.76** |

16.66 percentage points and demonstrating superior generalization to high intra-class variability. The consistent performance across datasets is consistent with the hypothesis that homeomorphic structure enables geometric separation of unseen classes through preserved topology. Unlike methods such as Prototypical Networks (67.76%, 49.01%, 53.64%) or Triplet Networks (77.25%, 60.40%, 51.03%) that show inconsistent performance across datasets, ULHM maintains strong accuracy by leveraging the bi-Lipschitz encoder mapping that preserves local neighborhood structure while preventing manifold collapse. All baselines are provided the same transductive access to unseen-class images, ensuring the comparison is on equal footing. We note that these results are not directly comparable to inductive zero-shot benchmarks where unseen-class images are entirely absent from training.

## 5 Conclusion

We introduced the Universal Latent Homeomorphic Manifold (ULHM) framework, establishing homeomorphism as the foundational mathematical criterion for unifying semantic and observation representations into a single latent structure. Critically, ULHM employs homeomorphism at two complementary stages: training objectives actively learn bi-Lipschitz homeomorphic properties during optimization ($\mathcal{L}_{\text{recon}}$ maintains the lower bound preventing collapse, $\mathcal{L}_{\text{local}}$ bounds the upper constant preventing tearing, $\mathcal{L}_{\text{consist}}$ ensures domain overlap), while post-training verification algorithms assess finite-sample evidence consistent with structural compatibility on finite samples. By leveraging conditional variational inference alongside practical verification algorithms that provide necessary but not sufficient empirical evidence for homeomorphic structure (trust, continuity, Wasserstein distance, Betti numbers, alignment) organized in a three-level hierarchical protocol, ULHM learns continuous manifold-to-manifold transformations that preserve relational structure between modalities. Experimental validation demonstrates three core capabilities enabled by learned homeomorphic structure together with empirical structural-compatibility diagnostics. First, semantic-guided sparse recovery achieves superior reconstruction quality, with ULHM outperforming all baselines on CelebA under noise. Second, cross-domain classifier transfer succeeds without retraining, achieving 86.73% MNIST→F-MNIST accuracy and surpassing state-of-the-art domain adaptation methods. Third, transductive zero-shot classification on unseen classes reaches 78.76% on CIFAR-10, exceeding prior work by 16.66%. The homeomorphism criterion provides a principled framework, validated on the tested domains, that suggests a structured basis for decomposing broad models into domain-specific components, enabling principled model composition with empirically supported conditions rather than formal guarantees. Future work will extend to efficient amortized inference, non-Euclidean latent geometries, and networked cyber-physical and IoT systems. Future work will also include fully inductive zero-shot evaluations, systematic sensitivity analyses of verification thresholds, and uncertainty quantification through repeated experimental runs.

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

# A   Appendix

In this appendix, we provide rigorous derivations for the theoretical results presented in the main text. We assume the standard metric topology induced by the Euclidean distance on $\mathbb{R}^d$ and the geodesic distance on the manifolds.

## A.1   Proof of Theorem 1 (Topological Unification)

**Theorem 1 (Topological Unification via Latent Identifications).** *Let $\{\mathcal{M}_x^{[k]}\}_{k=1}^K$ be pairwise disjoint topological spaces with continuous encoders $E^{[k]} : \mathcal{M}_x^{[k]} \to \mathbb{R}^d$. Define the latent images $\mathcal{M}_z^{[k]} := E^{[k]}(\mathcal{M}_x^{[k]})$ and unified latent support*

$$\mathcal{U} := \bigcup_{k=1}^K \mathcal{M}_z^{[k]} \subset \mathbb{R}^d$$

*equipped with the subspace topology induced from $\mathbb{R}^d$. Let $X := \bigsqcup_{k=1}^{K} \mathcal{M}_x^{[k]}$ (disjoint union with coproduct topology) and define $E : X \to \mathcal{U}$ by $E|_{\mathcal{M}_x^{[k]}} = E^{[k]}$. Define an equivalence relation $\sim$ on $X$ by $x \sim x'$ iff $E(x) = E(x')$. Then the induced map $\widetilde{E} : X/\sim \to \mathcal{U}$ is a homeomorphism provided:*

*(i)* (Local Homeomorphic Embeddings) *Each $E^{[k]}$ is a homeomorphism onto its image $\mathcal{M}_z^{[k]}$ (with the subspace topology inherited from $\mathbb{R}^d$), and either all $E^{[k]}$ are open maps or all $E^{[k]}$ are closed maps.*

*(ii)* (Open/Closed Cover) *If all $E^{[k]}$ are open, then each $\mathcal{M}_z^{[k]}$ is open in $\mathcal{U}$ and $\{\mathcal{M}_z^{[k]}\}_{k=1}^{K}$ covers $\mathcal{U}$. If all $E^{[k]}$ are closed, then each $\mathcal{M}_z^{[k]}$ is closed in $\mathcal{U}$ and $\{\mathcal{M}_z^{[k]}\}_{k=1}^{K}$ covers $\mathcal{U}$.*

*Proof.* Condition (i) ensures that each component $\mathcal{M}_x^{[k]}$ is embedded faithfully into latent space with no local collapse or topological distortion. Condition (ii) ensures that the latent images glue together properly in $\mathcal{U}$.

Let $\pi : X \to X/\sim$ be the quotient map. Since $E$ is constant on $\sim$-equivalence classes by definition, the universal property of quotient spaces yields a unique continuous map $\widetilde{E} : X/\sim \to \mathcal{U}$ such that

$$E = \widetilde{E} \circ \pi.$$

**Bijectivity.** By construction, $\mathcal{U} = E(X)$, so $\widetilde{E}$ is surjective. If $\widetilde{E}([x]) = \widetilde{E}([x'])$, then $E(x) = E(x')$, hence $x \sim x'$ and $[x] = [x']$. Thus $\widetilde{E}$ is injective.

**$E$ is a quotient map.** We prove that $E$ is an open surjection (the closed case is analogous, replacing "open" by "closed"). Let $U \subseteq X$ be open. Since $X$ has the coproduct topology, $U \cap \mathcal{M}_x^{[k]}$ is open in $\mathcal{M}_x^{[k]}$ for each $k$. Because each $E^{[k]}$ is an open homeomorphism onto its image,

$$E^{[k]}(U \cap \mathcal{M}_x^{[k]}) = E(U) \cap \mathcal{M}_z^{[k]}$$

is open in $\mathcal{M}_z^{[k]}$ with its subspace topology.

By Condition (ii), each $\mathcal{M}_z^{[k]}$ is open in $\mathcal{U}$. Therefore a set that is open in $\mathcal{M}_z^{[k]}$ is also open in $\mathcal{U}$, so $E(U) \cap \mathcal{M}_z^{[k]}$ is open in $\mathcal{U}$ for every $k$. Hence

$$E(U) = \bigcup_{k=1}^{K} (E(U) \cap \mathcal{M}_z^{[k]})$$

is a finite union of open sets in $\mathcal{U}$, and is therefore open in $\mathcal{U}$. Thus $E$ is an open surjection, hence a quotient map.

**$\widetilde{E}$ is a homeomorphism.** Let $V \subseteq \mathcal{U}$. Since $E = \widetilde{E} \circ \pi$, we have

$$E^{-1}(V) = \pi^{-1}(\widetilde{E}^{-1}(V)).$$

Since $\pi$ is a quotient map, $\widetilde{E}^{-1}(V)$ is open in $X/\sim$ if and only if $\pi^{-1}(\widetilde{E}^{-1}(V)) = E^{-1}(V)$ is open in $X$. Since $E$ is a quotient map, $E^{-1}(V)$ is open in $X$ if and only if $V$ is open in $\mathcal{U}$. Therefore $V$ is open in $\mathcal{U}$ if and only if $\widetilde{E}^{-1}(V)$ is open in $X/\sim$, so $\widetilde{E}$ is a quotient map.

Since $\widetilde{E}$ is bijective and quotient, it is a homeomorphism. $\qquad\qquad\square$

## A.2 Proof of Theorem 2 (Bi-Lipschitz Sufficiency)

**Theorem 2.** *Let $(\mathcal{M}_x, d_X)$ be a metric space and let $\mathcal{M}_z \subseteq \mathbb{R}^d$ be equipped with the Euclidean metric $d_Z = \|\cdot\|_2$ (i.e., $\mathcal{M}_z$ has the subspace topology inherited from $\mathbb{R}^d$). If the encoder $E : \mathcal{M}_x \to \mathcal{M}_z$ is $(c_1, c_2)$-bi-Lipschitz with $c_1 > 0$, then $E$ is an open homeomorphism onto its image. Thus, the bi-Lipschitz constraint satisfies Condition (i) of Theorem 1.*

*Proof.* Recall that $E$ is $(c_1, c_2)$-bi-Lipschitz if for all $x, y \in \mathcal{M}_x$,

$$c_1 d_X(x, y) \leq d_Z(E(x), E(y)) \leq c_2 d_X(x, y). \tag{28}$$

We must show that $E$ is a continuous bijection onto its image with continuous inverse.

**1. Injectivity (Prevention of Collapse)** Let $x, y \in \mathcal{M}_x$ be distinct. Since $d_X$ is a metric, $d_X(x, y) > 0$. Using the lower bound in equation 28:

$$d_Z(E(x), E(y)) \geq c_1 d_X(x, y) > 0, \tag{29}$$

where the final inequality holds since $c_1 > 0$. Therefore $E(x) \neq E(y)$, proving $E$ is injective.

**2. Continuity of $E$ (Prevention of Tearing)** If $c_2 = 0$, then equation 28 implies $d_Z(E(x), E(y)) = 0$ for all $x, y$, so $E$ is constant and $\mathcal{M}_x$ must be a singleton (since $c_1 > 0$), in which case the claim is trivial. Hence assume $c_2 > 0$. The upper bound in equation 28 gives $d_Z(E(x), E(y)) \leq c_2 d_X(x, y)$, i.e., $E$ is Lipschitz and therefore continuous.

**3. Continuity of $E^{-1}$ (Metric Stability).** Let $\mathcal{M}'_z := E(\mathcal{M}_x) \subseteq \mathcal{M}_z$, equipped with the subspace metric induced by $d_Z$. By definition, $E : \mathcal{M}_x \to \mathcal{M}'_z$ is surjective, and by Paragraph 1 it is injective; hence $E$ is bijective and admits an inverse $E^{-1} : \mathcal{M}'_z \to \mathcal{M}_x$.

Let $u, v \in \mathcal{M}'_z$. Then there exist unique $x, y \in \mathcal{M}_x$ such that $E(x) = u$ and $E(y) = v$. Using the lower bound in equation 28, we obtain

$$c_1 d_X(x, y) \leq d_Z(E(x), E(y)) = d_Z(u, v), \tag{30}$$

$$\text{so } d_X(x, y) \leq \frac{1}{c_1} d_Z(u, v), \text{ i.e., } d_X(E^{-1}(u), E^{-1}(v)) \leq \frac{1}{c_1} d_Z(u, v). \tag{31}$$

Thus $E^{-1}$ is Lipschitz continuous with constant $1/c_1$, and in particular continuous.

**Conclusion** Therefore $E : \mathcal{M}_x \to E(\mathcal{M}_x)$ is a homeomorphism onto its image. Since a homeomorphism is an open map, $E$ is an open map onto $E(\mathcal{M}_x)$. In particular, applying the same argument to each modality shows that each $E^{[k]} : \mathcal{M}_x^{[k]} \to \mathcal{M}_z^{[k]}$ is an open homeomorphism onto its image, satisfying Condition (i) of Theorem 1. $\qquad\square$

### A.3 Mathematical Justification for Metric Selection

We establish the connection between violations of theoretical manifold properties (Theorems 1 and 2) and detectable signatures in computable metrics. The arguments are one-directional: each theoretical failure mode implies characteristic degradation in at least one metric in our verification pipeline. While these do not constitute formal proofs of equivalence, they provide principled justification for our metric choices.

**Level 1: Global Match (Unification Gatekeepers) Part 1: Structural Fragmentation.** A necessary empirical indicator of successful topological unification is that the sampled latent union

$$\mathcal{Z}_{\text{total}} := \bigcup_k \mathcal{M}_z^{[k]}$$

behaves as a single connected component at an appropriate neighborhood scale. The 0-th Betti number $\beta_0$, computed via persistent homology, counts the number of connected components.

If the conditions of Theorem 1 fail such that domains remain topologically disjoint, then for $\epsilon$ in the operating range (or across a stable persistence interval), there exist disjoint subsets $\mathcal{A}, \mathcal{B} \subset \mathcal{Z}_{\text{total}}$ with no connecting paths in the $\epsilon$-neighborhood graph. This topological disconnection is detected by $\beta_0 > 1$.

Because finite samples may appear disconnected at very small scales, $\beta_0$ is interpreted jointly with geometric metrics rather than in isolation. The combination $\beta_0 > 1$ with high pairwise Wasserstein distances provides strong evidence of genuine structural fragmentation.

**Part 2: Geometric Misalignment.** Let $\mu = \hat{\mathbb{P}}_z^{[i]}$ and $\nu = \hat{\mathbb{P}}_z^{[j]}$ be empirical latent distributions for two domains. The Wasserstein-2 distance is defined as

$$W_2^2(\mu, \nu) = \inf_{\gamma \in \Pi(\mu,\nu)} \mathbb{E}_{(z,z')\sim\gamma}[\|z - z'\|^2],$$

where $\Pi(\mu, \nu)$ denotes all couplings with marginals $\mu$ and $\nu$.

Suppose the supports are geometrically separated by a margin $\Delta > 0$:

$$\inf_{z\in\text{supp}(\mu),\, z'\in\text{supp}(\nu)} \|z - z'\| \geq \Delta.$$

Then for any coupling $\gamma \in \Pi(\mu, \nu)$, we have

$$\mathbb{E}_{(z,z')\sim\gamma}[\|z - z'\|^2] \geq \Delta^2,$$

since any transport plan must move mass across the separation gap. Therefore,

$$W_2^2(\mu, \nu) \geq \Delta^2.$$

This shows that when two latent domain supports are separated by a positive margin, the Wasserstein distance must be large. Such geometric separation is a common failure mode of unification—leading to negligible overlap in latent space—and is therefore detectable via $W_2$.

**Level 2: Local Match (Homeomorphic Integrity)  Part 3: Manifold Collapse.** The bi-Lipschitz lower bound $c_1 > 0$ (Theorem 2) guarantees injectivity. If this bound is violated (i.e., $c_1 \to 0$), there exist distinct points $x_q \neq x_{\text{adv}}$ such that

$$\|E(x_q) - E(x_{\text{adv}})\| \to 0.$$

When these points correspond to different semantic labels ($y_q \neq y_{\text{adv}}$), the $\kappa$-nearest neighborhood $\mathcal{N}_\kappa(E(x_q))$ in latent space will contain semantically inconsistent points.

The neighborhood label purity score

$$\text{Purity}_\kappa(x_q) = \frac{1}{\kappa} \sum_{z_j \in \mathcal{N}_\kappa(E(x_q))} \mathbb{1}[y_j = y_q]$$

measures the fraction of neighbors sharing the same label. As label-inconsistent points enter the neighborhood due to collapse, the purity score decreases. Systematic collapse across the manifold yields statistically significant purity degradation relative to a well-separated baseline embedding.

Note that this argument assumes semantic labels correlate with manifold structure, which is reasonable for supervised settings but may not hold universally.

**Part 4: Structural Incoherence.** The Continuity metric (Venna & Kaski, 2001) evaluates whether $\kappa$-nearest neighbors in the latent space $\mathcal{Z}$ were also neighbors in the observation space $\mathcal{X}$:

$$\text{Cont}_\kappa = 1 - \frac{2}{N\kappa(2N - 3\kappa - 1)} \sum_{i=1}^{N} \sum_{j \in \mathcal{N}_i^\kappa(\mathcal{Z})} \max(0, r_\mathcal{X}(i, j) - \kappa),$$

where $r_\mathcal{X}(i, j)$ denotes the rank of $j$ in $i$'s neighborhood in the original space.

While Theorem 1 only requires homeomorphisms (which preserve topology but not necessarily distances), practical embeddings should preserve local geometric structure. If the embedding severely distorts local neighborhoods (e.g., via folding, tearing, or strong metric distortion inconsistent with the bi-Lipschitz upper bound $c_2$), points that were distant in $\mathcal{X}$ may appear as neighbors in $\mathcal{Z}$. This increases the penalty terms $\max(0, r_\mathcal{X}(i, j) - \kappa)$ and reduces Continuity.

**Level 3: Semantic Match (Synchronization) Part 5: Inconsistent Mapping.** For multimodal datasets where paired modalities $(x_i^{(m)}, s_i^{(l)})$ represent the same underlying entity, successful unification requires consistent latent mappings. The Alignment Error

$$\tau_a = \frac{1}{N_{\text{paired}}} \sum_{i=1}^{N_{\text{paired}}} \|E_x^{(m)}(x_i^{(m)}) - E_s^{(l)}(s_i^{(l)})\|_2$$

quantifies whether paired samples map to nearby latent coordinates.

Large values of $\tau_a$ indicate failure of cross-modal synchronization. While this condition is not explicitly part of Theorems 1 or 2, it is necessary for practical unified reasoning across modalities. High alignment error suggests that either the encoders have not learned the correct cross-modal correspondence, or the modalities contain complementary rather than redundant information requiring separate latent subspaces.

### A.4 Guide to Supplementary Material

The supplementary document provides four bodies of supporting evidence organized into the following sections.

**Section S1: Sensitivity Analysis and Ablation Studies.** This section validates the contribution of each loss component across all three applications by varying one weight at a time while holding all others fixed. It reports the effect of the cross-modal consistency weight $\lambda_c$ on sparse recovery (Application 1) and on cross-domain transfer (Application 2), and the effects of the local preservation weight $\lambda_\ell$ and the perceptual weight $\lambda_p$ on zero-shot classification (Application 3). For the $\lambda_c$ sweeps in Applications 1 and 2, downstream task performance is paired with explicit verification metrics and pass/fail outcomes; the $\lambda_\ell$ and $\lambda_p$ sweeps focus on downstream sensitivity. The key finding from the explicit verification sweeps is that $\lambda_c = 0$ triggers categorical failure in Applications 1 and 2, while the other reported ablations show smoother performance degradation rather than abrupt task collapse.

**Section S2: Negative Control Experiments.** This section provides the full evidence for the three-scenario transfer spectrum discussed in Section 4. Complete $3 \times 3$ transfer accuracy matrices and verification metric breakdowns are reported for two structurally incompatible domain triples—MNIST paired with non-digit domains—alongside the compatible MNIST/USPS/SVHN baseline. A particularly important control replaces one domain with a label-scrambled version that preserves all pixel statistics while destroying semantic structure. Near-chance transfer ($\leq 12\%$ in all cross-domain directions) under this condition rules out label supervision as the driver of successful transfer, isolating homeomorphic latent structure as the necessary condition.

**Section S3: Scalability of the Verification Protocol.** This section examines how many samples are needed for reliable metric estimates. Convergence of all five verification metrics is evaluated by drawing ten independent subsamples at increasing sizes and reporting mean $\pm$ standard deviation, demonstrating that stable estimates are obtained well before the full dataset is required.

**Section S4: Additional Domain Experiments.** This section tests the generality of the ULHM framework beyond the image benchmarks in the main paper. The first experiment applies the framework to accelerated MRI reconstruction, reporting sparse recovery performance on brain MRI data under clean and noisy conditions at multiple undersampling ratios, with verification metrics that are consistent with structural compatibility between image-space and frequency-space ($k$-space) manifolds. The second experiment applies the framework to power grid state estimation, where inputs are multivariate sensor readings and the transfer challenge involves unseen grid topologies. Both experiments include full verification metric tables, demonstrating that Algorithm 1 correctly identifies structural compatibility in non-visual, non-classification domains.

**Remark 5** (Connection to Transfer Learning). *When all verification metrics indicate successful unification—specifically, when domains $i$ and $j$ satisfy $\beta_0 \approx 1$, low pairwise $W_2$, high neighborhood label purity, high Continuity, and (for paired data) low Alignment Error. We have empirical evidence that the latent manifolds $\mathcal{M}_z^{[i]}$ and $\mathcal{M}_z^{[j]}$ are indistinguishable at the resolution of our measurements.*

*In this regime, a neural network trained on $\mathcal{M}_z^{[i]}$ for a fixed task encounters no detectable distributional shift at the resolution of our measurements when applied to $\mathcal{M}_z^{[j]}$ (up to finite-sample variability). This justifies direct transfer learning: the same network parameters can be applied to both domains without fine-tuning, as both domains are represented within the same geometric and topological structure.*

*This empirical transfer principle is a practical consequence of successful manifold unification, though it is not a formal mathematical theorem.*

