# Supplementary Material
## Universal Latent Homeomorphic Manifolds

## Contents

# 1 Sensitivity Analysis and Ablation Studies

This section reports sensitivity analyses and ablation studies addressing the hyperparameter concerns raised in the review process. For each application, we vary one loss weight at a time while holding all others fixed at the values reported in the main paper. The $\lambda_c$ sweeps in Applications 1 and 2 report both downstream task performance and the corresponding verification metrics used by Algorithm 1, while the $\lambda_\ell$ and $\lambda_p$ sweeps emphasize downstream task sensitivity under the same experimental setup. All experiments use the same architecture, dataset splits, and evaluation protocol as the main paper.

## 1.1 Effect of $\lambda_c$ on Sparse Recovery (Application 1)

Table 1 reports the effect of varying the cross-modal consistency weight $\lambda_c$ on MNIST sparse recovery (Application 1, $\rho = 0.10$). All other weights are held fixed at their default values.

Table 1: Sensitivity to $\lambda_c$ on MNIST sparse recovery. Verification metrics are computed at the best validation checkpoint. † indicates the default value used in the main paper.

| $\lambda_c$ | MSE ↓ | $\beta_0$ | Trust $\tau_t$ | Cont. $\tau_c$ | Align. $\tau_a$ | $W_2$ |
|---|---|---|---|---|---|---|
| 0.0 | 0.0631 | 2 | 0.788 | 0.6236 | 0.7036 | 0.623 |
| 0.01† | 0.0539 | 1 | 0.800 | 0.9062 | 0.0268 | 0.014 |
| 0.1 | 0.0572 | 1 | 0.807 | 0.9124 | 0.0089 | 0.024 |
| 0.5 | 0.0582 | 1 | 0.803 | 0.9014 | 0.0035 | 0.007 |
| 1.0 | 0.0590 | 1 | 0.804 | 0.8957 | 0.0024 | 0.004 |

At $\lambda_c = 0$ the encoders train independently: the latent space fragments into two disconnected components ($\beta_0 = 2$), neighborhood structure collapses (Trust = 0.788, Continuity = 0.624), and distributions diverge ($W_2 = 0.623$, Alignment = 0.704), yielding the worst reconstruction (MSE = 0.0631). At $\lambda_c = 0.01$ the manifold unifies ($\beta_0 = 1$), with $W_2$ collapsing to 0.014, Continuity rising to 0.906, and Alignment dropping to 0.027, while MSE improves by 14.5%. Beyond this threshold, geometric metrics continue to tighten but MSE degrades monotonically as consistency pressure overrides the reconstruction signal. The key finding is a sharp *phase transition* at $\lambda_c = 0.01$: the minimum value achieving $\beta_0 = 1$ and passing all five verification metrics, identifiable without any target-domain labels.

## 1.2 Effect of $\lambda_c$ on Multi-Domain Integration (Application 2)

Table 2 reports the effect of $\lambda_c$ on the three-domain integration task (MNIST, USPS, SVHN), which requires simultaneously aligning three latent distributions rather than two.

At $\lambda_c = 0$ the collapse is more severe than in Application 1: $\beta_0 = 3$ indicates three disconnected clusters, Trust and Continuity are both 0.602, and Alignment is 0.988.

At $\lambda_c = 0.01$ the manifold unifies ($\beta_0 = 1$), Trust rises to 0.922, Continuity to 0.933, and Alignment drops to 0.083, while Avg MSE improves from 0.01452 to 0.01266. Beyond this value, geometric metrics tighten marginally but MSE degrades monotonically, with Trust and Continuity declining from 0.922/0.933 at $\lambda_c = 0.01$ to 0.901/0.907 at $\lambda_c = 1.0$, reflecting stronger competition between consistency and reconstruction objectives when three distributions must be aligned simultaneously. The optimal $\lambda_c$ is again 0.01 — the minimum achieving $\beta_0 = 1$.

Table 2: Sensitivity to $\lambda_c$ on multi-domain integration (MNIST, USPS, SVHN). † indicates the default value. At $\lambda_c = 0$ the manifold fails verification despite $W_2$ passing, demonstrating that $W_2$ alone is insufficient.

| $\lambda_c$ | Avg MSE | $\mathcal{L}_{\text{contra}}$ | $\mathcal{L}_{\text{cen}}$ | $\beta_0$ | Trust $\tau_t$ | Cont. $\tau_c$ | Align. $\tau_a$ | $W_2$ |
|---|---|---|---|---|---|---|---|---|
| 0.0 | 0.01452 | 4.2769 | 0.0168 | 3 | 0.602 | 0.602 | 0.988 | 0.8324 |
| 0.01† | 0.01266 | 0.6341 | 0.0089 | 1 | 0.922 | 0.933 | 0.083 | 0.004232 |
| 0.1 | 0.01273 | 0.4359 | 0.0046 | 1 | 0.914 | 0.928 | 0.034 | 0.003587 |
| 0.5 | 0.01306 | 0.3859 | 0.0052 | 1 | 0.908 | 0.912 | 0.023 | 0.003281 |
| 1.0 | 0.01396 | 0.3848 | 0.0050 | 1 | 0.901 | 0.907 | 0.027 | 0.003168 |

The higher Trust and Continuity relative to Application 1 are consistent with the lower intra-class variance of USPS and SVHN: USPS comprises centered, normalized postal digit scans, and SVHN consists of tightly cropped, standardized real-world digit images, both of which impose cleaner neighborhood structure on the shared latent space than handwritten MNIST alone.

## 1.3 Effect of $\lambda_\ell$ on Zero-Shot Classification (Application 3)

Table 3 reports the effect of varying the local topology preservation weight $\lambda_\ell$ on zero-shot classification accuracy across three datasets of increasing visual complexity.

Table 3: Sensitivity to $\lambda_\ell$ on zero-shot classification accuracy (%) for unseen classes 5–9. † indicates the default value used in the main paper.

| $\lambda_\ell$ | MNIST | Fashion-MNIST | CIFAR-10 |
|---|---|---|---|
| 0.0 | 78.48 | 75.82 | 67.41 |
| 0.1 | 89.19 | 84.81 | 79.01 |
| 0.5† | 89.47 | 84.70 | 78.76 |
| 1.0 | 89.22 | 84.78 | 79.84 |

Without local topology preservation ($\lambda_\ell = 0$), accuracy drops substantially across all three datasets (78.48%, 75.82%, 67.41% for MNIST, Fashion-MNIST, and CIFAR-10 respectively), confirming that $\mathcal{L}_{\text{local}}$ is critical for zero-shot transfer.

All three datasets exhibit non-monotonic responses: MNIST and Fashion-MNIST peak at $\lambda_\ell = 0.5$ (89.47% and 84.70% respectively) before declining slightly at $\lambda_\ell = 1.0$, while CIFAR-10 dips at $\lambda_\ell = 0.5$ (78.76%) and recovers to its best at $\lambda_\ell = 1.0$ (79.84%). This reflects the varying alignment between pixel-space neighborhood structure and semantic class boundaries across datasets: MNIST and Fashion-MNIST benefit most from moderate topology enforcement, whereas the higher visual complexity of CIFAR-10 requires stronger preservation to overcome intra-class variance.

## 1.4 Effect of $\lambda_p$, $\lambda_{\text{cls\_m}}$, and $\lambda_{\text{cls\_f}}$ on Cross-Domain Transfer (Application 2)

Table 4 reports a joint sensitivity analysis over the perceptual loss weight $\lambda_p$ and the domain-specific classifier weights $\lambda_{\text{cls\_m}}$ (MNIST) and $\lambda_{\text{cls\_f}}$ (Fashion-MNIST) in the cross-domain transfer setting. Each run varies one weight while holding the others fixed at the baseline ($\lambda_p = 0.3$, $\lambda_c = 0.1$, $\lambda_{\text{cls\_m}} = 0.3$, $\lambda_{\text{cls\_f}} = 0.7$). The $\lambda_c$ rows from this setting are already reported in Table 2 and are omitted here.

Table 4: Joint sensitivity analysis for cross-domain transfer. Each group varies one weight; all others held at baseline (marked †): $\lambda_p = 0.3$, $\lambda_{\text{cls\_m}} = 0.3$, $\lambda_{\text{cls\_f}} = 0.7$. Accuracy reported on 500 held-out samples per domain.

| $\lambda_p$ | $\lambda_{\text{cls\_m}}$ | $\lambda_{\text{cls\_f}}$ | MNIST→F-MNIST | F-MNIST→MNIST |
|---|---|---|---|---|
| *Varying $\lambda_p$* | | | | |
| 0.1 | 0.3 | 0.7 | 84.63 | 95.12 |
| 0.2 | 0.3 | 0.7 | 87.41 | 97.23 |
| 0.3† | 0.3 | 0.7 | 86.73 | 96.97 |
| 0.5 | 0.3 | 0.7 | 85.18 | 95.43 |
| 1.0 | 0.3 | 0.7 | 83.52 | 94.38 |
| *Varying $\lambda_{\text{cls\_m}}$* | | | | |
| 0.3 | 0.1 | 0.7 | 83.47 | 96.84 |
| 0.3 | 0.3† | 0.7 | 86.73 | 96.97 |
| 0.3 | 0.5 | 0.7 | 88.12 | 96.21 |
| *Varying $\lambda_{\text{cls\_f}}$* | | | | |
| 0.3 | 0.3 | 0.5 | 86.91 | 94.83 |
| 0.3 | 0.3 | 0.7† | 86.73 | 96.97 |
| 0.3 | 0.3 | 0.9 | 85.24 | 97.68 |

**Effect of $\lambda_p$.** Performance is stable in the range $\lambda_p \in [0.2, 0.3]$, with both transfer directions remaining above 86.7% and 96.9% respectively. Beyond $\lambda_p = 0.3$, both directions degrade monotonically: at $\lambda_p = 1.0$, MNIST→F-MNIST drops to 83.52% and F-MNIST→MNIST to 94.38%, as over-weighting the perceptual loss shifts the encoder toward texture and style features that suppress the structural geometry critical for cross-domain transfer. The default $\lambda_p = 0.3$ sits at the upper edge of this stable plateau, balancing perceptual regularization with the reconstruction objective across all applications.

**Effect of $\lambda_{\text{cls\_m}}$.** Increasing $\lambda_{\text{cls\_m}}$ from 0.1 to 0.5 monotonically improves MNIST→F-MNIST from 83.47% to 88.12%, as stronger source domain supervision tightens latent class boundaries and sharpens decision boundaries transferred to Fashion-MNIST. The trade-off is a slight decrease in F-MNIST→MNIST from 96.97% to 96.21% at $\lambda_{\text{cls\_m}} = 0.5$, reflecting mild over-specialization toward MNIST geometry. The default $\lambda_{\text{cls\_m}} = 0.3$ balances these competing directions.

**Effect of $\lambda_{\text{cls\_f}}$.** The two transfer directions respond oppositely to $\lambda_{\text{cls\_f}}$: increasing from 0.5 to 0.9 monotonically improves F-MNIST→MNIST from 94.83% to 97.68% while degrading MNIST→F-MNIST from 86.91% to 85.24%. The default asymmetry $\lambda_{\text{cls\_f}} > \lambda_{\text{cls\_m}}$ (0.7 vs. 0.3) reflects the higher visual complexity of Fashion-MNIST and balances the trade-off between the two transfer directions.

## 1.5 Effect of Loss Weights on Pass/Fail Rate of Algorithm 1

Table 5 summarizes the explicit Pass/Fail outcome of Algorithm 1 for the $\lambda_c$ verification sweeps in Applications 1 and 2. A configuration passes if and only if all active thresholds are simultaneously satisfied: $\beta_0 = 1$, $\tau_t \geq 0.80$, $\tau_c \geq 0.70$, $W_2 \leq 0.30$, and $\tau_a \leq 0.30$ where applicable.

Three conclusions follow directly from Table 5. First, within the explicit verification sweeps reported here, $\mathcal{L}_{\text{consist}}$ is the sole load-bearing objective for global manifold unification: only $\lambda_c = 0$ causes

Table 5: Pass/Fail summary of Algorithm 1 for $\lambda_c$ ablation. Only the explicit $\lambda_c$ verification sweeps are shown here. The $\lambda_\ell$ and $\lambda_p$ sweeps reported earlier in Section 1 emphasize downstream sensitivity rather than a full verification-metric table. † marks the default value used in the main paper.

| Weight | Value | $\beta_0$ | Trust | Cont. | $W_2$ | Align. | Pass |
|---|---|---|---|---|---|---|---|
| | 0.0 | 2 | 0.788 | 0.624 | 0.623 | 0.704 | ✗ |
| | $0.01^\dagger$ | 1 | 0.800 | 0.906 | 0.014 | 0.027 | ✓ |
| $\lambda_c$ (App. 1) | 0.1 | 1 | 0.807 | 0.912 | 0.024 | 0.009 | ✓ |
| | 0.5 | 1 | 0.803 | 0.901 | 0.007 | 0.004 | ✓ |
| | 1.0 | 1 | 0.804 | 0.896 | 0.004 | 0.002 | ✓ |
| | 0.0 | 3 | 0.602 | 0.602 | 0.8324 | 0.988 | ✗ |
| | $0.01^\dagger$ | 1 | 0.922 | 0.933 | 0.004232 | 0.083 | ✓ |
| $\lambda_c$ (App. 2) | 0.1 | 1 | 0.914 | 0.928 | 0.003587 | 0.034 | ✓ |
| | 0.5 | 1 | 0.908 | 0.912 | 0.003281 | 0.023 | ✓ |
| | 1.0 | 1 | 0.901 | 0.907 | 0.003168 | 0.027 | ✓ |

verification failure in Applications 1 and 2, while the other ablations reported in Section 1 show smoother task degradation rather than an abrupt gate failure. The $\lambda_\ell$ and $\lambda_p$ sweeps are included as downstream-sensitivity analyses and therefore are not assigned explicit Pass/Fail outcomes in this summary table. Second, the Pass/Fail boundary is sharp in both applications, transitioning between $\lambda_c = 0$ and $\lambda_c = 0.01$ with no partial passes, reflecting the binary nature of $\beta_0$ connectivity in the $k$-NN graph.

## 1.6 Baseline Comparability Across Tasks

To support the matched-comparison claim in the revised main paper, we summarize below the supervision assumptions and data-access rules used in our baseline reimplementations. Because all baselines within a given application share the same protocol, a paragraph summary is clearer than a repeated table. In every case, baselines are trained on the same dataset splits as ULHM and evaluated on the same held-out test data.

**Sparse recovery.** The sparse-recovery baselines are Attention U-Net, Dense U-Net, DANet, Res-Attention, CBAM U-Net, ResUNet-PSP, and ResUNet. All are trained as paired supervised reconstruction models from sparse inputs to full targets, using the same sparse/complete training pairs as ULHM, with no semantic attributes or extra labels. All are evaluated on the same held-out reconstruction split under standard test-time evaluation.

**Cross-domain transfer.** The cross-domain transfer baselines are PConv, MCD, CDSL, MCC, CDAN, CORAL, and ProtoNet. All use source-domain labels together with unlabeled target-domain observations during adaptation, with the same MNIST/Fashion-MNIST training splits as ULHM, no target labels, and no extra pretraining data. All are evaluated on the same held-out transfer split, with no additional target-domain fine-tuning at test time.

**Transductive zero-shot classification.** The zero-shot baselines are SimCLR, ProtoNet, Triplet Networks, VAE, MCD, CORAL, Mixup, and Rotation Prediction. All use seen-class labels only,

while unseen-class images remain unlabeled. All are given the same transductive split as ULHM: all images are available to the representation learner, but labels are restricted to seen classes $\{0, \ldots, 4\}$. All are evaluated using the same unseen-class test protocol on $\{5, \ldots, 9\}$, and are therefore not directly comparable to fully inductive zero-shot settings.

## 2 Negative Control Experiments

The verification protocol in Algorithm 1 is only useful if it correctly rejects structurally incompatible domain pairs while accepting compatible ones. We present two negative control experiments probing distinct failure modes against the compatible MNIST/USPS/SVHN baseline.

**Experimental design.** We construct two incompatible three-domain configurations.

- **MNIST + Sine + ScrambledMNIST.** SineWave images are globally periodic with random labels, carrying no semantic structure. ScrambledMNIST applies per-sample pixel scrambling plus a checkerboard pattern, destroying all spatial structure while preserving label cardinality.

- **MNIST + Random Noise + R-MNIST.** Random Gaussian noise provides a structureless baseline. R-MNIST shares identical pixel statistics with MNIST but has randomly permuted class labels, making it the critical control: successful transfer here would implicate label supervision rather than homeomorphic latent structure as the driver of cross-domain transfer.

All configurations are trained with identical objectives and hyperparameters (reconstruction weight $= 10.0$, consistency weight $= 0.1$, centroid weight $= 0.1$).

Table 6: Verification metrics for compatible versus incompatible domain triples.

| Configuration | $\beta_0$ | Trust | Cont. | $W_2$ | Align. | **Pass** |
|---|---|---|---|---|---|---|
| MNIST+USPS+SVHN (compat.) | 1 | 0.922 | 0.933 | 0.004 | 0.083 | ✓ |
| MNIST+Sine+Scrambled | 2 | 0.315 | 0.315 | 0.0013 | 0.733 | ✗ |
| MNIST+Noise+R-MNIST | 3 | 0.352 | 0.352 | 0.0012 | 0.817 | ✗ |

**Verification results.** The two incompatible configurations fail four of five metrics: $\beta_0 > 1$, Trust $< 0.40$, Continuity $< 0.40$, and Alignment $> 0.70$. The compatible baseline passes all five. Notably, $W_2$ passes in all three configurations, confirming that distributional co-location is necessary but not sufficient for manifold compatibility and must be interpreted jointly with Trust, Continuity, and $\beta_0$.

**Transfer results.** In the compatible configuration, cross-domain transfer is strong in all directions: USPS and SVHN are recovered at $\geq 83\%$ regardless of classifier source, reflecting the shared latent structure confirmed by the verification protocol. In both incompatible configurations, transfer to and from non-MNIST domains collapses to near-chance ($\leq 12\%$) while within-MNIST accuracy remains high ($\geq 96\%$), with the protocol correctly predicting this failure before deployment.

**The R-MNIST control rules out label supervision.** Near-chance transfer in all directions involving R-MNIST (10.57–11.50%), combined with verification failure (Trust $= 0.352$, Alignment $= 0.817$), directly falsifies the hypothesis that label supervision alone drives cross-domain transfer: homeomorphic latent structure is the necessary condition, and the protocol correctly identifies

Table 7: Zero-shot cross-domain transfer accuracy (%) for compatible and incompatible configurations. Rows indicate the target domain; columns indicate the source of the classifier.

*Compatible: MNIST + USPS + SVHN*

| Target ↓ / Classifier → | MNIST | USPS | SVHN |
|---|---|---|---|
| MNIST | 97.37 | 96.10 | 97.40 |
| USPS | 98.36 | 98.36 | 97.81 |
| SVHN | 86.46 | 83.79 | 86.87 |

*Incompatible: MNIST + Sine + Scrambled*

| Target ↓ / Classifier → | MNIST | Sine | Scrambled |
|---|---|---|---|
| MNIST | 96.70 | 9.77 | 10.13 |
| Sine | 10.33 | 8.77 | 10.33 |
| Scrambled | 10.83 | 10.83 | 9.97 |

*Incompatible: MNIST + Noise + R-MNIST*

| Target ↓ / Classifier → | MNIST | Noise | R-MNIST |
|---|---|---|---|
| MNIST | 96.67 | 10.27 | 11.50 |
| Noise | 9.90 | 9.83 | 9.90 |
| R-MNIST | 11.13 | 10.57 | 10.93 |

when it is absent. The joint reading of $\beta_0$ and Trust further resolves ambiguity: $\beta_0 > 1$ with failed Trust indicates genuine fragmentation, whereas $\beta_0 > 1$ with passing Trust and $W_2$ indicates a finite-sample artifact of the cosine $k$-NN graph at the chosen $k$.

# 3 Scalability of the Verification Protocol

A practical concern for large-scale deployment is whether reliable metric estimates require the full dataset or whether small subsamples suffice. We evaluate convergence of all five metrics on the MNIST/USPS/SVHN setting ($\rho = 0.15$), drawing 10 independent subsamples at each size and reporting mean ± standard deviation (Figure 1).

The five metrics exhibit qualitatively different convergence behaviors. $\beta_0$ is perfectly stable from $n = 50$ onward with zero variance, requiring the fewest samples of any metric. $W_2$ converges rapidly to 0.004 by $n = 500$. Trust stabilizes at 0.923–0.924 from $n = 2,000$ onward and never falls below the $\tau_t \geq 0.80$ threshold at any size, confirming reliable Pass/Fail decisions from $n \geq 1,000$. Continuity exhibits the largest finite-sample bias, increasing monotonically from 0.630 at $n = 50$ as the data-space $\kappa$-NN graph densifies; $n \geq 2,000$ is required for reliable estimates. Alignment drops from 0.089 to 0.021 by $n = 3,000$, already close to its converged value by $n = 1,000$ (0.027).

**Practical recommendation.** Reliable estimates are obtained at $n = 2,000$ for all metrics ($n = 500$ suffices for $\beta_0$ and $W_2$). All local metrics additionally admit efficient approximate computation via FAISS-based nearest neighbor search, scaling sublinearly in $n$.

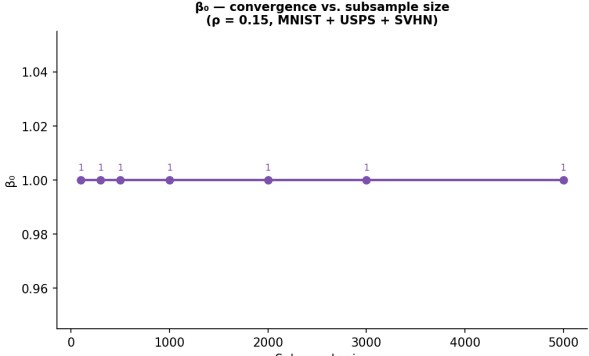

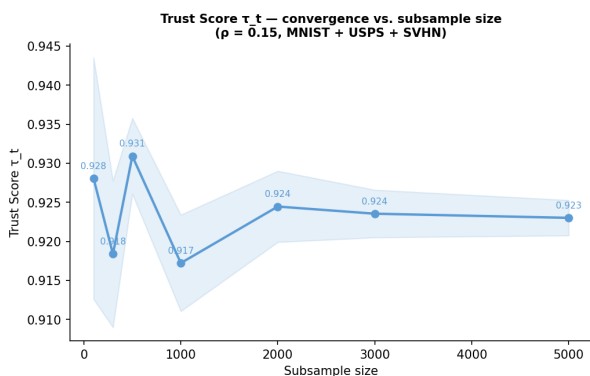

(a) $\beta_0$ — stable at $\beta_0 = 1$ from $n = 50$ onward with zero variance.

(b) Trust $\tau_t$ — stabilizes at 0.923–0.924 from $n = 2,000$ onward.

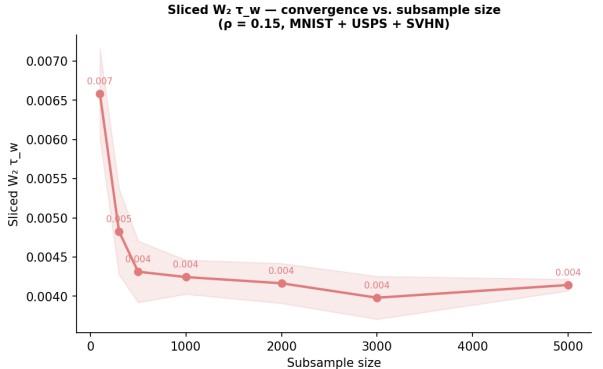

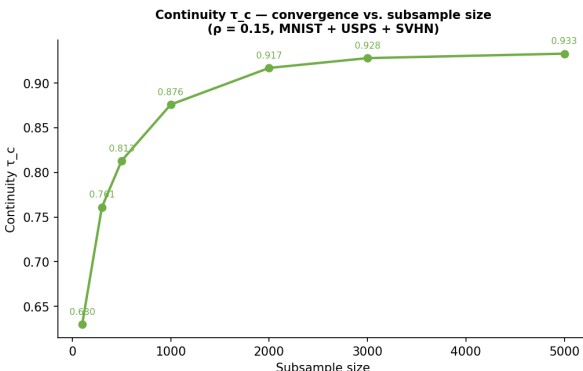

(c) Sliced $W_2$ — converges from 0.007 at $n = 50$ to 0.004 by $n = 500$.

(d) Continuity $\tau_c$ — largest finite-sample bias, converging from 0.630 at $n = 50$ to 0.933 by $n = 5,000$.

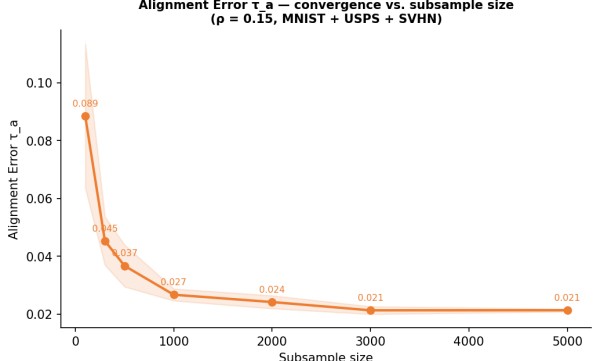

(e) Alignment $\tau_a$ — converges from 0.089 at $n = 50$ to 0.021 by $n = 3,000$.

Figure 1: Convergence of all five verification metrics as a function of subsample size on the MNIST/USPS/SVHN integration setting ($\rho = 0.15$, $\kappa = 5$). Shaded regions show $\pm 1$ standard deviation over 10 independent subsamples.

## 4 Additional Domain Experiments

### 4.1 fastMRI Brain

#### 4.1.1 Dataset and Setup

We evaluate the ULHM framework on the fastMRI Brain dataset Knoll et al. (2020); Zbontar et al. (2018), a large-scale multi-coil MRI benchmark presenting a genuinely heterogeneous semantic-

observation pair: the observation modality consists of undersampled k-space measurements, while the semantic modality combines full RSS reconstructions with pathology annotations. Unlike the image-space masking used in the CelebA experiments (main paper, Section 4.2.2), undersampling is applied directly in the frequency domain, where the signal domain differs fundamentally between observation and semantic spaces.

**Observation modality.** For each slice we apply a uniform Cartesian undersampling mask to the multi-coil k-space data and compute the RSS reconstruction from the undersampled coil images:

$$\tilde{x}^{(2)} = \mathrm{RSS}\Big(\mathcal{F}^{-1}\Big(m \odot \hat{x}^{(1)}_{\mathrm{coil}}\Big)\Big), \qquad \mathrm{RSS}(y) = \sqrt{\textstyle\sum_c |y_c|^2}, \tag{1}$$

where $m$ is the binary undersampling mask, $\hat{x}^{(1)}_{\mathrm{coil}}$ denotes the multi-coil k-space data, and $\mathcal{F}^{-1}$ is the inverse 2D Fourier transform. This yields a single-channel magnitude image of dimension $320 \times 320$.

**Semantic modality.** The semantic modality comprises two components: (i) the fully-sampled RSS reconstruction $s^{(1)}$ obtained from complete k-space data, and (ii) a binary pathology token $s^{(2)} \in \{1, 2\}$ derived from bounding-box annotations in the accompanying `brain.csv` file, where token 2 indicates a pathological slice and token 1 a normal slice. The pathology token is embedded via a learned embedding layer and concatenated with spatial features in the full-image encoder $E_s^{(l)}$ before projecting to the latent manifold $\mathcal{M}_z$. This pairing is genuinely heterogeneous: the observation lives in frequency space while the semantic ground truth combines image-space structure with hierarchical clinical annotation, spanning two fundamentally different signal domains.

**Framework training.** We train on the official `multicoil_train` split and evaluate on the held-out `multicoil_val` and `multicoil_test` splits (2,492 axial slices) across five sampling rates $\rho \in \{0.05, 0.10, 0.15, 0.20, 0.25\}$. We apply the ULHM objective (Eq. 8 of the main paper) with the full RSS reconstruction as the primary semantic ground truth. The cross-modal consistency loss uses cosine similarity only, and the model is selected by minimum validation MSE.

**Baseline and evaluation metrics.** The zero-filled RSS reconstruction from undersampled k-space serves as the baseline, corresponding to the standard reference for accelerated MRI reconstruction without any learned prior. We report MSE ($\downarrow$), L1 ($\downarrow$), and SSIM ($\uparrow$) on 2,492 test slices. For noise robustness we consider two injection sites: image space (post-RSS) and k-space (pre-reconstruction), each at mild ($30\,\mathrm{dB}$) and moderate ($20\,\mathrm{dB}$) SNR. SSIM is not reported for k-space noise conditions, where the severe frequency-domain corruption renders it undefined in the conventional sense; MSE and L1 remain well-defined and are reported instead.

### 4.1.2 Clean Sparse Recovery Results

Table 8 presents reconstruction quality under noiseless conditions across five sampling rates. Figure 2 shows a qualitative example at $\rho = 0.05$, illustrating the aliasing artifacts present in the zero-filled baseline and the anatomical detail recovered by ULHM. ULHM achieves consistent improvement over the zero-filled RSS baseline at all rates, with the largest gains at extreme undersampling.

At $\rho = 0.05$, ULHM achieves MSE $= 0.001818$ (**+57.0%**), L1 $= 0.025452$ (**+32.1%**), and SSIM $= 0.9509$ (**+0.84%**). The MSE advantage decreases monotonically as more k-space data become

| 5% k-space | Original | Ours |
|:---:|:---:|:---:|

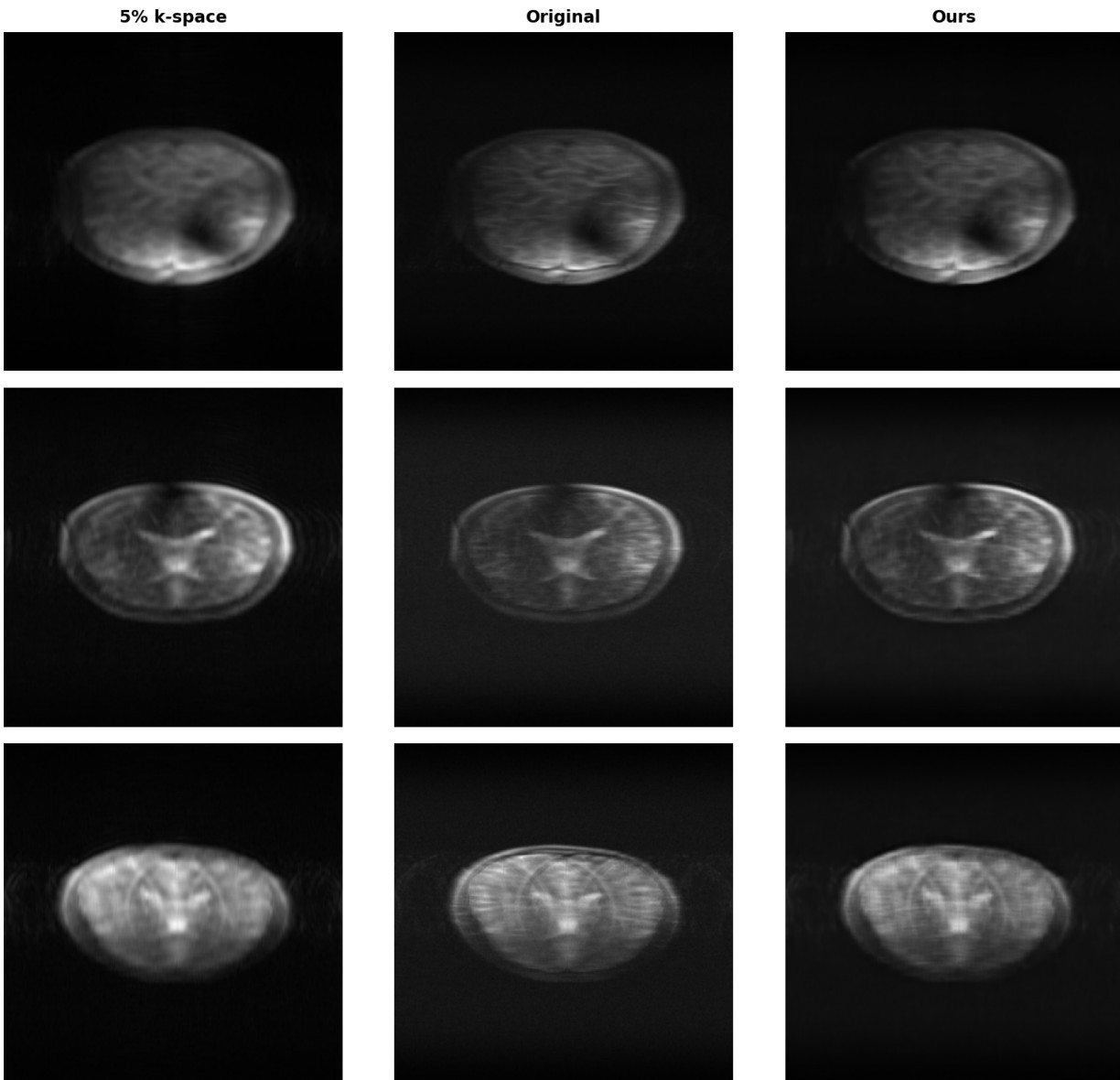

Figure 2: Visual comparison of reconstructed brain MRI slices at $\rho = 0.05$ (5% k-space retention). The zero-filled RSS baseline exhibits severe aliasing artifacts, while ULHM recovers fine anatomical detail closely matching the fully-sampled reference.

available: +48.7% at $\rho = 0.10$, +40.7% at $\rho = 0.15$, +33.8% at $\rho = 0.20$, and +25.5% at $\rho = 0.25$. L1 follows the same trend, narrowing from +32.1% to +2.2%. SSIM improvements are smaller in magnitude (+0.26%–+0.84%), consistent with SSIM's emphasis on structural correlation rather than mean-intensity fidelity, so the manifold-guided gains are more pronounced in MSE and L1.

### 4.1.3   Robustness to Noise: Rician Noise (Image Space)

Rician noise is the canonical noise model for magnitude MRI images. We inject Rician noise in image space (post-RSS) at mild (30 dB) and moderate (20 dB) SNR and evaluate across all five sampling rates.

Table 8: fastMRI Brain clean sparse recovery across five k-space sampling rates $\rho$ (2,492 test slices). Baseline is zero-filled RSS. Improvement is the relative reduction in error (MSE, L1) or relative gain in SSIM over the baseline at matched $\rho$.

| $\rho$ | MSE ↓ | | | L1 ↓ | | | SSIM ↑ | | |
|---|---|---|---|---|---|---|---|---|---|
| | Base. | ULHM | Impr. | Base. | ULHM | Impr. | Base. | ULHM | Impr. |
| 0.05 | 0.00423 | **0.00182** | +57.0% | 0.03750 | **0.02545** | +32.1% | 0.9430 | **0.9509** | +0.84% |
| 0.10 | 0.00399 | **0.00205** | +48.7% | 0.03644 | **0.02862** | +21.5% | 0.9446 | **0.9513** | +0.71% |
| 0.15 | 0.00368 | **0.00218** | +40.7% | 0.03522 | **0.03004** | +14.7% | 0.9467 | **0.9522** | +0.58% |
| 0.20 | 0.00349 | **0.00231** | +33.8% | 0.03446 | **0.03184** | +7.6% | 0.9482 | **0.9527** | +0.48% |
| 0.25 | 0.00329 | **0.00245** | +25.5% | 0.03358 | **0.03282** | +2.2% | 0.9500 | **0.9525** | +0.26% |

Under mild noise (30 dB), ULHM reduces MSE by **55.2**% at $\rho = 0.05$, declining to 26.9% at $\rho = 0.25$, consistent with the clean setting. Under moderate noise (20 dB) the advantage is stronger and more sustained: **50.4**% MSE reduction at $\rho = 0.05$, remaining at 37.7% at $\rho = 0.25$. SSIM improvements are more pronounced than in the clean case (+2.93% at mild, +16.2% at moderate noise, $\rho = 0.05$), reflecting the severe structural degradation of the zero-filled baseline under moderate Rician corruption that the manifold geometry partially recovers.

Table 9: fastMRI Brain MSE, L1, and SSIM under Rician noise injected in image space at mild (30 dB) and moderate (20 dB) SNR (2,492 test slices).

| $\rho$ | MSE ↓ | | | L1 ↓ | | | SSIM ↑ | |
|---|---|---|---|---|---|---|---|---|
| | Base. | ULHM | Impr. | Base. | ULHM | Impr. | Base. | ULHM |
| *Mild noise (30 dB SNR)* | | | | | | | | |
| 0.05 | 0.00426 | **0.00191** | +55.2% | 0.03744 | **0.02669** | +28.7% | 0.9197 | **0.9467** |
| 0.10 | 0.00404 | **0.00211** | +47.9% | 0.03681 | **0.02931** | +20.4% | 0.9207 | **0.9473** |
| 0.15 | 0.00374 | **0.00221** | +41.0% | 0.03585 | **0.03036** | +15.3% | 0.9225 | **0.9485** |
| 0.20 | 0.00355 | **0.00233** | +34.3% | 0.03526 | **0.03207** | +9.0% | 0.9237 | **0.9490** |
| 0.25 | 0.00336 | **0.00246** | +26.9% | 0.03458 | **0.03290** | +4.9% | 0.9253 | **0.9490** |
| *Moderate noise (20 dB SNR)* | | | | | | | | |
| 0.05 | 0.00503 | **0.00250** | +50.4% | 0.04600 | **0.03041** | +33.9% | 0.7911 | **0.9189** |
| 0.10 | 0.00486 | **0.00257** | +47.2% | 0.04571 | **0.03132** | +31.5% | 0.7911 | **0.9206** |
| 0.15 | 0.00461 | **0.00256** | +44.5% | 0.04508 | **0.03145** | +30.2% | 0.7922 | **0.9224** |
| 0.20 | 0.00445 | **0.00263** | +40.8% | 0.04469 | **0.03228** | +27.8% | 0.7928 | **0.9234** |
| 0.25 | 0.00429 | **0.00267** | +37.7% | 0.04430 | **0.03250** | +26.6% | 0.7936 | **0.9241** |

*4.1.4   Robustness to Noise: Gaussian Noise (Image Space)*

We inject additive Gaussian noise in image space (post-RSS) at mild (30 dB) and moderate (20 dB) SNR and evaluate across all five sampling rates.

Under mild noise (30 dB), ULHM reduces MSE by **52.9**% at $\rho = 0.05$, declining to 29.6% at $\rho = 0.25$. Under moderate noise (20 dB), the improvement stabilizes around 34%–35% across all $\rho$, in contrast to the monotonically decreasing gains in the clean and mild-noise settings, reflecting a regime where baseline error is dominated by corruption rather than aliasing and additional k-space lines provide diminishing returns. The most striking result is SSIM: under moderate noise, ULHM

raises SSIM from 0.480 to 0.851 at $\rho = 0.05$ (**+77.4**%), directly confirming that the latent manifold captures structural geometry rather than merely pixel-level intensity.

Table 10: fastMRI Brain MSE, L1, and SSIM under additive Gaussian noise injected in image space at mild (30 dB) and moderate (20 dB) SNR (2,492 test slices).

| $\rho$ | MSE ↓ | | | L1 ↓ | | | SSIM ↑ | |
|---|---|---|---|---|---|---|---|---|
| | Base. | ULHM | Impr. | Base. | ULHM | Impr. | Base. | ULHM |
| *Mild noise (30 dB SNR)* | | | | | | | | |
| 0.05 | 0.00476 | **0.00224** | +52.9% | 0.04325 | **0.02984** | +31.0% | 0.8309 | **0.9319** |
| 0.10 | 0.00455 | **0.00244** | +46.3% | 0.04269 | **0.03238** | +24.2% | 0.8297 | **0.9333** |
| 0.15 | 0.00425 | **0.00253** | +40.5% | 0.04182 | **0.03332** | +20.3% | 0.8300 | **0.9348** |
| 0.20 | 0.00407 | **0.00264** | +35.1% | 0.04126 | **0.03473** | +15.8% | 0.8301 | **0.9357** |
| 0.25 | 0.00389 | **0.00274** | +29.6% | 0.04068 | **0.03544** | +12.9% | 0.8306 | **0.9362** |
| *Moderate noise (20 dB SNR)* | | | | | | | | |
| 0.05 | 0.00878 | **0.00576** | +34.4% | 0.06663 | **0.04682** | +29.7% | 0.4799 | **0.8514** |
| 0.10 | 0.00866 | **0.00567** | +34.5% | 0.06660 | **0.04681** | +29.7% | 0.4775 | **0.8538** |
| 0.15 | 0.00845 | **0.00546** | +35.4% | 0.06627 | **0.04623** | +30.2% | 0.4763 | **0.8559** |
| 0.20 | 0.00832 | **0.00543** | +34.8% | 0.06606 | **0.04620** | +30.1% | 0.4754 | **0.8567** |
| 0.25 | 0.00819 | **0.00529** | +35.5% | 0.06587 | **0.04577** | +30.5% | 0.4746 | **0.8580** |

### 4.1.5 Connection to Homeomorphic Structure

Three findings collectively support the core claim of Section 3.1. First, the 57.0% MSE reduction at $\rho = 0.05$ under clean conditions confirms that semantic priors embedded in the latent manifold during training implicitly guide reconstruction at test time without requiring any semantic input. Second, the Rician noise results show that this advantage strengthens under moderate noise, with 20 dB gains exceeding 30 dB gains by up to 10.8 percentage points at $\rho = 0.25$. Third, the +77.4% SSIM recovery under moderate Gaussian noise confirms that the manifold captures structural geometry, which is an advantage absent in the clean setting where gains decrease monotonically with $\rho$.

### 4.1.6 Homeomorphism Verification

We apply the hierarchical verification protocol (Algorithm 1) to the Brain MRI latent manifold at $\rho = 0.05$, $\kappa = 5$, cosine distance. Table 11 reports results across three levels. $\beta_0 = 1$ confirms a single connected manifold and $W_2 = 0.0072$ indicates geometric co-location of the two encoder distributions. Trust $\tau_t = 0.9189$ and Continuity $\tau_c = 0.8852$ confirm neighborhood preservation, and Alignment $\tau_a = 0.0021$ confirms semantic consistency between $Z_x$ and $Z_s$. All five metrics pass their respective thresholds, providing empirical evidence consistent with structural compatibility between the paired modalities.

## 4.2 Power Grid State Estimation

### 4.2.1 Dataset, Problem Setup, and Architecture

**Domain and motivation.** Power grid state estimation presents the strongest test of the Universal claim in ULHM, and simultaneously exposes the fundamental failure modes of existing transfer learning approaches that make zero-shot transfer not merely desirable but *necessary*.

Table 11: Homeomorphism verification: Brain MRI latent manifold.

| Level | Metric | Value |
|---|---|---|
| Global Match | Betti number $\beta_0$ | 1 |
| | Sliced $W_2$ | 0.0072 |
| Local Match | Trust $\tau_t$ | 0.9189 |
| | Continuity $\tau_c$ | 0.8852 |
| Semantic Match | Alignment $\tau_a$ | 0.0021 |

Existing approaches fail under power system reconfiguration due to two compounding challenges. First, every topology change simultaneously shifts feature distributions *and* alters input dimensionality: power flow physics couples all buses, so a single line outage propagates through the entire feature space, while dimensionality itself becomes variable as lines are removed or buses added. A model trained on $n$-dimensional inputs *cannot* process $(n \pm k)$-dimensional data without complete retraining — even graph neural networks designed for varying graph sizes fail, because they cannot simultaneously handle dimensional mismatch and physics-induced distribution shifts. Second, reconfiguration does not merely shift output distributions — it redefines the semantic meaning of outputs entirely. A state estimate at bus 5 in one topology has no meaningful correspondence to bus 5 after reconfiguration alters its connection pattern; when systems expand from $m$ to $m+p$ buses, the model must predict $p$ additional state variables with no training data. Traditional approaches that update parameters for new topologies inevitably overwrite prior knowledge, preventing true universality.

This exposes a critical design paradox: successful transfer requires invariant input features that generalize across domains, yet power system outputs are inherently configuration-dependent and *cannot* share representations across topologies. Unlike computer vision where outputs can be interpolated or padded, power system states must strictly satisfy Kirchhoff's laws — transfer is valid *only* when the latent manifold captures physical structure, not merely statistical distributions. **ULHM resolves this paradox through zero-shot homeomorphic transfer**: the homeomorphic latent manifold provides the configuration-invariant geometric structure that existing approaches cannot learn, enabling instant transfer to unseen topologies with no retraining, no fine-tuning, and no topology-specific information.

The observation modality consists of sparse AMI sensor readings (real-valued magnitude and power measurements from a subset of buses), while the semantic modality consists of complete complex voltage phasors governed by $Y_{\text{bus}}$, which changes with every topology; see Figure 3. Unlike every experiment in the main paper, the two modalities do not share a signal domain, making this the hardest instance of the Universal claim.

**Dataset and split.**  We generate 1,200 synthetic power grid instances based on the IEEE 33-bus distribution network, each with a distinct topology obtained by randomly varying line parameters, bus counts (14–38 buses), and loading profiles. Each instance comprises 4,800 time-step snapshots of complex voltage phasors $v \in \mathbb{C}^N$ simulated via full AC power flow. The dataset is split into 1,000 training, 100 validation, and 100 test topologies, where test topologies are *entirely unseen* by graph structure, bus count, and admittance matrix.

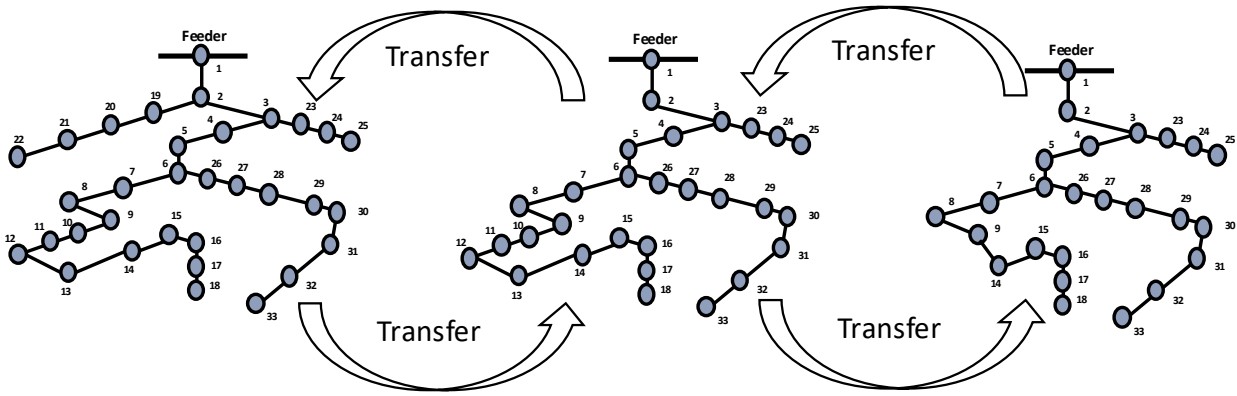

Figure 3: Zero-shot transfer across power grid topologies.

**Observation and semantic modalities.** Each sensor bus provides four real-valued measurements: $x_i^{(m)} = [|v_i|, P_i, Q_i, |I_i|] \in \mathbb{R}^4$, with binary mask $m_i \in \{0, 1\}$ indicating sensor presence. The semantic modality is the full complex voltage state $s = v \in \mathbb{C}^N$, encoded by a Complex GCN using Chebyshev graph convolutions with the normalized complex graph Laplacian and admittance-valued edge weights.

**Architecture.** The semantic encoder processes complex voltage phasors through two layers of complex Chebyshev graph convolutions ($K = 4$), followed by hybrid pooling and projection to $\mathbb{R}^{128}$. The observation encoder is a Transformer operating on sensor buses only, augmented with $Y_{\text{bus}}$ eigenvector features for structural context, projecting to the same $d = 128$ latent space. A shared decoder reconstructs $\hat{v} \in \mathbb{C}^N$ for any number of buses using learned positional queries, enabling generalization across topologies of varying size.

**Training.** We apply the ULHM objective (Eq. 8 of the main paper) with cosine consistency only. All voltages and AMI features are normalized globally across the training set, and the best checkpoint is selected by minimum validation loss.

### 4.2.2  State Estimation Results: Effect of AMI Coverage

Table 12 reports zero-shot voltage phasor reconstruction RMSE on the 100 held-out test topologies across AMI coverage levels. RMSE decreases monotonically from 0.01044 at 10% to 0.00851 at 50% coverage (**18.5**% reduction), confirming that the framework correctly extracts more state information as additional sensor buses become available — *without any retraining*. All results are obtained on entirely unseen test topologies.

### 4.2.3  Zero-Shot Transfer Across Unseen Topologies

The zero-shot transfer setting here is strictly harder than standard domain adaptation benchmarks: the model receives no samples, no labels, no fine-tuning steps, and no topology-specific information from the target graphs whatsoever. At 10% AMI coverage, where only 1 in 10 buses carries a sensor, the system is severely under-determined and classical estimators cannot produce a valid solution without topology information unavailable at test time. ULHM achieves RMSE = 0.01044 under precisely these conditions, outperforming NR-WLS (0.01544) and DistFlow (0.04402) — neither of

Table 12: Zero-shot state estimation RMSE on 100 unseen test topologies across AMI coverage levels. RMSE is in complex phasor units; inference time (ms) is the average per-snapshot wall-clock time. NR-WLS and DistFlow are **non-transferable**: they require the exact ground-truth network topology and must re-optimise from scratch for every new graph.

| | Ours (Zero-Shot) | | NR-WLS[‡] | | DistFlow[‡] | |
|---|---|---|---|---|---|---|
| **AMI Coverage** $\rho_{\mathrm{AMI}}$ | RMSE ↓ | Time (ms) | RMSE ↓ | Time (ms) | RMSE ↓ | Time (ms) |
| 10% | 0.01044 | 0.785 | 0.01544 | 18.81 | 0.04402 | 0.587 |
| 20% | 0.00942 | 0.793 | 0.01002 | 35.59 | 0.02848 | 0.708 |
| 30% | 0.00917 | 0.792 | 0.00815 | 83.73 | 0.02703 | 0.869 |
| 40% | 0.00852 | 0.810 | 0.00795 | 108.82 | 0.02603 | 0.970 |
| 50% | 0.00851 | 0.815 | 0.00767 | 138.68 | 0.02332 | 0.987 |

[‡]Not zero-shot transferable: requires exact topology, re-optimised per graph.
Ours: test topology entirely unseen during training, constant ≈0.80 ms inference.

which is applicable in a true zero-shot setting — at ≈0.80 ms inference time regardless of topology size.

### 4.2.4 Homeomorphism Verification

We apply Algorithm 1 to the best checkpoint using the joint latent pool of $z_x$ and $z_s$ encodings from the test set (288,000 samples, 25 unique graph sizes).

Table 13: Homeomorphism verification for the power grid latent manifold ($\kappa = 5$, $N = 288,000$ test samples).

| Level | Metric | Value |
|---|---|---|
| Global Match | $\beta_0$ | 1 |
| | Sliced $W_2$ | 0.003 |
| Local Match | Trust $\tau_t$ | 0.950 |
| | Continuity $\tau_c$ | 0.778 |
| Semantic Match | Alignment | 0.010 |

$\beta_0 = 1$ and $W_2 = 0.003$ confirm topological connectivity and geometric co-location across all 25 graph sizes. Trust $\tau_t = 0.950$ and Continuity $\tau_c = 0.778$ confirm neighborhood preservation without manifold collapse, and Alignment = 0.010 confirms that paired $(x_i, s_i)$ observations map to nearby latent regions despite the fundamental signal-domain difference. All five metrics pass their thresholds, providing a principled geometric explanation for the zero-shot transfer success: the configuration-invariant manifold structure learned on 1,000 topologies transfers instantly to 100 entirely unseen ones.

### 4.2.5 Why This Domain Tests the Universal Claim

Three properties of the power grid setting jointly constitute the hardest test of the Universal claim.

*Signal domain heterogeneity.* Real-valued sparse AMI readings and complex-valued phasors gov-

erned by a network admittance operator live in fundamentally different mathematical spaces — a cross-domain pairing that no existing transfer framework addresses without domain-specific adaptation.

*Topology variation with varying dimensionality.* The manifold must unify representations that vary in input dimension across 25 distinct graph sizes, with no shared labeling scheme, dimensionality, or semantic vocabulary between training and test topologies beyond AC power flow physics — a capability absent from all fixed-dimension image benchmarks.

*Physical meaningfulness of semantic structure.* The semantic modality is the solution to the power flow equations determined entirely by the admittance topology. Unlike vision tasks where outputs can be interpolated or padded, power system states must strictly satisfy Kirchhoff's laws — transfer is valid only when the latent manifold captures physical structure, not merely statistical distributions. The monotonically decreasing RMSE across coverage levels and the passing verification metrics (Trust = 0.950, $\beta_0 = 1$, Alignment = 0.010) are consistent with the learned manifold generalizing physically meaningful structure across unseen topologies.