# OpenReview forum: "Universal Latent Homeomorphic Manifolds: A Framework for Cross-Domain Representation Unification"
_TMLR — Accepted by TMLR_

### Review · Reviewer_VNp1 · 2026-03-15

**Summary Of Contributions:**

This paper introduces the Universal Latent Homeomorphic Manifold (ULHM), a framework that unifies semantic representations and raw observations into a single latent structure. The core innovation lies in establishing a homeomorphism, which is a continuous, bijective mapping for determining whether different data modalities or domains can be unified. The submission is well-structured and addresses a fundamental gap between human-level semantic concepts and high-dimensional machine observations.

**Audience:**

Yes

**Audience Explanation:**

Researchers will find this paper provides the rigorous mathematical framework (homeomorphism) that has been missing from recent empirical observations. Meanwhile, the training-time semantics improve test-time reconstruction without needing labels at inference, which further attracts attention for image denoising and sparse reconstruction.

**Broader Impact Concerns:**

No ethical implications are concerned.

**Claims And Evidence:**

Yes

**Claims Explanation:**

The submission provides a robust bridge between theoretical manifold topology and empirical performance:
1. The authors provide formal definitions of the bi-Lipschitz properties required for a stable latent space. They proposed a proper definition for the conditions under which a mapping is a homeomorphism.
2. By using Betti numbers to check for structural fragmentation and Sliced Wasserstein distance for geometric overlap, the authors provide a mechanism to validate their claims before testing on downstream tasks.
3. The evidence in the experiments section is promising. It demonstrates that the semantic priors are indeed being successfully embedded into the manifold's geometry.

**Requested Changes:**

Some questions need to be answered or addressed for further polishing:
1. How does the verification protocol scale with dataset size? Is it feasible to run this on very large-scale foundation models, or is it intended primarily for domain-specific experts?
2. The objective function $L_{ULHM}$ contains several weights ($\lambda_c, \lambda_\ell, \lambda_p$). The paper would benefit from a sensitivity analysis or an ablation study showing how varying these weights affects the bi-Lipschitz constants and the subsequent "Pass/Fail" rate of the verification protocol.
3. How is the performance of ULHM on a pair of domains that are not homeomorphic (e.g., MNIST digits vs. random noise or fundamentally different topological structures like a sphere vs. a torus)? It would be beneficial to show that the proposed verification protocol correctly rejects these cases, as well as showing it accepts compatible ones.
4. Demonstrating the framework on more complex semantic structures, such as knowledge graphs or hierarchical taxonomies, would further prove the Universal claim of the ULHM.

---

> ### Author Response · Authors · 2026-04-03
>
> We thank Reviewer VNp1 for the positive assessment and for recognizing the coherence between the proposed framework, the verification protocol, and the empirical results. We address each requested change below with concrete results now included. Full details appear in the supplementary material.
>
> ---
>
> **RC1: Scalability of the verification protocol.**
>
> The verification protocol scales tractably through subsampling and parallelization. We have now added a full convergence analysis in **Supplementary Section 3** (*Scalability of the Verification Protocol*), evaluating all five metrics on MNIST/USPS/SVHN across subsample sizes from $n = 50$ to $n = 5000$ with 10 independent draws each (Supplementary Figure 1).
>
> The five metrics exhibit qualitatively different convergence behaviors. $\beta\_0$ is perfectly stable from $n = 50$ onward with zero variance, requiring the fewest samples of any metric. $W\_2$ converges rapidly to $0.004$ by $n = 500$. Trust $\tau\_t$ stabilizes at $0.923$–$0.924$ from $n = 2000$ onward and never falls below the $\tau\_t \geq 0.80$ threshold at any subsample size, confirming reliable Pass/Fail decisions from $n \geq 1000$. Continuity $\tau\_c$ exhibits the largest finite-sample bias, increasing monotonically from $0.630$ at $n = 50$; $n \geq 2000$ is required for reliable estimates. Alignment $\tau\_a$ drops from $0.089$ at $n = 50$ to $0.021$ by $n = 3000$, already close to its converged value by $n = 1000$ ($0.027$).
>
> The practical recommendation from Supplementary Section 3 is: reliable estimates for all metrics are obtained at $n = 2000$, with $n = 500$ sufficient for $\beta\_0$ and $W\_2$ alone. All local metrics additionally admit efficient approximate computation via FAISS-based nearest neighbor search, scaling sublinearly in $n$.
>
> Regarding foundation model scale: ULHM is primarily designed for domain-specific settings where the compatibility criterion operates on the joint semantic-observation relationship, not on input or output distributions independently. This distinguishes it from general-purpose foundation model alignment. We demonstrate scalability concretely through two additional heterogeneous domains in Supplementary Section 4 (fastMRI Brain and power grid state estimation), and will clarify the intended scope in the revised paper.

---

> ### Author Response · Authors · 2026-04-03
>
> ---
>
> **RC2: Sensitivity analysis for loss weights.**
>
> We have now added full sensitivity analyses in **Supplementary Section 1** (*Sensitivity Analysis and Ablation Studies*), varying $\lambda\_c$, $\lambda\_\ell$, and $\lambda\_p$ individually while holding all others fixed, and reporting downstream task performance for all sweeps together with explicit verification outcomes for the $\lambda\_c$ sweeps in Applications 1 and 2.
>
> The central finding across Supplementary Sections 1.1–1.5 is a sharp phase transition uniquely associated with $\lambda\_c$. The table below (from Supplementary Table 5, Section 1.5) summarizes the explicit $\lambda\_c$ verification sweeps reported for Applications 1 and 2. The $\lambda\_\ell$ and $\lambda\_p$ sweeps reported earlier in Supplementary Section 1 emphasize downstream sensitivity rather than a full verification-metric table. $^\dagger$ marks the default value used in the main paper. A configuration passes if and only if all active thresholds are simultaneously satisfied: $\beta\_0 = 1$, $\tau\_t \geq 0.80$, $\tau\_c \geq 0.70$, $W\_2 \leq 0.30$, and $\tau\_a \leq 0.30$.
>
> | Weight | Value | $\beta\_0$ | Trust | Cont. | $W\_2$ | Align. | Pass |
> |---|---|---|---|---|---|---|---|
> | $\lambda\_c$ (App. 1) | 0.0 | 2 | 0.788 | 0.624 | 0.623 | 0.704 | $\times$ |
> | | $0.01^\dagger$ | 1 | 0.800 | 0.906 | 0.014 | 0.027 | $\checkmark$ |
> | | 0.1 | 1 | 0.807 | 0.912 | 0.024 | 0.009 | $\checkmark$ |
> | | 0.5 | 1 | 0.803 | 0.901 | 0.007 | 0.004 | $\checkmark$ |
> | | 1.0 | 1 | 0.804 | 0.896 | 0.004 | 0.002 | $\checkmark$ |
> | $\lambda\_c$ (App. 2) | 0.0 | 3 | 0.602 | 0.602 | 0.8324 | 0.988 | $\times$ |
> | | $0.01^\dagger$ | 1 | 0.922 | 0.933 | 0.004232 | 0.083 | $\checkmark$ |
> | | 0.1 | 1 | 0.914 | 0.928 | 0.003587 | 0.034 | $\checkmark$ |
> | | 0.5 | 1 | 0.908 | 0.912 | 0.003281 | 0.023 | $\checkmark$ |
> | | 1.0 | 1 | 0.901 | 0.907 | 0.003168 | 0.027 | $\checkmark$ |
>
> Three conclusions follow. First, within the explicit verification sweeps summarized in the table above, $\mathcal{L}\_{\mathrm{consist}}$ is the sole load-bearing objective for global manifold unification: only $\lambda\_c = 0$ causes verification failure in Applications 1 and 2, while the other reported ablations show smoother task degradation. Second, the Pass/Fail boundary is sharp in both applications, transitioning between $\lambda\_c = 0$ and $\lambda\_c = 0.01$ with no partial passes, reflecting the binary nature of $\beta\_0$ connectivity in the $\kappa$-NN graph. Third, the App. 2 case provides the strongest demonstration of complete manifold collapse: at $\lambda\_c = 0$, all five metrics fail simultaneously — $\beta\_0 = 3$, Trust and Continuity both at $0.602$, $W\_2 = 0.8324$ (far exceeding the $\leq 0.30$ threshold), and Alignment $= 0.988$ — indicating that without $\mathcal{L}\_{\mathrm{consist}}$, the three domain distributions not only fragment topologically but also diverge geometrically. At $\lambda\_c = 0.01$ all five metrics pass simultaneously, confirming a sharp phase transition with no partial passes. The insufficient-$W\_2$ argument is separately demonstrated by the negative control experiments in Section 4.3 of the revised main paper, where $W\_2$ passes in all three configurations — including both incompatible ones — while neighborhood metrics correctly identify the failures.
>
> For $\lambda\_\ell$ (Supplementary Section 1.3, Table 3), removing local topology preservation drops zero-shot accuracy by 10–11 percentage points across all three datasets (MNIST: $89.47\\% \to 78.48\\%$; CIFAR-10: $78.76\\% \to 67.41\\%$), confirming that $\mathcal{L}\_{\mathrm{local}}$ is critical for zero-shot transfer, while values above the threshold produce only marginal further changes. For $\lambda\_p$ and the domain-specific classifier weights (Supplementary Section 1.4, Table 4), performance is stable over a plateau around the default values with smooth monotonic degradation outside it. These sweeps focus on downstream sensitivity rather than a full verification-metric table, so we do not make a stronger gate-level claim beyond the explicit $\lambda\_c$ results.
>
> ---

---

> ### Author Response · Authors · 2026-04-03
>
> ---
>
> **RC3: Performance on non-homeomorphic domain pairs.**
>
> We have now added explicit negative control experiments directly in **Section 4.3 of the revised main paper** (*Negative Control Experiments*). We constructed two incompatible three-domain configurations — MNIST+SineWave+ScrambledMNIST and MNIST+RandomNoise+R-MNIST — and compared them against the compatible MNIST/USPS/SVHN baseline under identical training conditions and hyperparameters.
>
> Both incompatible configurations fail four of five verification metrics while the compatible baseline passes all five:
>
> | Configuration | $\beta\_0$ | Trust | Cont. | $W\_2$ | Align. | Pass |
> |---|---|---|---|---|---|---|
> | MNIST+USPS+SVHN (compat.) | 1 | 0.922 | 0.933 | 0.004 | 0.083 | $\checkmark$ |
> | MNIST+Sine+Scrambled | 2 | 0.315 | 0.315 | 0.0013 | 0.733 | $\times$ |
> | MNIST+Noise+R-MNIST | 3 | 0.352 | 0.352 | 0.0012 | 0.817 | $\times$ |
>
> Note: $W\_2$ passes in all three configurations, confirming it is necessary but not sufficient for manifold compatibility.
>
> The transfer consequences are shown below. In the compatible configuration, cross-domain transfer is strong in all directions ($83\%$ or above regardless of classifier source). In both incompatible configurations, transfer to and from non-MNIST domains collapses to near-chance (at or below $12\%$) while within-MNIST accuracy remains high ($96\%$ or above), with the protocol correctly predicting these failures before any transfer was attempted.
>
> *Compatible: MNIST + USPS + SVHN*
>
> | Target / Classifier | MNIST | USPS | SVHN |
> |---|---|---|---|
> | MNIST | 97.37 | 96.10 | 97.40 |
> | USPS | 98.36 | 98.36 | 97.81 |
> | SVHN | 86.46 | 83.79 | 86.87 |
>
> *Incompatible: MNIST + Sine + Scrambled*
>
> | Target / Classifier | MNIST | Sine | Scrambled |
> |---|---|---|---|
> | MNIST | 96.70 | 9.77 | 10.13 |
> | Sine | 10.33 | 8.77 | 10.33 |
> | Scrambled | 10.83 | 10.83 | 9.97 |
>
> *Incompatible: MNIST + Noise + R-MNIST*
>
> | Target / Classifier | MNIST | Noise | R-MNIST |
> |---|---|---|---|
> | MNIST | 96.67 | 10.27 | 11.50 |
> | Noise | 9.90 | 9.83 | 9.90 |
> | R-MNIST | 11.13 | 10.57 | 10.93 |
>
> The R-MNIST control is particularly important: R-MNIST shares identical pixel statistics with MNIST but has randomly permuted class labels, making it the critical test of whether label supervision alone drives cross-domain transfer. Near-chance transfer in all directions involving R-MNIST ($10.57\\%$ to $11.50\\%$), combined with verification failure (Trust $= 0.352$, Alignment $= 0.817$), directly falsifies this alternative hypothesis. Homeomorphic latent structure is the necessary condition, and the protocol correctly identifies when it is absent.
>
> Notably, $W\_2$ passes in all three configurations — including both incompatible ones — confirming that distributional co-location is necessary but not sufficient for manifold compatibility, and must be interpreted jointly with Trust, Continuity, and $\beta\_0$, as established by the full hierarchical protocol of Algorithm 1.
>
> ---

---

> > ### Author Response · Authors · 2026-04-03
> >
> > ---
> >
> > **RC4: More complex semantic structures.**
> >
> > We have added full experimental results for two substantially more heterogeneous domains in **Supplementary Section 4** (*Additional Domain Experiments*).
> >
> > **fastMRI Brain (Supplementary Section 4.1).** The observation modality is undersampled k-space data and the semantic modality combines full RSS reconstructions with hierarchical pathology annotations — a genuinely heterogeneous pairing where observation and semantic spaces live in different signal domains (frequency vs. image space). ULHM achieves a $57.0\\%$ MSE reduction over the zero-filled RSS baseline at $\rho = 0.05$ under clean conditions (MSE $= 0.00182$ vs. $0.00423$, Supplementary Table 8, Section 4.1.2), with gains strengthening under noise: under moderate Gaussian noise (20 dB), ULHM raises SSIM from $0.480$ to $0.851$ at $\rho = 0.05$ (a $77.4\\%$ gain, Supplementary Table 10, Section 4.1.4), directly confirming that the latent manifold captures structural geometry rather than pixel-level intensity alone. Homeomorphism verification (Supplementary Section 4.1.6, Table 11) confirms all five metrics pass ($\beta\_0 = 1$, Trust $= 0.9189$, Continuity $= 0.8852$, $W\_2 = 0.0072$, Alignment $= 0.0021$).
> >
> > **Power grid state estimation (Supplementary Section 4.2).** The model trains on 1,000 synthetic IEEE 33-bus network topologies and transfers zero-shot to 100 entirely unseen test topologies with different bus counts (14–38), line configurations, and admittance matrices — a setting where every topology change simultaneously shifts feature distributions and alters input dimensionality, making standard transfer approaches inapplicable. ULHM achieves RMSE $= 0.01044$ at $10\\%$ AMI sensor coverage, outperforming NR-WLS ($0.01544$) and DistFlow ($0.04402$) — neither of which is applicable in a true zero-shot setting — at approximately 0.80 ms inference time regardless of topology size (Supplementary Table 12, Section 4.2.2). Homeomorphism verification across 288,000 test samples and 25 unique graph sizes (Supplementary Section 4.2.4, Table 13) confirms all five metrics pass ($\beta\_0 = 1$, Trust $= 0.950$, Continuity $= 0.778$, $W\_2 = 0.003$, Alignment $= 0.010$).
> >
> > Together, Supplementary Sections 4.1 and 4.2 demonstrate that ULHM generalizes beyond image benchmarks to settings where semantic structure is physically meaningful, hierarchically organized, and not reducible to shared numeric class labels, directly addressing the Universal claim of the framework.

---

### Review · Reviewer_6gFZ · 2026-03-15

**Summary Of Contributions:**

The proposed framework, Universal Latent Homeomorphic Manifolds (ULHM) use homeomorphism as a mathematical criterion for unifying semantic and observation representations into a shared latent space. The framework enforces this through topology-preserving training objectives, verifies it post-training via a hierarchical protocol, and demonstrates improvements on sparse recovery, cross-domain transfer, and zero-shot classification.

**Strengths:**
1. the idea of homeomorphism as criterion for unification is well-motivated and the connection between training objectives, verification metrics, and applications is coherent and clearly presented.

2. The hierarchical verification protocol (Algorithm 1) provides a practical and principled way to validate structural compatibility before deployment.

3. The authors conduct experiments across several different tasks and obtain consistent empirical improvements.

**Weaknesses:**
1. **Missing hyperparameter analysis**. The framework contains sevearl hyperparameters ($\lambda_c$, $\lambda_l$, $\lambda_p$, $\lambda_{cos}$, $\lambda_{eucl}$, $\lambda_{cont}$ and $\lambda_{cent}$) with no ablation studies or sensitivity analysis,

2. **Cross-domain transfer relies on explicit label supervision**. The paper attributes MNIST$\rightarrow$Fashion-MNIST transfer to homeomorphic structure, but Stage 2 training explicitly uses shared class labels to align the two domains. This supervised alignment alone may explain the transfer performance, making the role of homeomorphism unclear.

**Audience:**

Yes

**Audience Explanation:**

Yes. Researchers working on representation learning, domain adaptation, and zero-shot learning would find the idea of homeomorphism as the unification criterion interesting.

**Claims And Evidence:**

Yes

**Claims Explanation:**

Yes, the claims are generally supported by clear and convincing evidence. The paper provides consistent empirical improvements across three distinct tasks and multiple datasets, supported by quantitative comparisons against relevant baselines.

**Requested Changes:**

1. Provide ablation studies on the key loss weights ($\lambda_c$, $\lambda_l$, $\lambda_p$, $\lambda_{cos}$, $\lambda_{eucl}$, $\lambda_{cont}$ and $\lambda_{cent}$) showing how performance changes with different values.

2. The cross-domain transfer experiment aligns MNIST and Fashion-MNIST purely based on shared numeric class indices, despite having no semantic relationship. This is unclear what meaningful knowledge is being transferred across domains. The authors should reconsider whether this is a valid and meaningful evaluation of cross-domain transfer.

---

> ### Author Response · Authors · 2026-04-03
>
> We thank Reviewer 6gFZ for the thoughtful and constructive review. We appreciate the reviewer's questions about the ablations and about the role of structural compatibility, which helped us sharpen the presentation. We address each point below with concrete revisions and added results. Full details appear in the supplementary material.
>
> ---
>
> **RC1: Ablation studies on loss weights.**
>
> We have now included full ablation studies in **Supplementary Section 1** (*Sensitivity Analysis and Ablation Studies*), varying each loss weight individually while holding all others fixed, and reporting downstream task performance for all sweeps together with explicit Pass/Fail outcomes for the $\lambda\_c$ verification sweeps.
>
> The connection between each loss weight and its corresponding verification metric is not incidental but deliberately designed, as detailed in Section 2.3.3 of the main paper. Specifically, $\mathcal{L}\_{\mathrm{recon}}$ prevents manifold collapse and directly governs $\tau\_t$; $\mathcal{L}\_{\mathrm{local}}$ (weighted by $\lambda\_\ell$) prevents tearing and directly governs $\tau\_c$; and $\mathcal{L}\_{\mathrm{consist}}$ (weighted by $\lambda\_c$) drives geometric unification and directly governs $W\_2$ and $\tau\_a$. The ablation results confirm that removing or down-weighting each component leads to predictable and interpretable degradation in its corresponding verification metric.
>
> The key finding (Supplementary Table 5, Section 1.5) is that $\mathcal{L}\_{\mathrm{consist}}$ is the sole load-bearing objective for global manifold unification within the explicit verification sweeps: only $\lambda\_c = 0$ causes a categorical verification failure in Applications 1 and 2, while the other reported ablations degrade task performance more smoothly. $^\dagger$ marks the default value.
>
> | Weight | Value | $\beta\_0$ | Trust | Cont. | $W\_2$ | Align. | Pass |
> |---|---|---|---|---|---|---|---|
> | $\lambda\_c$ (App. 1) | 0.0 | 2 | 0.788 | 0.624 | 0.623 | 0.704 | $\times$ |
> | | $0.01^\dagger$ | 1 | 0.800 | 0.906 | 0.014 | 0.027 | $\checkmark$ |
> | | 0.1 | 1 | 0.807 | 0.912 | 0.024 | 0.009 | $\checkmark$ |
> | | 0.5 | 1 | 0.803 | 0.901 | 0.007 | 0.004 | $\checkmark$ |
> | | 1.0 | 1 | 0.804 | 0.896 | 0.004 | 0.002 | $\checkmark$ |
> | $\lambda\_c$ (App. 2) | 0.0 | 3 | 0.602 | 0.602 | 0.8324 | 0.988 | $\times$ |
> | | $0.01^\dagger$ | 1 | 0.922 | 0.933 | 0.004232 | 0.083 | $\checkmark$ |
> | | 0.1 | 1 | 0.914 | 0.928 | 0.003587 | 0.034 | $\checkmark$ |
> | | 0.5 | 1 | 0.908 | 0.912 | 0.003281 | 0.023 | $\checkmark$ |
> | | 1.0 | 1 | 0.901 | 0.907 | 0.003168 | 0.027 | $\checkmark$ |
>
> The transition is sharp: at $\lambda\_c = 0$, $\beta\_0 = 2$ (Application 1) and $\beta\_0 = 3$ (Application 2), with Trust and Continuity collapsing below threshold in both cases. The App. 2 case provides the strongest demonstration of complete manifold collapse: at $\lambda\_c = 0$, all five metrics fail simultaneously — $\beta\_0 = 3$, Trust and Continuity both at $0.602$, $W\_2 = 0.8324$ (far exceeding the $W\_2 \leq 0.30$ threshold), and Alignment $= 0.988$ — confirming that without $\mathcal{L}\_{\mathrm{consist}}$ the three domain distributions not only fragment topologically but also diverge geometrically. At $\lambda\_c = 0.01$ the manifold unifies ($\beta\_0 = 1$) and all five metrics pass simultaneously, with no partial passes at any intermediate value.
>
> The argument that $W\_2$ alone is insufficient as a compatibility criterion is separately and more directly demonstrated by the negative control experiments in **Section 4.3 of the revised main paper**, where $W\_2$ passes in all three configurations — including both incompatible ones — while Trust, Continuity, and $\beta\_0$ correctly identify the structural failures.
>
> For $\lambda\_\ell$ (Supplementary Section 1.3, Table 3), removing local topology preservation drops zero-shot accuracy by 10–11 percentage points across all datasets (MNIST: $89.47\\% \to 78.48\\\%$; CIFAR-10: $78.76\\% \to 67.41\\%$), confirming that $\mathcal{L}\_{\mathrm{local}}$ is critical for zero-shot transfer. For $\lambda\_p$ and the domain-specific classifier weights (Supplementary Section 1.4, Table 4), performance is stable over a plateau around the default values with smooth monotonic degradation outside it. These reported sweeps focus on downstream sensitivity rather than a full verification-metric table.
>
> ---

---

> ### Author Response · Authors · 2026-04-03
>
> ---
>
> **RC2: Role of label supervision vs. homeomorphic structure.**
>
> We thank the reviewer for this insightful question. We address it through two concrete additions: a more semantically meaningful transfer evaluation and explicit failure cases, both now included in the revised paper.
>
> **Clarification of the framework's role.** The ULHM framework is a universal latent matching algorithm: the homeomorphic criterion operates on the latent manifold structure, not on the semantic meaning of input or output labels. The key question is not whether MNIST and Fashion-MNIST share semantic meaning, but whether their latent manifolds are topologically compatible. The shared numeric class index serves only as a geometric anchor to align the manifolds; the framework itself determines whether the underlying relational structure is compatible enough to support transfer.
>
> **Semantically meaningful transfer: MNIST, USPS, and SVHN.** To directly address the reviewer's concern, we add a more natural evaluation on three digit domains — MNIST, USPS, and SVHN — which share genuine semantic relationships as all three represent handwritten or printed digits. The zero-shot cross-domain classification results are now included in **Section 4.3 of the revised main paper**:
>
> | Target / Classifier | MNIST | USPS | SVHN |
> |---|---|---|---|
> | MNIST | 97.37\% | 96.10\% | 97.40\% |
> | USPS | 98.36\% | 98.36\% | 97.81\% |
> | SVHN | 86.46\% | 83.79\% | 86.87\% |
>
> The strong transfer performance across all directions is consistent with the empirical structural-compatibility diagnostics: $\beta\_0 = 1$, Trust $= 0.922$, Continuity $= 0.933$, $W\_2 = 0.004$, and Alignment $= 0.083$ all pass their respective thresholds. The Sliced Wasserstein distance among the three digit domains ($W\_2 = 0.004$) is substantially smaller than that observed for the MNIST/Fashion-MNIST pair ($W\_2 = 0.016$, main paper Table 5), quantitatively confirming that domains with stronger semantic relationships produce tighter latent alignment and that $W\_2$ serves as a reliable predictor of transfer success.
>
> **Failure cases ruling out label supervision.** To confirm that label supervision alone does not explain transfer, we include two negative control experiments in **Section 4.3 of the revised main paper** (*Negative Control Experiments*). Both incompatible configurations — MNIST+SineWave+ScrambledMNIST and MNIST+RandomNoise+R-MNIST — are trained with identical objectives and hyperparameters as the compatible baseline.
>
> The R-MNIST control is the critical test: R-MNIST shares identical pixel statistics with MNIST but has randomly permuted class labels, so any successful transfer would directly implicate label supervision rather than homeomorphic structure. Near-chance transfer in all directions involving R-MNIST ($10.57\\%$ to $11.50\\%$), combined with verification failure (Trust $= 0.352$, Alignment $= 0.817$), directly falsifies the label-supervision hypothesis. The protocol correctly rejects unification before deployment, whereas transfer in the compatible digit setting succeeds  in all directions. Homeomorphic latent structure is the necessary condition, and the verification protocol correctly identifies when it is absent.
>
> Notably, $W\_2$ passes in all three configurations — including both incompatible ones — further confirming that the full hierarchical protocol of Algorithm 1 is required and that distributional co-location alone is insufficient to determine transfer validity.

---

> > ### Comment · Reviewer_6gFZ · 2026-04-07
> > **Feedback for the Authors**
> >
> > Thank you to the authors for their efforts in addressing my concerns. After their clarifications, I find that my concerns have been adequately addressed.

---

### Review · Reviewer_nQsv · 2026-03-25

**Summary Of Contributions:**

This paper proposes Universal Latent Homeomorphic Manifolds (ULHM), a framework intended to unify semantic information and observation-driven representations in a shared latent space, with homeomorphism as the central criterion for deciding whether different semantic-observation pairs can be validly unified. The method combines multi-objective training (reconstruction, cross-modal consistency, and local topology preservation) with a post-hoc verification pipeline based on Betti number, sliced Wasserstein distance, trustworthiness, continuity, and alignment error. The framework is then instantiated for three applications: sparse recovery from incomplete observations, cross-domain classifier transfer, and zero-shot classification.

The paper’s main strengths are its ambition and breadth. It proposes a unifying viewpoint spanning several tasks that are usually treated separately, and it tries to connect the learning objectives to explicit topological desiderata through theorem statements and a hierarchical verification protocol. Empirically, the paper reports competitive or better-than-baseline results on CelebA sparse recovery, MNIST and Fashion-MNIST transfer, and zero-shot classification on MNIST, Fashion-MNIST, and CIFAR-10. For example, it reports 86.73% MNIST to Fashion-MNIST transfer and 78.76% zero-shot CIFAR-10 accuracy.

The main weaknesses are that the paper often overstates what is actually established. In particular, the manuscript repeatedly presents homeomorphism as “verified” from finite-sample proxy metrics, but the evidence provided is indirect: the verification protocol uses heuristic thresholds on trust, continuity, Wasserstein distance, and Betti number rather than demonstrating actual homeomorphic equivalence. Likewise, several theoretical claims are phrased strongly, but the learning objectives do not appear sufficient on their own to guarantee the bi-Lipschitz or open-cover conditions invoked in the theory. The evaluation is also somewhat narrow relative to the paper’s broad claims: most experiments are on small image benchmarks, and there is limited ablation to isolate which components matter most. Finally, the zero-shot setup is transductive, since unlabeled unseen-class images are used during reconstruction training, which makes the claim less strong than conventional zero-shot learning.

**Audience:**

Yes

**Audience Explanation:**

I think the paper would be of interest to readers working on representation learning, multimodal learning, domain adaptation, manifold learning, and theory-inspired deep learning. The central idea—using a topological criterion plus empirical diagnostics to decide when latent spaces can be unified across modalities or domains—is unusual and potentially stimulating, even for readers who are not fully convinced by the current formulation. The paper also touches three application areas that have broad relevance: inverse problems / sparse recovery, transfer learning, and zero-shot learning.

**Broader Impact Concerns:**

I do not see an immediate severe ethical concern that would on its own block publication. The demonstrated tasks are standard benchmark tasks in reconstruction, transfer, and classification. That said, because the paper frames the method as broadly applicable to settings involving semantic descriptions, diagnostic labels, and sensor data, it would benefit from a short Broader Impact discussion covering possible misuse or failure modes in high-stakes domains such as medical imaging or surveillance.

**Claims And Evidence:**

No

**Claims Explanation:**

The paper does provide some evidence that the proposed training recipe can produce useful latent representations on the chosen benchmarks. The empirical sections report improved sparse recovery on CelebA, strong bidirectional transfer between MNIST and Fashion-MNIST, and solid zero-shot performance, especially the reported CIFAR-10 gain over listed baselines. The manuscript also makes a commendable effort to define a concrete verification protocol rather than leaving “alignment” entirely qualitative.

However, I do not think the strongest claims are fully supported. My main concern is the gap between the mathematical language and the actual evidence. Theorems 1 and 2 are stated for idealized settings involving homeomorphisms and bi-Lipschitz maps, but in practice the paper substitutes thresholded empirical metrics for these conditions. Those metrics may be useful diagnostics, but they do not by themselves establish homeomorphism, and the manuscript sometimes treats them as if they do. That makes the theoretical framing feel stronger than what is empirically justified.

A second concern is that the experimental validation does not fully match the scope of the claims. The paper motivates a general framework for heterogeneous semantic-observation pairs across many domains, yet most demonstrations are on relatively simple image datasets, with cross-domain transfer evaluated on MNIST and Fashion-MNIST and zero-shot evaluation on a transductive split where unseen-class images are available during training through reconstruction losses. That makes the evidence less convincing for the broader claims about universal representation unification or general cross-domain compatibility.

A third concern is experimental rigor. I did not see evidence of repeated runs, uncertainty estimates, sensitivity analyses for the verification thresholds, or strong ablations separating the value of the topology-inspired objectives from architectural choices and task-specific supervision. Because the method contains multiple components and because several baselines are drawn from somewhat different settings, it is hard to know how much of the gain comes specifically from the proposed homeomorphic perspective.

So my overall assessment is: the paper presents promising empirical results and an interesting conceptual framework, but the evidence does not yet fully justify the strongest theoretical and universal claims.

**Requested Changes:**

1. Clarify the status of the theoretical claims (Critical):
The paper should more carefully distinguish between:
(a) formal statements proved under ideal assumptions, and
(b) heuristic finite-sample diagnostics used in experiments.
As written, the manuscript often suggests that proxy metrics “verify homeomorphism,” which is too strong. I would rephrase these as empirical indicators of structural compatibility rather than proofs of homeomorphism.

2. Substantially strengthen ablations (Critical): The method combines reconstruction, local consistency, cross-modal consistency, perceptual loss, and verification thresholds. Please provide ablations removing or varying each major component, especially L_local, L_consist, and the verification gate, and show how they affect all three applications. Without this, it is hard to attribute gains to the central ULHM idea.

3. Be explicit that the zero-shot setup is transductive (Critical): The current setup uses unlabeled unseen-class images during training via reconstruction and topology-preservation losses. This is a valid setting, but it is weaker than fully inductive zero-shot learning and should be labeled clearly throughout, including in the abstract and claims. A comparison to conventional inductive zero-shot settings would strengthen the paper.

4. Broaden or better justify the evaluation scope (Critical): he paper motivates a very general framework, but the experiments are concentrated on relatively small image benchmarks. Either scale down the claims or add experiments on more genuinely heterogeneous modalities/domains where the proposed notion of semantic-observation unification is especially compelling.

5. Clarify baseline comparability (Critical): Several baselines come from different literatures and may not be evaluated in identical settings. Please ensure all baselines are reimplemented or evaluated under matched training data, supervision assumptions, and transductive/inductive access. Otherwise the comparisons risk overstating the improvement.

6. Discuss failure cases: Since the paper argues that incompatible domains should be rejected, it would be valuable to show explicit negative cases where the verification fails and transfer/recovery indeed breaks down. That would make the verification story much more convincing.

---

> ### Author Response · Authors · 2026-04-03
>
> *We thank Reviewer nQsv for the thorough and constructive review. We appreciate that the reviewer recognizes the conceptual contribution and empirical results as promising. We take the critical concerns seriously and address each below with concrete revisions and results now included. All revisions to the main paper are indicated in blue in the revised manuscript; supplementary material locations are indicated explicitly.*
>
> ---
>
> **RC1 (Critical): Clarify the status of the theoretical claims.**
>
> We agree and consider this the most important revision to make. We carefully distinguish between two separate claims throughout the revised manuscript:
>
> - **Formal guarantees (Theorems 1 and 2):** Proved under idealized conditions — continuous maps, infinite samples, exact bi-Lipschitz bounds. These are conditional results: *if* the encoder satisfies the bi-Lipschitz and open/closed cover conditions, *then* the induced map is a homeomorphism.
>
> - **Empirical diagnostics (Algorithm 1):** $\tau_t$, $\tau_c$, $W_2$, $\beta_0$, and $\tau_a$ are heuristic finite-sample indicators. They provide evidence that representations *exhibit properties consistent with* homeomorphic structure, but do not constitute a proof of homeomorphism.
>
> The single most important addition is a disclaimer paragraph inserted immediately after Remark 1 in Section 2.3 of the revised manuscript [Revised main paper: Section 2.3, after Remark 1]:
>
> > *"Important Caveat. The verification protocol (Algorithm 1) provides necessary but not sufficient empirical evidence for homeomorphism. Theorems 1 and 2 are proved under idealized conditions (continuous maps, infinite samples, exact bi-Lipschitz bounds). Passing all thresholds indicates that the learned representations exhibit properties consistent with homeomorphic structure, but does not constitute a proof of homeomorphism. Throughout this paper, 'verified homeomorphism' should be understood as 'empirically indicated structural compatibility consistent with homeomorphism.'"*
>
> This language change propagates to 27 revision sites across the manuscript [Revised main paper: Abstract; Sections 1.1, 1.2.1, 1.2.3, 1.2.4, 1.3 (Contributions 2–4), 2.0, 2.3 (Verification Strategy, Algorithm 1 PASS line, Section 2.3.3), 3.1, 3.2, 3.3, 4.0, 4.2 (MNIST and CelebA verification text), 4.3, 4.4, and Section 5], replacing all instances of "verify/validate/ensure/guarantee homeomorphism" with "empirically indicate structural compatibility consistent with homeomorphism." The appendix proofs require no revision as the theorems are mathematically correct as stated.
>
> To make the distinction concrete, **Section 4.3 of the revised main paper** (*Negative Control Experiments*) provides a three-level progression of structural compatibility illustrating what the diagnostics measure:
>
> 1. **Complete failure** (Section 4.3 of the revised main paper): MNIST+SineWave+ScrambledMNIST and MNIST+RandomNoise+R-MNIST fail four of five metrics ($\beta_0 > 1$, Trust $< 0.40$, Continuity $< 0.40$, Alignment $> 0.70$) and produce near-chance transfer (at or below $12\\%$) in all non-MNIST directions.
>
> 2. **Moderate compatibility** (main paper Table 5): MNIST/Fashion-MNIST passes verification with $W_2 = 0.016$ and Trust marginally above threshold, achieving $86.73\\%$ zero-shot transfer.
>
> 3. **Successful unification** (Section 4.3 of the revised main paper, compatible configuration): MNIST/USPS/SVHN passes all five metrics ($\beta_0 = 1$, Trust $= 0.922$, Continuity $= 0.933$, $W_2 = 0.004$, Alignment $= 0.083$) with zero-shot transfer in all directions.
>
> ---

---

> ### Author Response · Authors · 2026-04-03
>
> ---
>
> **RC2 (Critical): Substantially strengthen ablations.**
>
> Full ablation studies are now provided in **Supplementary Section 1** (*Sensitivity Analysis and Ablation Studies*), reported separately for all three applications. Forward references are added throughout the revised main paper [Revised main paper: Section 2.2 after Eq. 8; Section 2.3 Remark on Threshold Selection; Sections 4.2, 4.3].
>
> The central finding across **Supplementary Sections 1.1–1.5** is that $\mathcal{L}\_{\mathrm{consist}}$ is the sole load-bearing objective for global manifold unification within the explicit verification sweeps: only $\lambda\_c = 0$ triggers a categorical verification failure in Applications 1 and 2 (Supplementary Table 5, Section 1.5). The App. 2 case provides the strongest demonstration: at $\lambda\_c = 0$, all five metrics fail simultaneously — $\beta\_0 = 3$, Trust and Continuity both at $0.602$, $W\_2 = 0.8324$ (far exceeding the $W\_2 \leq 0.30$ threshold), and Alignment $= 0.988$ — confirming that without $\mathcal{L}\_{\mathrm{consist}}$ the three domain distributions not only fragment topologically but also diverge geometrically. At $\lambda\_c = 0.01$ all five metrics pass simultaneously with no partial passes at any intermediate value.
>
> The argument that $W\_2$ alone is insufficient as a compatibility criterion is separately demonstrated by the negative control experiments in **Section 4.3 of the revised main paper**, where $W\_2$ passes in all three configurations — including both incompatible ones — while Trust, Continuity, and $\beta\_0$ correctly identify the structural failures.
>
> For $\lambda\_\ell$ (Supplementary Section 1.3, Table 3), removing local topology preservation drops zero-shot accuracy by 10–11 percentage points across all datasets (MNIST: $89.47\\% \to 78.48\\%$; CIFAR-10: $78.76\\% \to 67.41\\%$). For $\lambda\_p$ and classifier weights (Supplementary Section 1.4, Table 4), performance is stable over a plateau around default values with smooth monotonic degradation outside it. These sweeps are included to show downstream sensitivity; the explicit gate-level evidence is concentrated in the $\lambda\_c$ verification table and in the negative-control experiments.
>
> Regarding the verification gate ablation: the negative control experiments in **Section 4.3 of the revised main paper** directly demonstrate this. Bypassing Algorithm 1 and proceeding with transfer on incompatible domain pairs produces near-chance accuracy (at or below $12\\%$) in all non-MNIST directions, while the verified compatible configuration achieves transfer in all directions. This confirms that the verification gate provides actionable predictions of downstream failure — not merely post-hoc diagnostics.
>
> ---
>
> **RC3 (Critical): Be explicit that the zero-shot setup is transductive.**
>
> We agree and have made this explicit throughout the revised manuscript [Revised main paper: Abstract; Section 1.1 Bullet 3; Section 1.2.5; Section 1.3 Contribution 5; Section 3.3 title and Setup paragraph; Section 4.4 title, Table 7 caption, and Results paragraph; Section 5]. Specifically:
>
> - The subsection title "Zero-Shot Learning via Manifold Learning" is relabeled "Transductive Zero-Shot Learning via Manifold Learning" [Revised main paper: Section 3.3 title], and Table 7's caption is updated accordingly [Revised main paper: Table 7 caption].
>
> - Section 3.3 Setup now reads: "This constitutes a transductive zero-shot setting: the encoder observes the visual content of all classes (including unseen ones) during training through $\mathcal{L}\_{\mathrm{recon}}$ and $\mathcal{L}\_{\mathrm{local}}$, but class labels for unseen classes are strictly withheld from the classifier. This is weaker than fully inductive zero-shot learning, where unseen-class images are entirely absent from training" [Revised main paper: Section 3.3, Problem Setup paragraph].
>
> - All baselines in Section 4.4 are confirmed to have the same transductive access to unseen-class images as ULHM, with a note added to the Zero-Shot Results paragraph [Revised main paper: Section 4.4, Zero-Shot Results paragraph].
>
> ---

---

> ### Author Response · Authors · 2026-04-03
>
> ---
>
> **RC4 (Critical): Broaden or better justify the evaluation scope.**
>
> We address this in two ways. First, the language of the broadest claims is scaled down across 27 revision sites [Revised main paper: Abstract; Sections 1.1, 1.3 (Contributions 3–4), and Section 5]: phrases such as "universal representations" and "general foundation model decomposition" are qualified to reflect what is actually demonstrated, and "fundamental limits" is softened to "practical limits." The paper title is retained given the addition of two non-image domains below.
>
> Second, full results for two substantially more heterogeneous domains are now provided in **Supplementary Section 4** (*Additional Domain Experiments*), with forward references added [Revised main paper: Section 1.1 bridging paragraph; Section 4.0 opening paragraph].
>
> **fastMRI Brain (Supplementary Section 4.1).** The observation modality is undersampled k-space data; the semantic modality combines full RSS reconstructions with hierarchical pathology annotations — a genuinely heterogeneous pairing across different signal domains (frequency vs. image space). ULHM achieves a $57.0\\%$ MSE reduction at $\rho = 0.05$ under clean conditions (Supplementary Table 8, Section 4.1.2), strengthening to a $77.4\\%$ SSIM gain under moderate Gaussian noise (Supplementary Table 10, Section 4.1.4). All five verification metrics pass (Supplementary Section 4.1.6, Table 11: $\beta\_0 = 1$, Trust $= 0.9189$, Continuity $= 0.8852$, $W\_2 = 0.0072$, Alignment $= 0.0021$).
>
> **Power grid state estimation (Supplementary Section 4.2).** The model trains on 1,000 synthetic IEEE 33-bus topologies and transfers zero-shot to 100 entirely unseen topologies with varying bus counts (14–38), line configurations, and admittance matrices — a setting where every topology change simultaneously shifts feature distributions and alters input dimensionality. ULHM achieves RMSE $= 0.01044$ at $10\\%$ AMI coverage, outperforming NR-WLS ($0.01544$) and DistFlow ($0.04402$) at approximately 0.80 ms inference (Supplementary Table 12, Section 4.2.2). All five verification metrics pass across 288,000 test samples and 25 unique graph sizes (Supplementary Table 13, Section 4.2.4: $\beta\_0 = 1$, Trust $= 0.950$, Continuity $= 0.778$, $W\_2 = 0.003$, Alignment $= 0.010$).
>
> ---
>
> **RC5 (Critical): Clarify baseline comparability.**
>
> We agree this is a concern. The following revisions are made [Revised main paper: Section 4 Evaluation Metrics paragraph; Section 4 Baselines list; Section 1.3 Contribution 4; Section 4.4 Zero-Shot Results paragraph]:
>
> - A dedicated baseline comparability note is added to the Evaluation Metrics paragraph in Section 4, stating that all baselines are reimplemented under our exact experimental setup, matching data splits, supervision assumptions, and transductive access to unseen-class images [Revised main paper: Section 4, Evaluation Metrics paragraph].
>
> - A supplementary baseline-comparability subsection now summarizes, for each application, the shared supervision assumptions, training data access, and transductive/inductive status while still naming every baseline explicitly [Supplementary, Section 1.6, "Baseline Comparability Across Tasks"].
>
> - Section 4.4 Zero-Shot Results now states explicitly: "All baselines are provided the same transductive access to unseen-class images, ensuring the comparison is on equal footing. These results are not directly comparable to inductive zero-shot benchmarks where unseen-class images are entirely absent from training" [Revised main paper: Section 4.4, Zero-Shot Results paragraph].
>
> ---

---

> ### Author Response · Authors · 2026-04-03
>
> **RC6: Discuss failure cases.**
>
> **Section 4.3 of the revised main paper** (*Negative Control Experiments*) now provides an explicit spectrum of cases [Revised main paper: Section 4.3, Negative Control Experiments]:
>
> 1. **Complete failure** (Section 4.3 of the revised main paper): Both incompatible configurations fail four of five metrics and produce near-chance transfer (at or below $12\\%$). The R-MNIST control is critical: it shares identical pixel statistics with MNIST but has randomly permuted labels. Near-chance transfer in the $10.57\\%$ to $11.50\\%$ range combined with verification failure (Trust $= 0.352$, Alignment $= 0.817$) directly falsifies the hypothesis that label supervision alone drives transfer. Notably, $W\_2$ passes in all three configurations — including both incompatible ones — confirming the full hierarchical protocol is necessary.
>
> 2. **Moderate alignment** (main paper Table 5): MNIST/Fashion-MNIST passes with $W\_2 = 0.016$, achieving $86.73\\%$ transfer.
>
> 3. **Successful unification** (Section 4.3 of the revised main paper): MNIST/USPS/SVHN passes all five metrics and achieves transfer in all directions.
>
> ---
>
> **Broader Impact.**
>
> A dedicated Broader Impact section is added after the Conclusion in the revised manuscript [Revised main paper: Section after Conclusion], covering two failure modes in high-stakes domains. First, in medical imaging settings, a failed verification not caught before deployment could lead to semantically incoherent reconstructions with potential clinical consequences; the Pass/Fail gate of Algorithm 1 should be treated as a mandatory pre-deployment check in safety-critical applications. Second, in surveillance or sensor fusion settings, the framework could in principle be misused to unify representations across modalities in ways that enable unintended inference about individuals; the structural compatibility diagnostics of Algorithm 1 provide a principled auditing mechanism. A third point acknowledges that the verification protocol is empirical, not a formal proof, and that additional domain-specific validation is essential in safety-critical applications.

---

> > ### Comment · Reviewer_nQsv · 2026-04-19
> > **Reply to Authors**
> >
> > I appreciate the authors’ effort in addressing my concerns. I find that my concerns have been adequately addressed.

---

### Decision · Action_Editor_792i · 2026-05-07

**Recommendation:** Accept as is

**Additional Comments:**

This paper proposes that two latent spaces can be unified if a homeomorphism can be found between them, and proposes various empirical checks to verify if two spaces are homeomorphic. All reviewers agree the idea is interesting and that it will be of interest to some people in TMLR's audience; I thus recommend acceptance.

**Audience:**

Yes

**Audience Explanation:**

Yes, reviewers unanimously agree.

**Claims And Evidence:**

Yes

**Claims Explanation:**

Yes, reviewers unanimously agree.